# DAS²C: A Distributed Adaptive Minimax Method with Near-Optimal Convergence

## Abstract

Applying adaptive methods directly to distributed minimax problems can result in non-convergence due to inconsistency in locally computed adaptive stepsizes. To address this challenge, we propose DAS²C, a $\underline{D}$istributed $\underline{A}$daptive method with time-scale $\underline{S}$eparated $\underline{S}$tepsize $\underline{C}$ontrol for minimax optimization. The key strategy is to employ an adaptive stepsize control protocol involving the transmission of two extra (scalar) variables. This protocol ensures the consistency among stepsizes of nodes, eliminating the steady-state errors due to the lack of coordination of stepsizes among nodes that commonly exists in vanilla distributed adaptive methods, and thus guarantees exact convergence. For nonconvex-strongly-concave distributed minimax problems, we characterize the specific transient times that ensure time-scale separation of stepsizes and quasi-independence of networks, leading to a near-optimal convergence rate of $\tilde{\mathcal{O}}\left(\epsilon^{-(4+\delta)}\right)$ for any small $\delta > 0$, matching that of the centralized counterpart. To the best of our knowledge, DAS²C is the *first* distributed adaptive method guaranteeing exact convergence without requiring to know any problem-dependent parameters for nonconvex minimax problems.

## 1 Introduction

Distributed optimization has seen significant research progress over the last decade, resulting in numerous algorithms (Nedic and Ozdaglar, 2009; Yuan et al., 2016; Lian et al., 2017; Pu and Nedić, 2021). However, the traditional focus of distributed optimization has primarily been on minimization tasks. With the rapid growth of machine learning research, various applications have emerged that go beyond simple minimization, such as Generative Adversarial Networks (GANs) (Goodfellow et al., 2014; Gulrajani et al., 2017), robust optimization (Mohri et al., 2019; Sinha et al., 2017), adversary training of neural networks (Wang et al., 2021), fair machine learning (Madras et al., 2018), just to name a few. These tasks typically involve a minimax structure as follows

$$\min_{x \in \mathcal{X}} \max_{y \in \mathcal{Y}} f\left(x, y\right),$$

where $\mathcal{X} \subseteq \mathbb{R}^p$, $\mathcal{Y} \subseteq \mathbb{R}^d$, and $x, y$ are the primal and dual variables to be learned, respectively. One of the simplest yet effective methods for tackling minimax problems is Stochastic Gradient Descent Ascent (GDA) (Dem'yanov and Pevnyi, 1972; Nemirovski et al., 2009) which alternately performs stochastic gradient descent for the primal variable and stochastic gradient ascent for the dual variable. This approach has demonstrated its effectiveness in solving minimax problems, especially for convex-concave objectives (Hsieh et al., 2021; Daskalakis et al., 2021; Antonakopoulos et al., 2021), i.e., the function $f(\cdot, y)$ is convex for any $y \in \mathcal{Y}$, and $f(x, \cdot)$ is concave for any $x \in \mathcal{X}$.

Adaptive gradient methods, such as AdaGrad (Duchi et al., 2011), Adam (Kingma and Ba, 2014), and AMSGrad (Reddi et al., 2018), are often integrated with GDA to effectively solve minimax problems with theoretical guarantees in convex-concave settings (Diakonikolas, 2020; Antonakopoulos et al., 2021; Ene and Lê Nguyen, 2022). These adaptive methods are capable of adjusting stepsizes based on historical gradient information, making it robust to hyper-parameters tuning and can converge without requiring to know problem-dependent parameters (a characteristic often referred to as being "parameter-agnostic"). However, in the nonconvex regime, it has been shown by Lin et al. (2020); Yang et al. (2022b) that it is necessary to have a time-scale separation in stepsizes between the minimization and maximization processes to ensure the convergence of GDA and GDA-based

adaptive algorithms. In particular, the stepsize ratio between primal and dual variables needs to be smaller than a threshold depending on the properties of the problem such as the smoothness and strong-concavity parameters (Li et al., 2022; Guo et al., 2021; Huang et al., 2021), which are often unknown or difficult to estimate in real-world tasks like training deep neural networks.

Applying GDA-based adaptive methods into decentralized settings poses additional challenges due to the presence of inconsistency in locally computed adaptive stepsizes. In particular, it has been shown that the inconsistency of stepsizes can result in non-convergence in federated learning with heterogeneous computation speeds (Wang et al., 2020; Sharma et al., 2023). This is mainly due to the lack of a central node coordinating the stepsizes of nodes in distributed settings, making it difficult to converge, as observed in minimization problems (Liggett, 2022; Chen et al., 2023b). As a result, *the design of an adaptive minimax method capable of satisfying the time-scale separation requirement and being parameter-agnostic in fully distributed settings remains an open question.*

**Contributions.** In this paper, we aim to propose a distributed adaptive method for solving nonconvex-strongly-concave (NC-SC) minimax problems. The contributions are three folds:

- We construct counterexamples showing that directly applying adaptive methods designed for centralized problems might lead to inconsistencies in locally computed adaptive stepsizes, resulting in non-convergence in distributed settings. To tackle this issue, we propose the *first* distributed adaptive minimax method, named DAS²C, that incorporates an efficient stepsize control mechanism to maintain consistency across local stepsizes, which involves transmission of merely two extra (scalar) variables. The proposed algorithm exhibits time-scale separation in stepsizes and parameter-agnostic capability.

- Theoretically, we prove that DAS²C is able to achieve a near-optimal convergence rate of $\tilde{\mathcal{O}}\left(\epsilon^{-(4+\delta)}\right)$ with any small $\delta > 0$ to find an $\epsilon$-stationary point for distributed NC-SC problems. For comparison, we also prove the existence of a constant steady-state error in both the lower and upper bounds when directly applying a centralized adaptive algorithm without the stepsize control mechanism. Moreover, we characterize the specific transient times that ensure time-scale separation and quasi-independence of network respectively.

- We conduct extensive experiments on real-world datasets to verify our theoretical findings and the effectiveness of DAS²C on a variety of tasks, including the robust neural network training and optimizing Wasserstein GANs. In all tasks, we show the superiority of DAS²C comparing to several vanilla distributed adaptive methods across various graphs, initial stepsizes and data distributions (see also additional experiments in Appendix A).

## 1.1 RELATED WORKS

**Distributed nonconvex minimax methods.** In the realm of federated learning, Deng and Mahdavi (2021) introduce Local SGDA algorithm combining FedAvg/Local SGD with stochastic GDA and show an $\tilde{\mathcal{O}}\left(\epsilon^{-6}\right)$ sample complexity for NC-SC objective functions. Sharma et al. (2022) provide improved complexity result of $\tilde{\mathcal{O}}\left(\epsilon^{-4}\right)$ matching that of the lower bound (Li et al., 2021; Zhang et al., 2021a) for both NC-SC and nonconvex-Polyak-Lojasiewicz (NC-PL) settings. Yang et al. (2022a) combine Local SGDA with stochastic gradient estimators to eliminate the data heterogeneity. More recently, Zhang et al. (2023a) adopt compressed momentum methods with Local SGD to increase the communication efficiency of the algorithm. For decentralized nonconvex minimax problems, Liu et al. (2020) study the training of GANs using decentralized SGDA (D-SGDA) and provide non-asymptotic convergence with fixed stepsizes. Tsaknakis et al. (2020) propose a double-loop D-SGDA algorithm with gradient tracking techniques (Pu and Nedić, 2021) and achieve $\tilde{\mathcal{O}}\left(\epsilon^{-4}\right)$ sample complexity. There are also methods that integrate variance reduction techniques to achieve a faster convergence rate (Zhang et al., 2021b; Chen et al., 2022; Xian et al., 2021; Tarzanagh et al., 2022; Wu et al., 2023; Gao, 2022; Zhang et al., 2023b), but they require more memory for lager batch-size or full gradient. However, all the above-mentioned methods use a fixed or uniformly decaying stepsize requiring the prior knowledge of smoothness and concavity.

**(Distributed) adaptive minimax methods.** For centralized nonconvex minimax problems, Yang et al. (2022b) show that, even in deterministic settings, GDA-based methods necessitate the time-scale separation of the stepsizes for primal and dual updates. Many attempts have been made for ensuring the time-scale separation requirement (Lin et al., 2020; Yang et al., 2022c; Boţ and Böhm,

2023; Huang et al., 2023). However, these methods typically come with the prerequisite of having knowledge about problem-dependent parameters, which can be a significant drawback in practical scenarios. To this end, Yang et al. (2022b) introduce a nested adaptive algorithm named NeAda that incorporates an inner loop to effectively maximize the dual variable, yielding parameter-agnosticism and best-known sample complexity of $\tilde{\mathcal{O}}\left(\epsilon^{-4}\right)$. More recently, Li et al. (2023) introduce TiAda, a single-loop parameter-agnostic adaptive algorithm for nonconvex minimax optimization which employs separated exponential factors on the adaptive primal and dual stepsizes, improving upon NeAda on the noise-adaptivity and not requiring mini-batch. There has been few works dedicated to adaptive minimax optimization in federated learning settings. For instance, Huang (2022) introduce a federated adaptive algorithm that integrates the stepsize rule of Adam with full-clients participation, resembling the centralized counterpart. Ju et al. (2023) study a federated Adam algorithm for fair federated learning where the objective function is properly weighted to account for heterogeneous updates among nodes. To the best of our knowledge, it is still unknown how one can design an adaptive minimax method capable of fullfiling the time-scale separation requirement and being parameter-agnostic in *fully distributed settings*.

**Notations.** Throughout this paper, we denote by $\mathbb{E}\left[\cdot\right]$ the expectation of a stochastic variable, $\|\cdot\|$ the Frobenius norm, $\langle\cdot, \cdot\rangle$ the inner product of two vectors, $\odot$ the Schur product (entry wise), $\otimes$ the Kronecker product. We denote by $\mathbf{1}$ the all-ones vector, $\mathbf{I}$ the identity matrix and $\mathbf{J} = \mathbf{1}\mathbf{1}^T/n$ the averaging matrix with $n$ dimension. For a vector or matrix $A$ and constant $\alpha$, we denote $A^\alpha$ the entry-wise exponential operations. We denote $\Phi\left(x\right) := f\left(x, y^*\left(x\right)\right)$ as the primal function where $y^*\left(x\right) = \underset{y \in \mathcal{Y}}{\mathrm{argmax}} f\left(x, y\right)$, and $\mathcal{P}_\mathcal{Y}\left(\cdot\right)$ as the projection operation onto set $\mathcal{Y}$.

## 2 DISTRIBUTED ADAPTIVE MINIMAX METHODS

We consider the distributed minimax problem collaboratively solved by a set of agents over a communication network. The overall objective of the agents is to solve the following finite-sum problem:

$$\min_{x \in \mathbb{R}^p} \max_{y \in \mathcal{Y}} f\left(x, y\right) = \frac{1}{n} \sum_{i=1}^n \left(f_i(x, y) := \mathbb{E}_{\xi_i \sim \mathcal{D}_i}\left[F_i\left(x, y; \xi_i\right)\right]\right). \tag{1}$$

where $f_i : \mathbb{R}^{p+d} \to \mathbb{R}$ is the local private loss function accessible only by the associated node $i \in \mathcal{N} = \{1, 2, \cdots, n\}$, $\mathcal{Y} \subset \mathbb{R}^d$ is closed and convex, and $\xi_i \sim \mathcal{D}_i$ denotes the data sample locally stored at node $i \in \mathcal{N}$ with distribution $\mathcal{D}_i$. We consider a graph $\mathcal{G} = \left(\mathcal{V}, \mathcal{E}\right)$, here, $\mathcal{V} = \{1, 2, ..., n\}$ represents the set of agents, and $\mathcal{E} \subseteq \mathcal{V} \times \mathcal{V}$ denotes the set of edges consisting of ordered pairs $(i, j)$ representing the communication link from node $j$ to node $i$. For node $i$, we define $\mathcal{N}_i = \{j \mid (i, j) \in \mathcal{E}\}$ as the set of its neighboring nodes. Before proceeding to the discussion of distributed algorithms, we first introduce the following notations for brevity:

$$\mathbf{x}_k := [x_{1,k}, x_{2,k}, \cdots, x_{n,k}]^T \in \mathbb{R}^{n \times p}, \ \mathbf{y}_k := [y_{1,k}, y_{2,k}, \cdots, y_{n,k}]^T \in \mathbb{R}^{n \times d},$$

where $x_{i,k} \in \mathbb{R}^p, y_{i,k} \in \mathcal{Y}$ denote the primal and dual variable of node $i$ at each iteration $k$, and

$$\nabla_x F\left(\mathbf{x}_k, \mathbf{y}_k; \xi_k^x\right) := \left[\cdots, \nabla_x F_i\left(x_{i,k}, y_{i,k}; \xi_{i,k}^x\right), \cdots\right]^T \in \mathbb{R}^{n \times p},$$

$$\nabla_y F\left(\mathbf{x}_k, \mathbf{y}_k; \xi_k^y\right) := \left[\cdots, \nabla_y F_i\left(x_{i,k}, y_{i,k}; \xi_{i,k}^y\right), \cdots\right]^T \in \mathbb{R}^{n \times d},$$

are the corresponding partial stochastic gradients with i.i.d. samples $\xi_k^x, \xi_k^y$ in a compact form.

In what follows, we will first explain the pitfalls of directly applying centralized adaptive algorithms, and then introduce our newly proposed solution to address the challenge.

### 2.1 NON-CONVERGENCE OF NAIVE DISTRIBUTED ADAPTIVE METHODS

In centralized settings, designing parameter-agnostic adaptive methods for nonconvex-strongly-concave minimax problems is already challenging and demands careful considerations. In fact, simply employing adaptive methods such as AdaGrad and Adam can lead to convergence issues (Yang et al., 2022b). To the best of our knowledge, TiAda (Li et al., 2023) is the SOTA algorithm that achieves near-optimal rates with both parameter and noise adaptivity. Similar to extending SGD

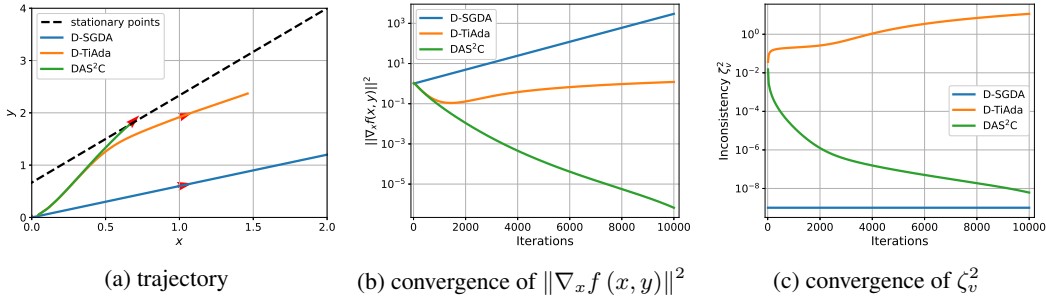

(a) trajectory · (b) convergence of $\left\|\nabla_x f\left(x, y\right)\right\|^2$ · (c) convergence of $\zeta_v^2$

Figure 1: Comparison among D-SGDA, D-TiAda and DAS²C for NC-SC quadratic objective function (5) with $n = 2$ nodes and $\gamma_x = \gamma_y$. In (a), it shows the trajectories of primal and dual variables of the algorithms, the points on the black dash line are stationary points of $f$. In (b), it shows the convergence of $\left\|\nabla_x f\left(x_k, y_k\right)\right\|^2$ over the iterations. In (c), it shows the convergence of the inconsistency of stepsizes, $\zeta_v^2$ defined in (7), over the iterations. Notably, $\zeta_v^2$ fails to converge for D-TiAda and $\zeta_v^2 = 0$ for non-adaptive D-SGDA.

into distributed settings (Nedic and Ozdaglar, 2009), TiAda can be adapted for distributed scenarios, which we will refer to as D-TiAda with the following update rules:

$$\mathbf{x}_{k+1} = W\left(\mathbf{x}_k - \gamma_x V_{k+1}^{-\alpha} \nabla_x F\left(\mathbf{x}_k, \mathbf{y}_k; \xi_k^x\right)\right), \tag{2a}$$

$$\mathbf{y}_{k+1} = \mathcal{P}_{\mathcal{Y}}\left(W\left(\mathbf{y}_k + \gamma_y U_{k+1}^{-\beta} \nabla_y F\left(\mathbf{x}_k, \mathbf{y}_k; \xi_k^y\right)\right)\right), \tag{2b}$$

where $\gamma_x$ and $\gamma_y$ are the stepsizes, $W$ is a doubly-stochastic weight matrix induced by graph $\mathcal{G}$, and

$$\begin{aligned} V_{k+1}^{-\alpha} &= \operatorname{diag}\left\{v_{i,k+1}^{-\alpha}\right\}_{i=1}^n, \quad v_{i,k+1} = \max\left\{m_{i,k+1}^x, m_{i,k+1}^y\right\}, \\ U_{k+1}^{-\beta} &= \operatorname{diag}\left\{u_{i,k+1}^{-\beta}\right\}_{i=1}^n, \quad u_{i,k+1} = m_{i,k+1}^y, \end{aligned} \tag{3}$$

where $m_{i,k+1}^x = m_{i,k}^x + \left\|\nabla_x F_i\left(x_{i,k}, y_{i,k}; \xi_{i,k}^x\right)\right\|^2$, $m_{i,k+1}^y = m_{i,k}^y + \left\|\nabla_y F_i\left(x_{i,k}, y_{i,k}; \xi_{i,k}^y\right)\right\|^2$ are the locally computed gradient norm information. TiAda employs a maximum operator in the preconditioner for $x$, specifically in the definition of $v_{i,k}$, as well as different stepsize decay rates, i.e., $0 < \beta < \alpha < 1$, for the two variables. Such design allows automatic balancing the stepsizes of $x$ and $y$ and achieves the desired time-scale separation without requiring any knowledge of parameters.

However, in the distributed setting, such naive extension may fail to converge to a stationary point because $v_{i,k}$ and $u_{i,k}$ can be inconsistent due to the difference of local objective functions $f_i$, In particular, we can rewrite the vanilla algorithm (2) above in the sense of average system of primal variable as below,

$$\bar{x}_{k+1} = \underbrace{\bar{x}_k - \gamma_x \bar{v}_k^{-\alpha} \frac{\mathbf{1}^T}{n} \nabla_x F\left(\mathbf{x}_k, \mathbf{y}_k; \xi_k^x\right)}_{\text{adaptive descent}} \underbrace{- \gamma_x \frac{\left(\tilde{\mathbf{v}}_{k+1}^{-\alpha}\right)^T}{n} \nabla_x F\left(\mathbf{x}_k, \mathbf{y}_k; \xi_k^x\right)}_{\text{inconsistancy}}, \tag{4}$$

where

$$\bar{x}_k := \frac{\mathbf{1}^T}{n} \mathbf{x}_k, \quad \bar{v}_k := \frac{1}{n}\sum_{i=1}^n v_{i,k}, \quad \left(\tilde{\mathbf{v}}_k^{-\alpha}\right)^T := \left[\cdots, v_{i,k}^{-\alpha} - \bar{v}_k^{-\alpha}, \cdots\right].$$

It is evident that, in comparison to centralized adaptive methods, an unexpected term on the right-hand side (RHS) arises due to inconsistencies, namely, $\tilde{\mathbf{v}}_k$. This term introduces inaccuracies in the directions of gradient descent, degrading the optimization performance. The following theorem provides an explicit lower bound consisting a constant steady-state-error regarding the non-convergence of D-TiAda, whose proof can be found in Appendix B.4.

**Theorem 1.** *There exists a distributed minimax problem in the form of Problem (1) and certain initialization such that after running D-TiAda with any $0 < \beta < 0.5 < \alpha$ and $\gamma_x, \gamma_y > 0$, it holds that for any $t = 0, 1, 2, \ldots$, we have*

$$\left\|\nabla_x f(x_t, y_t)\right\| = \left\|\nabla_x f(x_0, y_0)\right\| \quad \text{and} \quad \left\|\nabla_y f(x_t, y_t)\right\| = \left\|\nabla_y f(x_0, y_0)\right\|,$$

*where $\left\|\nabla_x f(x_0, y_0)\right\|$ and $\left\|\nabla_y f(x_0, y_0)\right\|$ can be arbitrarily large depending on the initialization.*

**Remark 1.** *The counterexample we constructed consists of three nodes, forming a complete graph. Without the stepsize control, TiAda will remain stationary, and the iterates will not progress if initiated along a specific line. In this counterexample, the only stationary point is at $(0, 0)$, but points along the line (c.f., Eq. (70)) can be positioned arbitrarily far away from this stationary point.*

Apart from the counterexample discussed in Theorem 1, where the iterates are stationary, we also experimentally observe the non-convergence for moving iterates of D-TiAda, D-SGDA, and D-AdaGrad (naively applying AdaGrad for each node) even in a simpler scenario involving only two connected agents. This is illustrated in Figure 1 and the functions are as depicted as follows.

$$f_1(x, y) = -\frac{9}{20}y^2 + \frac{3}{5}y - x + xy - \frac{1}{2}x^2, \ f_2(x, y) = -\frac{9}{20}y^2 + \frac{3}{5}y - x + 2xy - 2x^2. \quad (5)$$

It is not difficult to see that the points on the line $3y = 5x + 2$ are stationary points of $f(x, y) = 1/2(f_1(x, y) + f_2(x, y))$. It follows from Figure 1(a) and 1(b) that D-SGDA does not converge to a stationary point because of the lack of time-scale separation, and D-TiAda also fails to converge due to stepsizes inconsistency, as shown in Figure 1(c). In contrast, the utilization of the stepsize control protocol in DAS$^2$C ensures convergence to a stationary point, with the inconsistency in stepsizes gradually diminishing. These two motivating examples effectively highlight the challenges associated with applying minimax algorithms to distributed scenarios.

## 2.2 DAS$^2$C: A NEW ALGORITHM DESIGN WITH STEPSIZE CONTROL

To address the issue of inconsistent stepsizes across different nodes, we design the following distributed adaptive minimax optimization algorithm with stepsize control protocol, termed DAS$^2$C, which allows us to asymptotically track the centralized adaptive stepsize in a decentralized manner over networks. The pseudo-code for the algorithm is summarized in Algorithm 1, and can be rewritten in a compact form as follows

$$\mathbf{m}_{k+1}^x = W(\mathbf{m}_k^x + \mathbf{h}_k^x), \quad (6a)$$

$$\mathbf{m}_{k+1}^y = W(\mathbf{m}_k^y + \mathbf{h}_k^y), \quad (6b)$$

$$\mathbf{x}_{k+1} = W\left(\mathbf{x}_k - \gamma_x V_{k+1}^{-\alpha} \nabla_x F(\mathbf{x}_k, \mathbf{y}_k; \xi_k^x)\right), \quad (6c)$$

$$\mathbf{y}_{k+1} = \mathcal{P}_{\mathcal{Y}}\left(W\left(\mathbf{y}_k + \gamma_y U_{k+1}^{-\beta} \nabla_y F(\mathbf{x}_k, \mathbf{y}_k; \xi_k^y)\right)\right), \quad (6d)$$

where $\mathbf{m}_k^x$ and $\mathbf{m}_k^y$ denote the accumulation of historical gradients with

$$\boldsymbol{h}_k^x = \left[\cdots, \|g_{i,k}^x\|^2, \cdots\right]^T \in \mathbb{R}^n, \ \boldsymbol{h}_k^y = \left[\cdots, \|g_{i,k}^y\|^2, \cdots\right]^T \in \mathbb{R}^n,$$

and $V_k$ and $U_k$ are diagonal matrices with $v_{i,k} = \max\left\{m_{i,k}^x, m_{i,k}^y\right\}$, $u_{i,k} = m_{i,k}^x$, where the maximization operator is used to achieve time-scale separation as suggested in TiAda (Li et al., 2023). Note that we also provide a variant of DAS$^2$C with coordinate-wise adaptive stepsizes in Algorithm 2, along with its convergence analysis in Appendix B.6.

## 3 CONVERGENCE ANALYSIS

In this section, we introduce the main convergence results for the proposed DAS$^2$C algorithm and compare it with D-TiAda to show the effectiveness of the proposed stepsize control protocol. To this end, we define the following metrics to evaluate the level of inconsistency of stepsizes among nodes, which are ensured to be bounded with Assumption 4.

$$\zeta_v^2 := \sup_{k>0}\left\{\left(v_{i,k}^{-\alpha} - \bar{v}_k^{-\alpha}\right)^2 / \left(\bar{v}_k^{-\alpha}\right)^2\right\}, \ \zeta_u^2 := \sup_{k>0}\left\{\left(u_{i,k}^{-\beta} - \bar{u}_k^{-\beta}\right)^2 / \left(\bar{u}_k^{-\beta}\right)^2\right\}, \ i \in [n], \quad (7)$$

where $\bar{u}_k := 1/n \sum_{i=1}^n u_{i,k}$.

### 3.1 ASSUMPTIONS

We consider the NC-SC setting of Problem (1) with the following assumptions that are commonly used in the existing works (c.f., Remark 2).

---

**Algorithm 1 Distributed Adaptive Time-Scale Separated Stepsize Control Method (DAS$^2$C)**

---

**Initialization:** $x_{i,0} \in \mathbb{R}^p$, $y_{i,0} \in \mathcal{Y}$, buffers $m_{i,0}^x = m_{i,0}^y = c > 0$, stepsizes $\gamma_x, \gamma_y > 0$, exponential factors $0 < \beta < \alpha < 1$ and weight matrix $W$.

1: **for** iteration $k = 0, 1, \cdots$, each node $i \in [n]$, **do**

2:     Sample i.i.d. $g_{i,k}^x = \nabla_x F_i\left(x_{i,k}, y_{i,k}; \xi_{i,k}^x\right)$, $g_{i,k}^y = \nabla_y F_i\left(x_{i,k}, y_{i,k}; \xi_{i,k}^y\right)$.

3:     Accumulate gradient norm: $m_{i,k+1}^x = m_{i,k}^x + \|g_{i,k}^x\|^2$, $m_{i,k+1}^y = m_{i,k}^y + \|g_{i,k}^y\|^2$.

4:     Compute the ratio: $\psi_{i,k+1} = (m_{i,k+1}^x)^\alpha / \max\left\{(m_{i,k+1}^x)^\alpha, (m_{i,k+1}^y)^\alpha\right\} \leqslant 1$.

5:     Update primal and dual variables locally:

$$x_{i,k+1} = x_{i,k} - \gamma_x \psi_{i,k+1} \left(m_{i,k+1}^x\right)^{-\alpha} g_{i,k}^x, \ y_{i,k+1} = y_{i,k} - \gamma_y (m_{i,k+1}^y)^{-\beta} g_{i,k}^y.$$

6:     Communicate adaptive stepsizes and decision variables with neighbors:

$$\left\{m_{i,k+1}^x, m_{i,k+1}^y, x_{i,k+1}, y_{i,k+1}, \right\} \leftarrow \sum_{j \in \mathcal{N}_i} W_{i,j} \left\{m_{j,k+1}^x, m_{j,k+1}^y, x_{j,k+1}, y_{j,k+1}, \right\}.$$

7:     Projection of dual variable on to set $\mathcal{Y}$: $y_{i,k+1} \leftarrow \mathcal{P}_\mathcal{Y}(y_{i,k+1})$.

8: **end for**

---

**Assumption 1** ($\mu$-strong concavity in $y$). *Each objective function $f_i(x, y)$ is $\mu$-strongly concave in $y$, i.e., $\forall x \in \mathbb{R}^p$, $\forall y, y' \in \mathcal{Y}$ and $\mu > 0$,*

$$f_i(x, y) - f_i(x, y') \geqslant \langle \nabla_y f_i(x, y), y - y' \rangle + \frac{\mu}{2} \| y - y' \|^2. \tag{8}$$

**Assumption 2** (Joint smoothness). *Each objective function $f_i(x, y)$ is L-smooth $\forall x \in \mathbb{R}^p, y \in \mathcal{Y}$, i.e., $\forall x, x' \in \mathbb{R}^p$ and $\forall y, y' \in \mathcal{Y}$, there exists a constant $L$ such that*

$$\|\nabla_z f_i(x, y) - \nabla_z f_i(x', y')\|^2 \leqslant L^2 \left(\|x - x'\|^2 + \|y - y'\|^2\right), \text{ for } z \in \{x, y\}. \tag{9}$$

*Furthermore, $f_i$ is second-order Lipschitz continuous for $y$, i.e.,*

$$\left\|\nabla_{zy}^2 f_i(x, y) - \nabla_{zy}^2 f_i(x', y')\right\|^2 \leqslant L^2 \left(\|x - x'\|^2 + \|y - y'\|^2\right), \text{ for } z \in \{x, y\}. \tag{10}$$

**Assumption 3** (Interior optimal point). *For all $x \in \mathbb{R}^p$, $y^*(x)$ is in the interior of $\mathcal{Y}$.*

**Assumption 4** (Stochastic gradient). *For i.i.d. sample $\xi_i$, the stochastic gradient of each $i$ is unbiased, i.e., $\forall x \in \mathbb{R}^p, y \in \mathcal{Y}$, $\mathbb{E}_{\xi_i}[\nabla_z F_i(x, y; \xi_i)] = \nabla_z f_i(x, y)$, for $z \in \{x, y\}$, and there exists a constant $C > 0$ such that $\|\nabla_z F_i(x, y; \xi_i)\| \leqslant C$.*

**Remark 2.** *Assumption 1 does not require the convexity in $x$ and the objective function thus can be nonconvex. Assumption 2 and 3 ensure that $y^*(\cdot)$ is smooth (c.f., Lemma 3), which is essential for achieving (near) optimal convergence rate (Chen et al., 2021; Li et al., 2023). Assumption 3 also ensures that $\nabla_y f(x, y^*(x)) = 0$. This is important for AdaGrad-based methods to maintain stepsizes near $y^*(x)$ without being excessively small which otherwise lead to slow convergence. Assumption 4 on bounded stochastic gradient is widely used for establishing convergence rates of adaptive methods (Zou et al., 2019; Kavis et al., 2022; Chen et al., 2023a), and it can be satisfied in many tasks such as neural networks with rectified activation (Dinh et al., 2017) and GANs with projections on the critic (Gulrajani et al., 2017). To the best of our knowledge, for stochastic NC-SC minimax optimization, no existing parameter-agnostic method achieves a near-optimal convergence rate while also eliminating the bounded gradient assumption. In fact, this challenge persists even in stochastic minimization with strongly-convex objectives (Orvieto et al., 2022).*

Now, we make the following assumption to ensure the connectivity of the graph. Note that the weight matrix is not required to be symmetric thus the graph can be direct, e.g., direct ring and exponential graphs (Ying et al., 2021), which is more general than (Lian et al., 2017; Borodich et al., 2021).

**Assumption 5** (Graph connectivity). *The weight matrix $W$ induced by graph $\mathcal{G}$ is doubly stochastic, i.e., $W\mathbf{1} = \mathbf{1}, \mathbf{1}^T W = \mathbf{1}^T$ and $\rho_W := \|W - \mathbf{J}\|_2^2 < 1$.*

### 3.2 MAIN RESULTS

We begin by demonstrating in the following lemma that the inconsistency terms, as described in (4), exhibit asymptotic convergence in the case of the proposed DAS$^2$C algorithm. In contrast, these terms remain non-vanishing for D-TiAda (c.f., Lemma 11).

**Lemma 1** (Convergence of inconsistency terms). *Suppose Assumption 1-5 hold. For the proposed DAS$^2$C in Algorithm 1, we have*

$$\frac{1}{K} \sum_{k=0}^{K-1} \mathbb{E} \left[ \left\| \frac{\left( \tilde{\boldsymbol{v}}_{k+1}^{-\alpha} \right)^T}{n \bar{v}_{k+1}^{-\alpha}} \nabla_x F \left( \mathbf{x}_k, \mathbf{y}_k; \xi_k^x \right) \right\|^2 \right] \leqslant \sqrt{ \frac{1}{n^{1-\alpha}} \left( \frac{4 \rho_W}{\left( 1 - \rho_W \right)^2} \right)^\alpha \frac{\left( 1 + \zeta_v \right) \zeta_v C^{2-\alpha}}{\left( 1 - \alpha \right) K^\alpha}},$$

$$\frac{1}{K} \sum_{k=0}^{K-1} \mathbb{E} \left[ \left\| \frac{\left( \tilde{\boldsymbol{u}}_{k+1}^{-\alpha} \right)^T}{n \bar{u}_{k+1}^{-\alpha}} \nabla_y F \left( \mathbf{x}_k, \mathbf{y}_k; \xi_k^y \right) \right\|^2 \right] \leqslant \sqrt{ \frac{1}{n^{1-\beta}} \left( \frac{4 \rho_W}{\left( 1 - \rho_W \right)^2} \right)^\beta \frac{\left( 1 + \zeta_u \right) \zeta_u C^{2-\beta}}{\left( 1 - \beta \right) K^\beta}}.$$

$$(11)$$

*where* $\left( \tilde{\boldsymbol{u}}_k^{-\beta} \right)^T := \left[ \cdots, u_{i,k}^{-\beta} - \bar{u}_k^{-\beta}, \cdots \right]$. *The proof of Lemma 1 can be found Appendix B.3.*

We are now ready to present the key convergence results in terms of the primal function $\Phi(x) := f(x, y^*(x))$ with $y^*(x) = \underset{y \in \mathcal{Y}}{\mathrm{argmax}} f(x, y)$, whose proofs can be found in Appendix B.5.

**Theorem 2.** *Suppose Assumption 1-5 hold. Let* $0 < \alpha < \beta < 1$ *and the total iteration satisfy*

$$K = \Omega \left( \max \left\{ 1, \quad \left( \gamma_x^2 \kappa^4 / \gamma_y^2 \right)^{1/(\alpha-\beta)}, \quad \left( 1 / \left( 1 - \rho_W \right)^2 \right)^{\max\{1/\alpha, \, 1/\beta\}} \right\} \right), \tag{12}$$

*where* $\kappa := L/\mu$, *to ensure time-scale separation and quasi-independence of network. For DAS$^2$C,*

$$\frac{1}{K} \sum_{k=0}^{K-1} \mathbb{E} \left[ \left\| \nabla \Phi \left( \bar{x}_k \right) \right\|^2 \right] = \tilde{\mathcal{O}} \left( \frac{1}{K^{1-\alpha}} + \frac{1}{\left( 1 - \rho_W \right)^\alpha K^\alpha} + \frac{1}{K^{1-\beta}} + \frac{1}{\left( 1 - \rho_W \right) K^\beta} \right). \tag{13}$$

**Remark 3** (Near-optimal convergence). *Theorem 2 implies that if the total number of iterations satisfies the conditions (12), the proposed DAS$^2$C algorithm converges to a stationary point exactly for Problem 1 with an* $\tilde{\mathcal{O}} \left( \epsilon^{-(4+\delta)} \right)$ *sample complexity for any small* $\delta > 0$ *with* $\alpha = 0.5 + \delta/(8+2\delta)$ *and* $\beta = 0.5 - \delta/(8+2\delta)$. *It is worth to note that this rate is near-optimal comparing to the best-known result* $\tilde{\mathcal{O}} \left( \epsilon^{-4} \right)$ *(Li et al., 2021; Yang et al., 2022b) for centralized minimax problems, and recovers the centralized TiAda algorithm (Li et al., 2023) as special case, i.e., letting* $\rho_W = 0$.

**Remark 4** (Parameter-agnostic property and transit times). *The above results show that DAS$^2$C is parameter-agnostic without requiring to know any problem-dependent parameters. Furthermore, we characterize the specific transient times (c.f., (12)) that ensure time-scale separation and quasi-independence of network in the sense of* $\alpha, \beta < 1$, *respectively. Indeed, we can see that if* $\alpha$ *and* $\beta$ *are close to each other, the time required for time-scale separation to occur increases significantly, which has been observed in TiAda. If* $\alpha$ *or* $\beta$ *approaches* 0, *the transition time for achieving quasi-independence of the network will also increase. These observations highlight the importance of trade-offs between the convergence rate and the required duration of the transition phase.*

For comparison, we also derive an upper bound for D-TiAda as follows. Together with the lower bound in Theorem 1, we demonstrate that without the stepsize control, the inconsistencies between local stepsizes prevent D-TiAda to converge in the distributed setting.

**Corollary 1.** *Under the same conditions of Theorem 2. For D-TiAda algorithm, we have*

$$\frac{1}{K} \sum_{k=0}^{K-1} \mathbb{E} \left[ \left\| \nabla \Phi \left( \bar{x}_k \right) \right\|^2 \right]$$

$$= \tilde{\mathcal{O}} \left( \frac{1}{K^{1-\alpha}} + \frac{1}{\left( 1 - \rho_W \right)^\alpha K^\alpha} + \frac{1}{K^{1-\beta}} + \frac{1}{\left( 1 - \rho_W \right) K^\beta} \right) + \tilde{\mathcal{O}} \left( \left( \zeta_v^2 + \kappa^2 \zeta_u^2 \right) C^2 \right). \tag{14}$$

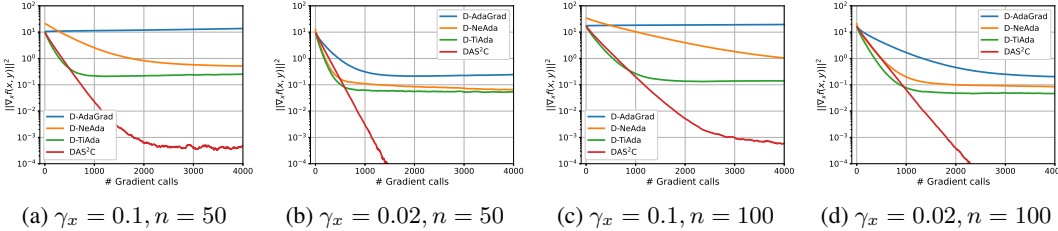

| (a) $\gamma_x = 0.1, n = 50$ | (b) $\gamma_x = 0.02, n = 50$ | (c) $\gamma_x = 0.1, n = 100$ | (d) $\gamma_x = 0.02, n = 100$ |

Figure 2: Performance comparison of algorithms on quadratic functions over exponential graphs with node counts $n = \{50, 100\}$ and *different initial stepsizes* ($\gamma_y = 0.1$).

## 4    EXPERIMENTS

In this section, we conduct experiments to validate the theoretical findings and demonstrate the effectiveness of the proposed algorithm on real-world machine learning tasks. We compare the proposed DAS$^2$C with the distributed variants of AdaGrad (Duchi et al., 2011), TiAda (Li et al., 2023) and NeAda (Yang et al., 2022b), namely D-AdaGrad, D-TiAda and D-NeAda, respectively. These experiments run across multiple nodes with different communication topologies, and we consider heterogeneous distributions of local objective functions/datasets. For example, each node can only access samples with a subset of labels on MNIST and CIFAR-10 datasets, which is a common scenario in decentralized and federated learning tasks (Sharma et al., 2023; Huang et al., 2022). The experiments cover three main tasks: synthetic function, robust training of the neural network, and training of Wasserstein GANs (Heusel et al., 2017). More experimental details and additional experiments under other settings can be found in Appendix A.

**Synthetic example.**  We consider a distributed minimax problem with the following NC-SC local objective functions over exponential networks with $n = 50$ ($\rho_w = 0.71$) and $n = 100$ ($\rho_w = 0.75$).

$$f_i(x, y) = -\frac{1}{2}y^2 + L_i xy - \frac{L_i^2}{2}x^2 - 2L_i x + L_i y, \tag{15}$$

where $L_i \sim \mathcal{U}(1.5, 2.5)$. The local gradient of each node is computed with an additive $\mathcal{N}(0, 0.1)$ Gaussian noise. For both D-TiAda and DAS$^2$C, we set the parameters as follows: $\alpha = 0.6$ and $\beta = 0.4$. It follows from Figure 2 (a) and 2 (b) that the proposed DAS$^2$C algorithm outperforms other distributed adaptive methods for both initial stepsize settings, especially in cases with a favorable initial stepsize ratio, as illustrated in plots (b) and (d) where $\gamma_x/\gamma_y = 0.2$. Similar observation can be found in Figure 2 (c) and 2 (d), demonstrating the effectiveness of DAS$^2$C.

**Robust training of neural networks.**  Next, we consider the task of robust training of neural networks, in the presence of adversarial perturbations on data samples (Sharma et al., 2022; Deng and Mahdavi, 2021). The problem can be formulated as $\min_x \max_y 1/n \sum_{i=1}^n f_i(x; \xi_i + y) - \eta \|y\|^2$, where $x$ denotes the parameters of the model, $y$ denotes the perturbation and $\xi_i$ denotes the data sample of node $i$. If $\eta$ is large enough, the problem is NC-SC. We conduct experiments on MNIST dataset over different networks, e.g., ring graph, exponential (exp.) graph (Ying et al., 2021) and dense graph with $n/2$ edges for each node. We consider a heterogeneous scenario in which each node possesses only two distinct classes of labeled samples, resulting in heterogeneity among the local datasets across nodes, while the data is i.i.d within each node.

In Figure 3, we compare DAS$^2$C with D-AdaGrad, D-TiAda and D-NeAda, using adaptive stepsizes in AdaGrad (first row) and Adam (second row, name suffixed with Adam) respectively, it can be observed from the first three columns that the proposed DAS$^2$C outperforms the others on three different graphs and it is not very sensitive to the graph connectivity (i.e., $\rho_W$), demonstrating the quasi-independence of network as indicated in Theorem 2. It should be noted that Adam-like algorithms fluctuate more in the later stages of optimization as the gradient norm vanishes, leading to the inevitable increase of the Adam stepsize as the optimization process approaches convergence (Kingma and Ba, 2014). In plots (d) and (h), we further demonstrate the efficient scalability of DAS$^2$C with respect to the number of nodes, while keeping a constant batch size of 64 for each node. This showcases the algorithm's ability to handle larger-scale distributed scenarios effectively.

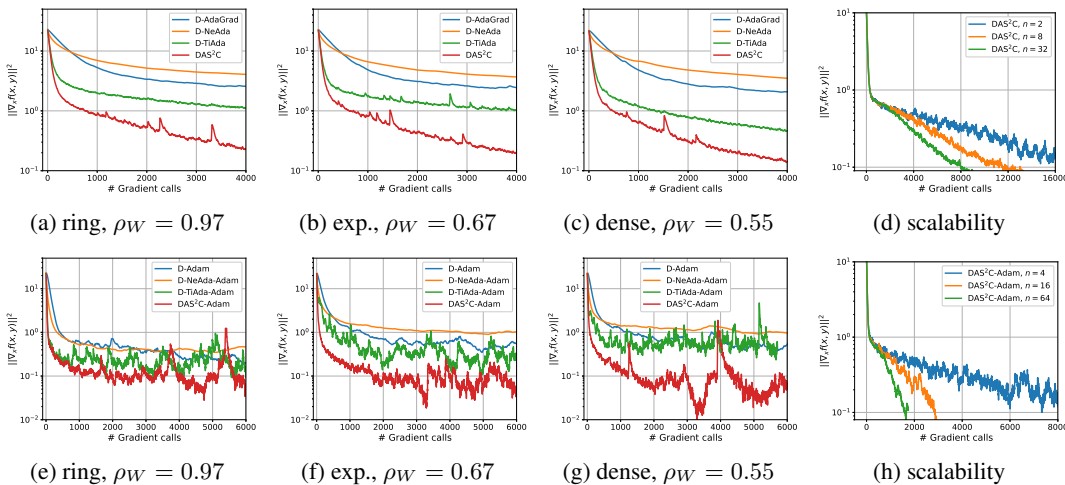

(a) ring, $\rho_W = 0.97$     (b) exp., $\rho_W = 0.67$     (c) dense, $\rho_W = 0.55$     (d) scalability

(e) ring, $\rho_W = 0.97$     (f) exp., $\rho_W = 0.67$     (g) dense, $\rho_W = 0.55$     (h) scalability

Figure 3: Comparison of the algorithms on training robust CNN on MNIST dataset. The first shows the results of AdaGrad-like stepsize, and the second row is for Adam-like stepsize. For the first three columns, we compare the algorithms on *different graphs* with $n = 20$. For the last column we show the scalability of DAS$^2$C in terms of number of nodes. Initial stepsizes are set as $\gamma_x = 0.01, \gamma_y = 0.1$ for AdaGrad-like stepsize, and $\gamma_x = 0.1, \gamma_y = 0.1$ for Adam-like stepsize.

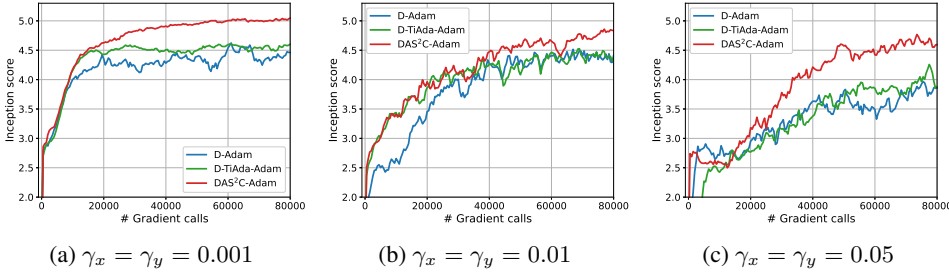

(a) $\gamma_x = \gamma_y = 0.001$     (b) $\gamma_x = \gamma_y = 0.01$     (c) $\gamma_x = \gamma_y = 0.05$

Figure 4: Training GANs on CIFAR-10 dataset over exponential graphs with $n = 10$ nodes using *different initial stepsizes*.

**Generative Adversarial Networks.** We further illustrate the effectiveness of DAS$^2$C on another popular task of training GANs, which has a generator and a discriminator used to generate and distinguish samples respectively (Goodfellow et al., 2014). In this experiment, we train Wasserstein GANs (Gulrajani et al., 2017) on CIFAR-10 dataset in decentralized setting where each discriminator is 1-Lipschitz and has access to only two classes of samples. We compare the inception score of DAS$^2$C with D-Adam and D-TiAda adopting Adam-like stepsizes in Figure 4. It can be observed from the figure that DAS$^2$C achieves higher inception scores in three cases with different initial stepsizes, and has a small score loss as the initial step size changes. We believe that this example shows the great potential of DAS$^2$C in solving real-world problems.

## 5   CONCLUSION

We introduced a new distributed adaptive minimax method, DAS$^2$C, designed to tackle the issue of non-convergence in nonconvex-strongly-concave minimax problems caused by locally computed adaptive stepsize inconsistencies. Vanilla distributed adaptive methods could suffer from such inconsistencies, as highlighted by the carefully designed counterexamples for demonstrating their potential non-convergence. In contrast, our proposed method employs an efficient adaptive stepsize control protocol that guarantees stepsize consistency among nodes, effectively eliminating steady-state errors. Theoretically, we showed that DAS$^2$C can achieve a near-optimal convergence rate of $\tilde{\mathcal{O}}\left(\epsilon^{-(4+\delta)}\right)$ with any small $\delta > 0$. Extensive experiments on real-world datasets have been conducted to validate our theoretical findings across various scenarios.

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

# A ADDITIONAL EXPERIMENTS

In this section, we provide detailed experimental settings and perform additional experiments on the task of training robust neural networks with different choices of hyper-parameters. All experiments are deployed in a server with Intel Xeon E5-2680 v4 CPU @ 2.40GHz and 8 Nvidia RTX 3090 GPUs, and implemented using distributed communication package *torch.distributed* in PyTorch 2.0, where each process serves as a node, and we use inter-process communication to mimic communication between nodes. We adapt code from (Yang et al., 2022b; Li et al., 2023) to decentralized settings. We use $\alpha = 0.6$ and $\beta = 0.4$ for all tasks.

## A.1 EXPERIMENTAL DETAILS

**Communication topology.** For the experiments in the main-text, we utilize three commonly used communication topologies: indirect ring, exponential graph and dense graph. Indirect ring is a sparse graph in which each node is sequentially connected to form a ring, with only two neighbors per node. Exponential graph (Ying et al., 2021) is a directed graph where each node is connected to nodes at distances of $2^0, 2^1...2^{log(n)}$. Exponential graphs achieve a good balance between the degree and connectivity of the graph. Dense graph is a indirect graph where each node is connected to nodes at distances of $1, 2, 4, ..., n$. We also consider directed ring and fully connected graphs, which are more sparsely and densely connected, respectively, in the additional experiments.

**Robust training of neural network.** In this task, we train CNNs with three convolutional layers and one fully connected layer on MNIST dataset containing 10 class images. Each layer adopts batch normalization and ELU activation. The total batch size is 1280, and the batch size of each node during training is $1280/n$. For Adam-like algorithms, we set the first and second moment parameters as $\beta_1 = 0.9, \beta_2 = 0.999$ respectively. Since NeAda is a double-loop algorithm, for fair comparison, we implement D-AdaGrad and D-Adam using 15 iterations of inner loop in this task.

**Generative Adversarial Networks.** In this task, we train Wasserstein GANs on CIFAR-10 dataset, where the model we use for discriminator is a four layer CNN, and for generator is a four layer CNN with transpose convolution layers. The total batch size is 1280, and the batch size of each node during training is 128 with 10 nodes. For Adam-like algorithms, we use $\beta_1 = 0.5, \beta_2 = 0.9$. To obtain the inception score, we use 8000 artificially generated samples to feed the previously trained inception network.

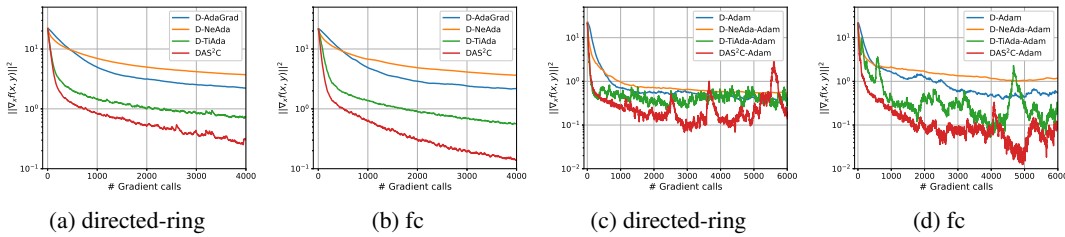

Figure 5: Train CNN on MNIST with $n = 20$ nodes over *directed ring and fully connected graphs*.

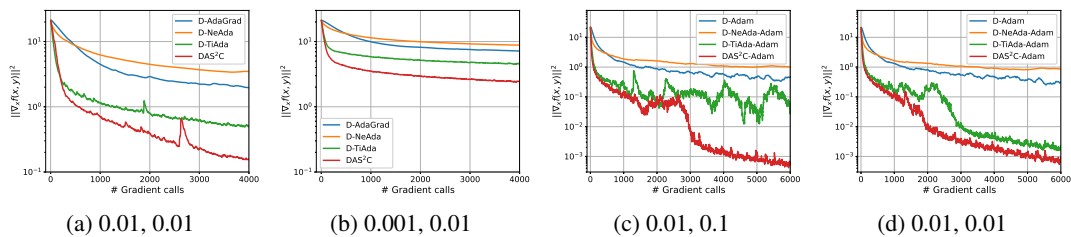

Figure 6: Train CNN on MNIST with $n = 20$ nodes with *different initial stepsizes* $\gamma_x$ and $\gamma_y$.

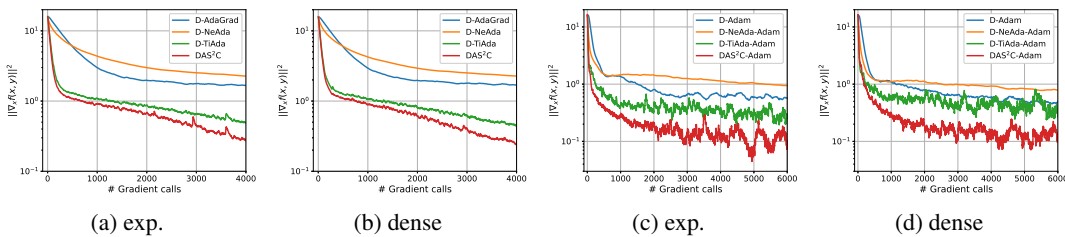

Figure 7: Train CNN on MNIST with $n = 20$ nodes over exponential and dense graphs where each node has *4 sample classes*.

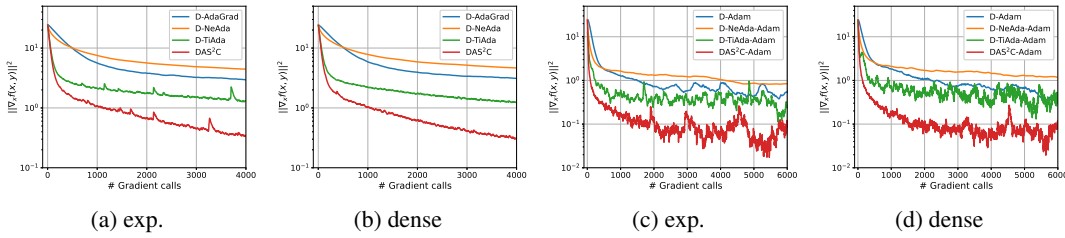

Figure 8: Train CNN on MNIST with $n = 40$ nodes over exponential and dense graphs.

### A.2 ADDITIONAL EXPERIMENTS ON ROBUST TRAINING OF NEURAL NETWORK.

In this part, we conduct additional experiments on robust training of CNNs on MNIST dataset considering a variety of settings. We compare the convergence performance of DAS$^2$C with D-AdaGrad, D-TiAda and D-NeAda using adaptive stepsizes in AdaGrad and Adam. Unless otherwise specified, the total batch-size is set to 1280; the initial stepsizes for $x$ and $y$ are assigned as $\gamma_x = 0.01, \gamma_y = 0.1$ for AdaGrad-like algorithms, and $\gamma_x = \gamma_y = 0.1$ for Adam-like algorithms. Specifically, we consider two extra graphs that are more sparse and more dense, respectively in Figure 5, e.g., directed ring and fully-connected (fc) graphs. We use more initial stepsizes settings for $x$ and $y$ respectively in Figure 6. Further, we consider another data distribution where each node has data from 4 of the 10 classes in Figure 7. Finally we perform a comparison experiment with 40 nodes. Under all settings, the proposed DAS$^2$C outperforms the others, demonstrating the superiority of DAS$^2$C.

## B    PROOF OF THE MAIN RESULTS

We recall here some notations used in the main text. The averaged variables and the inconsistency are defined as follows:

$$\bar{x}_k := \frac{\mathbf{1}^T}{n}\mathbf{x}_k, \quad \bar{v}_k := \frac{1}{n}\sum_{i=1}^{n} v_{i,k}, \quad \left(\tilde{\boldsymbol{v}}_k^{-\alpha}\right)^T := \left[\cdots, v_{i,k}^{-\alpha} - \bar{v}_k^{-\alpha}, \cdots\right],$$

$$\bar{y}_k := \frac{\mathbf{1}^T}{n}\mathbf{y}_k, \quad \bar{u}_k := \frac{1}{n}\sum_{i=1}^{n} u_{i,k}, \quad \left(\tilde{\boldsymbol{u}}_k^{-\beta}\right)^T := \left[\cdots, u_{i,k}^{-\beta} - \bar{u}_k^{-\beta}, \cdots\right].$$

The heterogeneity of stepsizes is defined as:

$$\zeta_v^2 := \sup_{k>0}\left\{\left(v_{i,k}^{-\alpha} - \bar{v}_k^{-\alpha}\right)^2 / \left(\bar{v}_k^{-\alpha}\right)^2\right\}, \quad \zeta_u^2 := \sup_{k>0}\left\{\left(u_{i,k}^{-\beta} - \bar{u}_k^{-\beta}\right)^2 / \left(\bar{u}_k^{-\beta}\right)^2\right\}, \quad i \in [n].$$

**Proof Sketch.** The convergence analysis of the main results in Theorem 2 mainly relays on the analysis of the average system as shown in (4), and the difference between the distributed system and the average system. In general, under the Assumption 1-5, we first give a telescoped descent lemma from $0$ to $K-1$ iterations in Lemma 5, which is upper bounded by several key error terms:

- $S_1 := \frac{1}{nK}\sum_{k=0}^{K-1}\mathbb{E}\left[\bar{v}_{k+1}^{-\alpha}\left\|\nabla_x F\left(\mathbf{x}_k, \mathbf{y}_k; \xi_k^x\right)\right\|^2\right]$: The asymptotically decaying terms by adopting adaptive stepsize;

- $S_2 := \frac{1}{nK}\sum_{k=0}^{K-1}\mathbb{E}\left[\left\|\mathbf{x}_k - \mathbf{1}\bar{x}_k\right\|^2 + \left\|\mathbf{y}_k - \mathbf{1}\bar{y}_k\right\|^2\right]$: The consensus error of $x$ and $y$ between the distributed system and the average system;

- $S_3 := \frac{1}{K}\sum_{k=0}^{K-1}\mathbb{E}\left[f\left(\bar{x}_k, y^*\left(\bar{x}_k\right)\right) - f\left(\bar{x}_k, \bar{y}_k\right)\right]$: the optimality gap in dual variable $y$;

- $S_4 := \frac{1}{K}\sum_{k=0}^{K-1}\mathbb{E}\left[\left\|\frac{\left(\tilde{\boldsymbol{v}}_{k+1}^{-\alpha}\right)^T}{n\bar{v}_{k+1}^{-\alpha}}\nabla_x F\left(\mathbf{x}_k, \mathbf{y}_k; \xi_k^x\right)\right\|^2\right]$: The inconsistency of stepsize of $x$;

Next, we prove that these terms are convergent in Lemma 6-10 and Lemma 1 respectively. Finally, these results are integrated into the descent lemma thus completing the proof. We note that the proof is not trivial in the sense that these terms are coupled and therefore need to be carefully analyzed. This proof can also be adapted to analyze the coordinate-wise adaptive stepsize variant of DAS$^2$C as explained in Appendix B.6, which is of independent interest.

### B.1    SUPPORTING LEMMAS

In this section, we provide several supporting lemmas that have been shown in the existing literature, which are essential to subsequent convergence analysis.

**Lemma 2** (Lemma A.2 in Yang et al. (2022b)). *Let $\{x_t\}_{t=0}^{T-1}$ be a sequence of non-negative real numbers, $x_0 > 0$ and $\alpha \in (0,1)$. Then we have,*

$$\left(\sum_{t=0}^{T-1} x_t\right)^{1-\alpha} \leqslant \sum_{t=0}^{T-1}\frac{x_t}{\left(\sum_{k=0}^{t} x_k\right)^\alpha} \leqslant \frac{1}{1-\alpha}\left(\sum_{t=0}^{T-1} x_t\right)^{1-\alpha}. \tag{16}$$

*When $\alpha = 0$, we have*

$$\sum_{t=0}^{T-1}\frac{x_t}{\left(\sum_{k=0}^{t} x_k\right)^\alpha} \leqslant 1 + \log\left(\frac{\sum_{t=0}^{T-1} x_t}{x_0}\right). \tag{17}$$

**Lemma 3.** *Under Assumption 1, 2 and 3. Define $\Phi(x) := f(x, y^*(x))$ as the envelope function and $y^*(x) = \underset{y \in \mathcal{Y}}{\operatorname{argmax}} f(x, y)$. Then, we have,*

1) *$\Phi(\cdot)$ is $L_\Phi$-smooth with $L_\Phi = L(1 + \kappa)$, and $\nabla\Phi(x) = \nabla_x f(x, y^*(x))$ (Lemma 4.3 in Lin et al. (2020));*

2) $y^*(\cdot)$ is $\kappa$-Lipschitz and $\hat{L}$-smooth with $\hat{L} = \kappa(1 + \kappa)^2$(Lemma 2 in *Chen et al. (2021)*).

**Lemma 4.** *Let $A, B \in \mathbb{R}^{n \times p}$ be matrices with the same dimension. By the definitions of Frobenius norm and Schur product, we have*

1) $\|A \odot B\|^2 \leqslant \|A\|^2 \|B\|^2$;

2) $\left\| \frac{\mathbf{1}^T}{n} A \odot B \right\|^2 \leqslant \frac{1}{n} \|A\|^2 \|B\|^2$;

3) *For a vector $\boldsymbol{a} \in \mathbb{R}^n$, $\left\| \boldsymbol{a}\mathbf{1}_p^T \odot B \right\|^2 = \|\mathrm{diag}(\boldsymbol{a}) B\|^2 \leqslant \|\boldsymbol{a}\|^2 \|B\|^2$.*

### B.2 KEY LEMMAS

In this subsection, we give the key lemmas to help the analysis of the main results. For simplicity, we define $\Delta_k := \|\mathbf{x}_k - \mathbf{1}\bar{x}_k\|^2 + \|\mathbf{y}_k - \mathbf{1}\bar{y}_k\|^2$ as the consensus error for primal and dual variables. Then, we have the following lemmas.

**Lemma 5** (Descent lemma). *Suppose Assumption 1-5 hold. we have*

$$\frac{1}{K} \sum_{k=0}^{K-1} \mathbb{E}\left[\|\nabla\Phi(\bar{x}_k)\|^2\right]$$

$$\leqslant \frac{8C^{2\alpha}(\Phi^{\max} - \Phi^*)}{\gamma_x K^{1-\alpha}} - \frac{4}{K} \sum_{k=0}^{K-1} \mathbb{E}\left[\|\nabla_x f(\bar{x}_k, \bar{y}_k)\|^2\right]$$

$$+ 8\gamma_x L_\Phi\left(1 + \zeta_v^2\right) \underbrace{\frac{1}{nK} \sum_{k=0}^{K-1} \mathbb{E}\left[\bar{v}_{k+1}^{-\alpha} \|\nabla_x F(\mathbf{x}_k, \mathbf{y}_k; \xi_k^x)\|^2\right]}_{S_1} + 8L^2 \underbrace{\frac{1}{nK} \sum_{k=0}^{K-1} \mathbb{E}[\Delta_k]}_{S_2}$$

$$+ 8\kappa L \underbrace{\frac{1}{K} \sum_{k=0}^{K-1} \mathbb{E}[f(\bar{x}_k, y^*(\bar{x}_k)) - f(\bar{x}_k, \bar{y}_k)]}_{S_3} + 16 \underbrace{\frac{1}{K} \sum_{k=0}^{K-1} \mathbb{E}\left[\left\|\frac{(\tilde{\boldsymbol{v}}_{k+1}^{-\alpha})^T}{n\bar{v}_{k+1}^{-\alpha}} \nabla_x F(\mathbf{x}_k, \mathbf{y}_k; \xi_k^x)\right\|^2\right]}_{S_4}.$$

$$(18)$$

*where $\kappa := L/\mu$ is the condition number of the function in $y$, $\Phi^{\max} = \max_x \Phi(x), \Phi^* = \min_x \Phi(x)$.*

*Proof.* By the smoothness of $\Phi$ given in Lemma 3, i.e.,

$$\Phi(\bar{x}_{k+1}) - \Phi(\bar{x}_k) \leqslant \langle\nabla\Phi(\bar{x}_k), \bar{x}_{k+1} - \bar{x}_k\rangle + \frac{L_\Phi}{2}\|\bar{x}_{k+1} - \bar{x}_k\|^2,$$

and noticing that the scalar $\bar{v}_k, \bar{u}_k$ are random variables, we have

$$\mathbb{E}\left[\frac{\Phi(\bar{x}_{k+1}) - \Phi(\bar{x}_k)}{\gamma_x \bar{v}_{k+1}^{-\alpha}}\right]$$

$$\leqslant -\mathbb{E}\left[\left\langle\nabla\Phi(\bar{x}_k), \frac{\mathbf{1}^T}{n}\nabla_x F(\mathbf{x}_k, \mathbf{y}_k; \xi_k)\right\rangle\right] - \mathbb{E}\left[\left\langle\nabla\Phi(\bar{x}_k), \frac{(\tilde{\boldsymbol{v}}_{k+1}^{-\alpha})^T}{n\bar{v}_{k+1}^{-\alpha}}\nabla_x F(\mathbf{x}_k, \mathbf{y}_k; \xi_k^x)\right\rangle\right]$$

$$+ \frac{\gamma_x L_\Phi}{2}\mathbb{E}\left[\frac{1}{\bar{v}_{k+1}^{-\alpha}}\left\|\left(\frac{\bar{v}_{k+1}^{-\alpha}\mathbf{1}^T}{n} + \frac{(\tilde{\boldsymbol{v}}_{k+1}^{-\alpha})^T}{n}\right)\nabla_x F(\mathbf{x}_k, \mathbf{y}_k; \xi_k^x)\right\|^2\right].$$

$$(19)$$

Then, we bound the inner-product terms on the RHS. Firstly,

$$
\begin{aligned}
&- \mathbb{E}\left[\left\langle \nabla \Phi\left(\bar{x}_k\right), \frac{\mathbf{1}^T}{n} \nabla_x F\left(\mathbf{x}_k, \mathbf{y}_k; \xi_k^x\right)\right\rangle\right] \\
&= -\mathbb{E}\left[\left\langle \nabla \Phi\left(\bar{x}_k\right), \frac{\mathbf{1}^T}{n} \nabla_x F\left(\mathbf{x}_k, \mathbf{y}_k\right) - \frac{\mathbf{1}^T}{n} \nabla_x F\left(\mathbf{1}\bar{x}_k, \mathbf{1}\bar{y}_k\right) + \frac{\mathbf{1}^T}{n} \nabla_x F\left(\mathbf{1}\bar{x}_k, \mathbf{1}\bar{y}_k\right)\right\rangle\right] \\
&\leqslant \frac{1}{4}\mathbb{E}\left[\|\nabla \Phi\left(\bar{x}_k\right)\|^2\right] + \mathbb{E}\left[\left\|\frac{\mathbf{1}^T}{n} \nabla_x F\left(\mathbf{x}_k, \mathbf{y}_k\right) - \frac{\mathbf{1}^T}{n} \nabla_x F\left(\mathbf{1}\bar{x}_k, \mathbf{1}\bar{y}_k\right)\right\|^2\right] \\
&\quad + \frac{1}{2}\left(\mathbb{E}\left[\|\nabla \Phi\left(\bar{x}_k\right) - \nabla_x f\left(\bar{x}_k, \bar{y}_k\right)\|^2\right] - \mathbb{E}\left[\|\nabla \Phi\left(\bar{x}_k\right)\|^2\right] - \mathbb{E}\left[\|\nabla_x f\left(\bar{x}_k, \bar{y}_k\right)\|^2\right]\right) \\
&\leqslant -\frac{1}{4}\mathbb{E}\left[\|\nabla \Phi\left(\bar{x}_k\right)\|^2\right] + \frac{L^2}{n}\mathbb{E}\left[\Delta_k\right] + \frac{L^2}{2}\mathbb{E}\left[\|\bar{y}_k - y^*\left(\bar{x}_k\right)\|^2\right] - \frac{1}{2}\mathbb{E}\left[\|\nabla_x f\left(\bar{x}_k, \bar{y}_k\right)\|^2\right].
\end{aligned}
\tag{20}
$$

wherein the last inequality we have used the smoothness of the objective functions. Then, for the second inner-product in (19), using Young's inequality we have

$$
\begin{aligned}
&- \mathbb{E}\left[\left\langle \nabla \Phi\left(\bar{x}_k\right), \frac{\left(\tilde{\boldsymbol{v}}_{k+1}^{-\alpha}\right)^T}{n \bar{v}_{k+1}^{-\alpha}} \nabla_x F\left(\mathbf{x}_k, \mathbf{y}_k; \xi_k^x\right)\right\rangle\right] \\
&\leqslant \frac{1}{8}\mathbb{E}\left[\|\nabla \Phi\left(\bar{x}_k\right)\|^2\right] + 2\mathbb{E}\left[\left\|\frac{\left(\tilde{\boldsymbol{v}}_{k+1}^{-\alpha}\right)^T}{n \bar{v}_{k+1}^{-\alpha}} \nabla_x F\left(\mathbf{x}_k, \mathbf{y}_k; \xi_k^x\right)\right\|^2\right].
\end{aligned}
\tag{21}
$$

Then, for the last term on the RHS of (18), recalling the definition of stepsize inconsistency in (7), we have

$$
\begin{aligned}
&\frac{\gamma_x L_\Phi}{2}\mathbb{E}\left[\frac{1}{\bar{v}_{k+1}^{-\alpha}}\left\|\left(\frac{\bar{v}_{k+1}^{-\alpha}\mathbf{1}^T}{n} + \frac{\left(\tilde{\boldsymbol{v}}_{k+1}^{-\alpha}\right)^T}{n}\right) \nabla_x F\left(\mathbf{x}_k, \mathbf{y}_k; \xi_k^x\right)\right\|^2\right] \\
&\leqslant \frac{\gamma_x L_\Phi\left(1 + \zeta_v^2\right)}{n}\mathbb{E}\left[\bar{v}_{k+1}^{-\alpha}\|\nabla_x F\left(\mathbf{x}_k, \mathbf{y}_k; \xi_k^x\right)\|^2\right].
\end{aligned}
\tag{22}
$$

Plugging the obtained inequalities into (18) and telescoping the terms, we get

$$
\begin{aligned}
&\sum_{k=0}^{K-1}\mathbb{E}\left[\|\nabla \Phi\left(\bar{x}_k\right)\|^2\right] \\
&\leqslant 8\sum_{k=0}^{K-1}\mathbb{E}\left[\frac{\Phi\left(\bar{x}_k\right) - \Phi\left(\bar{x}_{k+1}\right)}{\gamma_x \bar{v}_k^{-\alpha}}\right] - 4\sum_{k=0}^{K-1}\mathbb{E}\left[\|\nabla_x f\left(\bar{x}_k, \bar{y}_k\right)\|^2\right] \\
&\quad + 4L^2\sum_{k=0}^{K-1}\mathbb{E}\left[\|\bar{y}_k - \bar{y}^*\|^2\right] + \frac{8L^2}{n}\sum_{k=0}^{K-1}\mathbb{E}\left[\Delta_k\right] \\
&\quad + \frac{8\gamma_x L_\Phi\left(1 + \zeta_v^2\right)}{n}\sum_{k=0}^{K-1}\mathbb{E}\left[\bar{v}_k^{-\alpha}\|\nabla_x F\left(\mathbf{x}_k, \mathbf{y}_k; \xi_k^x\right)\|^2\right] \\
&\quad + 16\sum_{k=0}^{K-1}\mathbb{E}\left[\left\|\frac{\left(\tilde{\boldsymbol{v}}_{k+1}^{-\alpha}\right)^T}{n \bar{v}_{k+1}^{-\alpha}} \nabla_x F\left(\mathbf{x}_k, \mathbf{y}_k; \xi_k^x\right)\right\|^2\right].
\end{aligned}
\tag{23}
$$

Now it remains to bound the first term on the RHS of the above inequality. We have

$$
\begin{aligned}
\sum_{k=0}^{K-1} & \mathbb{E}\left[\frac{\Phi\left(\bar{x}_k\right)-\Phi\left(\bar{x}_{k+1}\right)}{\gamma_x \bar{v}_{k+1}^{-\alpha}}\right] \\
&= \sum_{k=0}^{K-1} \mathbb{E}\left[\frac{\Phi\left(\bar{x}_k\right)}{\gamma_x \bar{v}_k^{-\alpha}}-\frac{\Phi\left(\bar{x}_{k+1}\right)}{\gamma_x \bar{v}_{k+1}^{-\alpha}}+\Phi\left(\bar{x}_k\right)\left(\frac{1}{\gamma_x \bar{v}_{k+1}^{-\alpha}}-\frac{1}{\gamma_x \bar{v}_k^{-\alpha}}\right)\right] \\
&\leqslant \mathbb{E}\left[\frac{\Phi_{\max}}{\gamma_x \bar{v}_0^{-\alpha}}-\frac{\Phi^*}{\gamma_x \bar{v}_K^{-\alpha}}\right]+\sum_{k=0}^{K-1} \mathbb{E}\left[\Phi_{\max}\left(\frac{1}{\gamma_x \bar{v}_{k+1}^{-\alpha}}-\frac{1}{\gamma_x \bar{v}_k^{-\alpha}}\right)\right] \\
&\leqslant \frac{\left(\Phi_{\max}-\Phi^*\right)}{\gamma_x} \mathbb{E}\left[\bar{v}_K^\alpha\right] \\
&\leqslant \frac{\left(\Phi_{\max}-\Phi^*\right)\left(K C^2\right)^\alpha}{\gamma_x},
\end{aligned}
\tag{24}
$$

wherein the last inequality we have used Assumption 4. Noticing that $\mathbb{E}\left[\left\|\bar{y}_k-y^*\left(\bar{x}_k\right)\right\|^2\right] \leqslant \frac{2}{\mu} \mathbb{E}\left[f\left(\bar{x}_k, y^*\left(\bar{x}_k\right)\right)-f\left(\bar{x}_k, \bar{y}_k\right)\right]$, we thus complete the proof. $\qquad\square$

Next, we try to bound the last four terms $S_1$-$S_4$ in (18) respectively. For $S_1$, we have the asymptotic convergence for both primal and dual variables in the following lemma.

**Lemma 6.** *Suppose Assumption 1-5 hold. We have*

$$
\frac{1}{nK} \sum_{k=0}^{K-1} \mathbb{E}\left[\bar{v}_{k+1}^{-\alpha}\left\|\nabla_x F\left(\mathbf{x}_k, \mathbf{y}_k ; \xi_k^x\right)\right\|^2\right] \leqslant \frac{C^{2-2\alpha}}{(1-\alpha) K^\alpha},
\tag{25}
$$

*and*

$$
\frac{1}{nK} \sum_{k=0}^{K-1} \mathbb{E}\left[\bar{u}_{k+1}^{-\beta}\left\|\nabla_y F\left(\mathbf{x}_k, \mathbf{y}_k ; \xi_k^y\right)\right\|^2\right] \leqslant \frac{C^{2-2\beta}}{(1-\beta) K^\beta}.
\tag{26}
$$

*Proof.* With the help of Lemma 2 and Assumption 4, taking the primal variable $x$ as an example, noticing that $v_{i,0}>0$, we have

$$
\begin{aligned}
\frac{1}{K} \sum_{k=0}^{K-1} & \mathbb{E}\left[\bar{v}_{k+1}^{-\alpha}\left\|\nabla_x F\left(\mathbf{x}_k, \mathbf{y}_k ; \xi_k^x\right)\right\|^2\right] \\
&= \frac{1}{K} \sum_{k=0}^{K-1} \frac{1}{n} \sum_{i=1}^n \frac{\left\|\nabla_x F_i\left(x_{i,k}, y_{i,k} ; \xi_{i,k}^x\right)\right\|^2}{\bar{v}_{k+1}^\alpha} \\
&\leqslant \frac{1}{K} \sum_{k=0}^{K-1} \frac{1}{n} \sum_{i=1}^n \frac{\left\|\nabla_x F_i\left(x_{i,k}, y_{i,k} ; \xi_{i,k}^x\right)\right\|^2}{\left(\sum_{t=0}^k \frac{1}{n} \sum_{j=1}^n\left\|\nabla_x F_j\left(x_{j,t}, y_{j,t} ; \xi_{j,t}^x\right)\right\|^2\right)^\alpha} \\
&\leqslant \frac{1}{1-\alpha} \frac{1}{K}\left(\sum_{k=0}^{K-1} \frac{1}{n} \sum_{i=1}^n\left\|\nabla_x F_i\left(x_{i,k}, y_{i,k} ; \xi_{i,k}^x\right)\right\|^2\right)^{1-\alpha} \leqslant \frac{C^{2-2\alpha}}{(1-\alpha) K^\alpha}.
\end{aligned}
$$

The similar result can be obtained for dual variable $y$ and we thus complete the proof. $\qquad\square$

Next, we bound the the consensus error term $S_2$ in the following lemma.

**Lemma 7.** *Suppose Assumption 1-5 hold. We have*

$$
\begin{aligned}
\frac{1}{K} \sum_{k=0}^{K} \mathbb{E}\left[\Delta_k\right] &\leqslant \frac{2\mathbb{E}\left[\Delta_0\right]}{(1-\rho_W)K} \\
&+ \frac{8n\rho_W\gamma_x^2\left(1+\zeta_v^2\right)}{(1-\rho_W)^2}\left(\frac{C^{2-4\alpha}}{(1-2\alpha)K^{2\alpha}}\mathbb{I}_{\alpha<1/2} + \frac{1+\log v_K - \log v_1}{K\bar{v}_1^{2\alpha-1}}\mathbb{I}_{\alpha\geqslant1/2}\right) \\
&+ \frac{8n\rho_W\gamma_y^2\left(1+\zeta_u^2\right)}{(1-\rho_W)^2}\left(\frac{C^{2-4\beta}}{(1-2\beta)K^{2\beta}}\mathbb{I}_{\beta<1/2} + \frac{1+\log u_K - \log v_1}{K\bar{u}_1^{2\beta-1}}\mathbb{I}_{\beta\geqslant1/2}\right),
\end{aligned}
\tag{27}
$$

*where* $\mathbb{I}_{[\cdot]} \in \{0,1\}$ *is the indicator for specific condition, and the initial consensus error* $\Delta_0$ *can be set to* $0$ *with proper initialization.*

*Proof.* Firstly, for the primal variables, we have

$$
\begin{aligned}
&\mathbb{E}\left[\left\|\mathbf{x}_{k+1} - \mathbf{1}\bar{x}_{k+1}\right\|^2\right] \\
&= \mathbb{E}\left[\left\|W\left(\mathbf{x}_k - \gamma_x V_{k+1}^{-\alpha}\nabla_x F\left(\mathbf{x}_k, \mathbf{y}_k; \xi_k^x\right)\right) - \mathbf{J}\left(\mathbf{x}_k - \gamma_x V_{k+1}^{-\alpha}\nabla_x F\left(\mathbf{x}_k, \mathbf{y}_k; \xi_k^x\right)\right)\right\|^2\right] \\
&\leqslant \frac{1+\rho_W}{2}\mathbb{E}\left[\left\|\mathbf{x}_k - \mathbf{1}\bar{x}_k\right\|^2\right] + \frac{2\gamma_x^2\left(1+\rho_W\right)\rho_W}{1-\rho_W}\mathbb{E}\left[\bar{v}_{k+1}^{-2\alpha}\left\|\nabla_x F\left(\mathbf{x}_k, \mathbf{y}_k; \xi_k^x\right)\right\|^2\right] \\
&+ \frac{2\gamma_x^2\left(1+\rho_W\right)\rho_W}{1-\rho_W}\mathbb{E}\left[\left\|\left(V_{k+1}^{-\alpha} - \bar{v}_{k+1}^{-\alpha}\mathbf{I}\right)\nabla_x F\left(\mathbf{x}_k, \mathbf{y}_k; \xi_k^x\right)\right\|^2\right].
\end{aligned}
\tag{28}
$$

Then, by the definition of $\zeta_v$ in (7), we have

$$
\mathbb{E}\left[\left\|\left(V_{k+1}^{-\alpha} - \bar{v}_{k+1}^{-\alpha}\mathbf{I}\right)\nabla_x F\left(\mathbf{x}_k, \mathbf{y}_k; \xi_k^x\right)\right\|^2\right] \leqslant \zeta_v^2\mathbb{E}\left[\bar{v}_{k+1}^{-2\alpha}\left\|\nabla_x F\left(\mathbf{x}_k, \mathbf{y}_k; \xi_k^x\right)\right\|^2\right],
\tag{29}
$$

and thus

$$
\begin{aligned}
&\sum_{k=0}^{K-1} \mathbb{E}\left[\left\|\mathbf{x}_{k+1} - \mathbf{1}\bar{x}_{k+1}\right\|^2\right] \\
&\leqslant \frac{2}{1-\rho_W}\mathbb{E}\left[\left\|\mathbf{x}_k - \mathbf{1}\bar{x}_k\right\|^2\right] + \frac{8\gamma_x^2\rho_W\left(1+\zeta_v^2\right)}{(1-\rho_W)^2}\sum_{k=0}^{K-1}\mathbb{E}\left[\bar{v}_{k+1}^{-2\alpha}\left\|\nabla_x F\left(\mathbf{x}_k, \mathbf{y}_k; \xi_k^x\right)\right\|^2\right].
\end{aligned}
\tag{30}
$$

Then, we bound the last term on the RHS of the above inequality by Lemma 6. For the case $\alpha < 1/2$, by Assumption 4 we have

$$
\begin{aligned}
&\sum_{k=0}^{K-1}\mathbb{E}\left[\bar{v}_{k+1}^{-2\alpha}\left\|\nabla_x F\left(\mathbf{x}_k, \mathbf{y}_k; \xi_k^x\right)\right\|^2\right] \\
&= \sum_{k=0}^{K-1}\sum_{i=1}^{n}\mathbb{E}\left[\frac{\left\|\nabla_x F_i\left(x_{i,k}, y_{i,k}; \xi_{i,k}^x\right)\right\|^2}{\bar{v}_{k+1}^{2\alpha}}\right] \leqslant \frac{n\left(KC^2\right)^{1-2\alpha}}{(1-2\alpha)};
\end{aligned}
\tag{31}
$$

for the case $\alpha \geqslant 1/2$, with the help of Lemma 2, we have

$$
\begin{aligned}
&\sum_{k=0}^{K-1}\mathbb{E}\left[\bar{v}_{k+1}^{-2\alpha}\left\|\nabla_x F\left(\mathbf{x}_k, \mathbf{y}_k; \xi_k^x\right)\right\|^2\right] \\
&= \sum_{k=0}^{K-1}\sum_{i=1}^{n}\mathbb{E}\left[\frac{\left\|\nabla_x F_i\left(x_{i,k}, y_{i,k}; \xi_{i,k}^x\right)\right\|^2}{\bar{v}_{k+1} \cdot \bar{v}_{k+1}^{2\alpha-1}}\right] \leqslant \frac{n\left(1+\log v_T - \log v_1\right)}{\bar{v}_1^{2\alpha-1}}.
\end{aligned}
\tag{32}
$$

For the dual variable, we have

$$
\begin{aligned}
\mathbf{y}_{k+1} &= \mathcal{P}_{\mathcal{Y}}\left(W\left(\mathbf{y}_k + \gamma_y U_{k+1}^{-\beta}\nabla_y F\left(\mathbf{x}_k, \mathbf{y}_k; \xi_k^y\right)\right)\right) \\
&= W\mathbf{y}_k + \gamma_y \nabla_y \hat{G}
\end{aligned}
$$

where

$$\nabla_y \hat{G} = \frac{1}{\gamma_y} \left( \mathcal{P}_{\mathcal{Y}} \left( W \left( \mathbf{y}_k + \gamma_y U_{k+1}^{-\beta} \nabla_y F \left( \mathbf{x}_k, \mathbf{y}_k; \xi_k^y \right) \right) \right) - W \mathbf{y}_k \right).$$

Then, using Young's inequality with parameter $\lambda$, we have

$$\mathbb{E} \left[ \| \mathbf{y}_{k+1} - \mathbf{1} \bar{y}_{k+1} \|^2 \right]$$

$$= \mathbb{E} \left[ \left\| W \mathbf{y}_k + \gamma_y \nabla_y \hat{G} - \mathbf{J} \left( W \mathbf{y}_k + \gamma_y \nabla_y \hat{G} \right) \right\|^2 \right]$$

$$\leqslant (1 + \lambda) \rho_W \mathbb{E} \left[ \| \mathbf{y}_k - \mathbf{J} \mathbf{y}_k \|^2 \right]$$

$$+ \left( 1 + \frac{1}{\lambda} \right) \mathbb{E} \left[ \left\| \mathcal{P}_{\mathcal{Y}} \left( W \left( \mathbf{y}_k + \gamma_y U_{k+1}^{-\beta} \nabla_y F \left( \mathbf{x}_k, \mathbf{y}_k; \xi_k^y \right) \right) \right) - W \mathbf{y}_k \right\|^2 \right]$$

$$\leqslant \frac{1 + \rho_W}{2} \mathbb{E} \left[ \| \mathbf{y}_k - \mathbf{J} \mathbf{y}_k \|^2 \right]$$

$$+ \frac{1 + \rho_W}{1 - \rho_W} \mathbb{E} \left[ \left\| \mathcal{P}_{\mathcal{Y}} \left( W \left( \mathbf{y}_k + \gamma_y U_{k+1}^{-\beta} \nabla_y F \left( \mathbf{x}_k, \mathbf{y}_k; \xi_k^y \right) \right) \right) - W \mathbf{y}_k \right\|^2 \right].$$

Noticing that $W \mathbf{y}_k = \mathcal{P}_{\mathcal{Y}} \left( W \mathbf{y}_k \right)$ holds for convex set $\mathcal{Y}$, we get

$$\mathbb{E} \left[ \| \mathbf{y}_{k+1} - \mathbf{1} \bar{y}_{k+1} \|^2 \right]$$

$$\leqslant \frac{1 + \rho_W}{2} \mathbb{E} \left[ \| \mathbf{y}_k - \mathbf{J} \mathbf{y}_k \|^2 \right]$$

$$+ \frac{1 + \rho_W}{1 - \rho_W} \mathbb{E} \left[ \left( \left\| \mathcal{P}_{\mathcal{Y}} \left( W \left( \mathbf{y}_k + \gamma_y U_{k+1}^{-\beta} \nabla_y F \left( \mathbf{x}_k, \mathbf{y}_k; \xi_k^y \right) \right) \right) - \mathcal{P}_{\mathcal{Y}} \left( W \mathbf{y}_k \right) \right\| \right)^2 \right]$$

$$\leqslant \frac{1 + \rho_W}{2} \mathbb{E} \left[ \| \mathbf{y}_k - \mathbf{J} \mathbf{y}_k \|^2 \right] + \frac{1 + \rho_W}{1 - \rho_W} \mathbb{E} \left[ \left\| \gamma_y U_{k+1}^{-\beta} \nabla_y F \left( \mathbf{x}_k, \mathbf{y}_k; \xi_k^y \right) \right\|^2 \right]$$

$$\leqslant \frac{1 + \rho_W}{2} \mathbb{E} \left[ \| \mathbf{y}_k - \mathbf{J} \mathbf{y}_k \|^2 \right] + \frac{4 \gamma_y^2 \left( 1 + \zeta_u^2 \right)}{\left( 1 - \rho_W \right)} \mathbb{E} \left[ \bar{u}_{k+1}^{-2\beta} \| \nabla_y F \left( \mathbf{x}_k, \mathbf{y}_k; \xi_k^y \right) \|^2 \right],$$

where we have used the non-expansiveness of projection operator. Then, we have

$$\sum_{k=0}^{K-1} \mathbb{E} \left[ \| \mathbf{y}_k - \mathbf{1} \bar{y}_k \|^2 \right]$$

$$\leqslant \frac{2}{1 - \rho_W} \mathbb{E} \left[ \| \mathbf{y}_0 - \mathbf{J} \mathbf{y}_0 \|^2 \right] + \frac{8 \gamma_y^2 \left( 1 + \zeta_u^2 \right)}{\left( 1 - \rho_W \right)^2} \sum_{k=0}^{K-1} \mathbb{E} \left[ \bar{u}_{k+1}^{-2\beta} \| \nabla_y F \left( \mathbf{x}_k, \mathbf{y}_k; \xi_k^y \right) \|^2 \right].$$

Similar to the primal variable, we can bound the last term above, which completes the proof. $\qquad \square$

Finally, we need to bound the term $S_3$ i.e., the optimality gap in dual variable. The intuition of the proof relies on the adaptive two time-scale protocol, that is, for given $\alpha$ and $\beta$, we try to find the threshold value of the iterations $k_0$, after which the inner sub-problem can be well solved (faster) to ensure that the computation of outer sub-problem can be solved accurately (slower). In specific, we suppose $\bar{u}_k \leqslant G$ hold for $k = 0, 1, \cdots, k_0 - 1$, then the analysis is divided into two phases.

**Lemma 8** (First phase). *Suppose Assumption 1-5 hold. If $\bar{u}_k \leqslant G, k = 0, 1, \cdots, k_0 - 1$, we have*

$$
\sum_{k=0}^{k_0-1} \mathbb{E} \left[ f\left(\bar{x}_k, y^*\left(\bar{x}_k\right)\right) - f\left(\bar{x}_k, \bar{y}_k\right) \right]
$$

$$
\leqslant \sum_{k=0}^{k_0-1} \mathbb{E} \left[ S_{1,k} \right] + \frac{\gamma_x^2 \kappa^2 \left(1 + \zeta_v^2\right) G^{2\beta}}{n \mu \gamma_y^2} \sum_{k=0}^{k_0-1} \mathbb{E} \left[ \bar{v}_{k+1}^{-2\alpha} \left\| \nabla_x F\left(\mathbf{x}_k, \mathbf{y}_k; \xi_k^x\right) \right\|^2 \right]
$$

$$
+ \frac{\gamma_y \left(1 + \zeta_u^2\right)}{n} \sum_{k=0}^{k_0-1} \mathbb{E} \left[ \bar{u}_{k+1}^{-\beta} \left\| \nabla_y F\left(\mathbf{x}_k, \mathbf{y}_k; \xi_k\right) \right\|^2 \right] + \frac{4\kappa L}{n} \sum_{k=0}^{k_0-1} \mathbb{E} \left[ \left\| \mathbf{x}_k - \mathbf{1}\bar{x}_k \right\|^2 \right]
$$

$$
+ \frac{4}{\mu} \sum_{k=0}^{k_0-1} \mathbb{E} \left[ \left\| \frac{\tilde{\boldsymbol{u}}_{k+1}^{-\beta}}{n \bar{u}_{k+1}^{-\beta}} \nabla_y F\left(\mathbf{x}_k, \mathbf{y}_k; \xi_k^y\right) \right\|^2 \right] + C \sum_{k=0}^{k_0-1} \mathbb{E} \left[ \sqrt{\frac{1}{n} \left\| \mathbf{y}_k - \mathbf{1}\bar{y}_k \right\|^2} \right],
$$

(33)

*where*

$$
S_{1,k} := \frac{1 - 3\mu\gamma_y \bar{u}_{k+1}^{-\beta}/4}{2\gamma_y \bar{u}_{k+1}^{-\beta} n} \left\| \mathbf{y}_k - \mathbf{1}y^*\left(\bar{x}_k\right) \right\|^2 - \frac{\left\| \mathbf{y}_{k+1} - \mathbf{1}y^*\left(\bar{x}_{k+1}\right) \right\|^2}{\left( 2 + \mu\gamma_y \bar{u}_{k+1}^{-\beta} \right) \gamma_y \bar{u}_{k+1}^{-\beta} n}.
$$

(34)

*Proof.* Firstly, we use Young's inequality with parameter $\lambda_k$,

$$
\frac{1}{n} \left\| \mathbf{y}_{k+1} - \mathbf{1}\bar{y}^*\left(\bar{x}_{k+1}\right) \right\|^2
$$

$$
\leqslant \frac{\left(1 + \lambda_k\right)}{n} \left\| \mathbf{y}_{k+1} - \mathbf{1}y^*\left(\bar{x}_k\right) \right\|^2 + \left( 1 + \frac{1}{\lambda_k} \right) \left\| y^*\left(\bar{x}_k\right) - y^*\left(\bar{x}_{k+1}\right) \right\|^2.
$$

(35)

Recalling that $\mathbf{y}_{k+1} = \mathcal{P}_{\mathcal{Y}} \left( W \left( \mathbf{y}_k + \gamma_y U_{k+1}^{-\beta} \nabla_y F\left(\mathbf{x}_k, \mathbf{y}_k; \xi_k^y\right) \right) \right)$, we further define

$$
\hat{\mathbf{y}}_{k+1} = W \left( \mathbf{y}_k + \gamma_y U_{k+1}^{-\beta} \nabla_y F\left(\mathbf{x}_k, \mathbf{y}_k; \xi_k^y\right) \right).
$$

Then, for the first term on the RHS of (35), by the non-expansiveness property of projection operator $\mathcal{P}_{\mathcal{Y}}(\cdot)$ (c.f., Lemma 1 in (Nedic et al., 2010)), we have

$$
\frac{1}{n} \left\| \mathbf{y}_{k+1} - \mathbf{1}y^*\left(\bar{x}_k\right) \right\|^2
$$

$$
\leqslant \frac{1}{n} \left\| \hat{\mathbf{y}}_{k+1} - \mathbf{1}y^*\left(\bar{x}_k\right) \right\|^2 - \frac{1}{n} \left\| \mathbf{y}_{k+1} - \hat{\mathbf{y}}_{k+1} \right\|^2
$$

$$
\leqslant \frac{1}{n} \left\| \mathbf{y}_k - \mathbf{1}y^*\left(\bar{x}_k\right) \right\|^2 + \frac{\gamma_y^2}{n} \left\| U_{k+1}^{-\beta} \nabla_y F\left(\mathbf{x}_k, \mathbf{y}_k; \xi_k^y\right) \right\|^2
$$

$$
- \frac{1}{n} \sum_{i=1}^n 2 \left\langle \gamma_y \bar{u}_{k+1}^{-\beta} g_{i,k}^y, y_{i,k} - y^*\left(\bar{x}_k\right) \right\rangle - \frac{1}{n} \sum_{i=1}^n 2 \left\langle \gamma_y \left( u_{i,k+1}^{-\beta} - \bar{u}_{k+1}^{-\beta} \right) g_{i,k}^y, y_{i,k} - y^*\left(\bar{x}_k\right) \right\rangle,
$$

(36)

wherein the last inequality we have used the fact $\|W\|_2^2 \leqslant 1$. Then, multiplying by $1/\left(\gamma_y \bar{u}_{k+1}^{-\beta}\right)$ on both sides of (35) we get

$$
\frac{1}{n\gamma_y \bar{u}_{k+1}^{-\beta}} \left\| \mathbf{y}_{k+1} - \mathbf{1}y^*\left(\bar{x}_k\right) \right\|^2
$$

$$
\leqslant \frac{1 + \lambda_k}{\lambda_k \gamma_y \bar{u}_{k+1}^{-\beta}} \left\| \bar{y}^*\left(\bar{x}_k\right) - \bar{y}^*\left(\bar{x}_{k+1}\right) \right\|^2
$$

$$
+ \left(1 + \lambda_k\right) \left( \frac{1}{n\gamma_y \bar{u}_{k+1}^{-\beta}} \left\| \mathbf{y}_k - \mathbf{1}y^*\left(\bar{x}_k\right) \right\|^2 + \frac{\gamma_y}{n\bar{u}_{k+1}^{-\beta}} \left\| U_{k+1}^{-\beta} \nabla_y F\left(\mathbf{x}_k, \mathbf{y}_k; \xi_k^y\right) \right\|^2 \right)
$$

$$
- \left(1 + \lambda_k\right) \left( \frac{1}{n} \sum_{i=1}^n 2 \left\langle g_{i,k}^y, y_{i,k} - y^*\left(\bar{x}_k\right) \right\rangle - \frac{1}{n} \sum_{i=1}^n 2 \left\langle \left( \frac{u_{i,k+1}^{-\beta} - \bar{u}_{k+1}^{-\beta}}{\bar{u}_{k+1}^{-\beta}} \right) g_{i,k}^y, y_{i,k} - y^*\left(\bar{x}_k\right) \right\rangle \right).
$$

(37)

For the inner-product terms on the RHS, taking expectation on both sides, we have

$$
\frac{1}{n}\sum_{i=1}^{n}\mathbb{E}\left[-2\left\langle g_{i,k}^{y}, y_{i,k}-y^{*}\left(\bar{x}_{k}\right)\right\rangle\right]
$$

$$
=\frac{1}{n}\sum_{i=1}^{n}\mathbb{E}\left[-2\left\langle \nabla_{y}f_{i}\left(\bar{x}_{k}, y_{i,k}\right), y_{i,k}-y^{*}\left(\bar{x}_{k}\right)\right\rangle\right]
$$

$$
+\frac{1}{n}\sum_{i=1}^{n}\mathbb{E}\left[-2\left\langle \nabla_{y}f_{i}\left(x_{i,k}, y_{i,k}\right)-\nabla_{y}f_{i}\left(\bar{x}_{k}, y_{i,k}\right), y_{i,k}-y^{*}\left(\bar{x}_{k}\right)\right\rangle\right]
$$

$$
\leqslant\frac{1}{n}\sum_{i=1}^{n}\mathbb{E}\left[-2\left(f_{i}\left(\bar{x}_{k}, y^{*}\left(\bar{x}_{k}\right)\right)-f_{i}\left(\bar{x}_{k}, y_{i,k}\right)\right)-\mu\left\|y_{i,k}-y^{*}\left(\bar{x}_{k}\right)\right\|^{2}\right] \tag{38}
$$

$$
+\frac{1}{n}\sum_{i=1}^{n}\mathbb{E}\left[\frac{8}{\mu}\left\|\nabla_{y}f_{i}\left(x_{i,k}, y_{i,k}\right)-\nabla_{y}f_{i}\left(\bar{x}_{k}, y_{i,k}\right)\right\|^{2}+\frac{\mu}{8}\left\|y_{i,k}-\bar{y}^{*}\left(\bar{x}_{k}\right)\right\|^{2}\right]
$$

$$
\leqslant\mathbb{E}\left[-2\left(f\left(\bar{x}_{k}, y^{*}\left(\bar{x}_{k}\right)\right)-f\left(\bar{x}_{k}, \bar{y}_{k}\right)\right)\right]+\frac{1}{n}\sum_{i=1}^{n}\mathbb{E}\left[-2\left(f_{i}\left(\bar{x}_{k}, \bar{y}_{k}\right)-f_{i}\left(\bar{x}_{k}, y_{i,k}\right)\right)\right]
$$

$$
+\frac{8\kappa L}{n}\sum_{i=1}^{n}\mathbb{E}\left[\left\|x_{i,k}-\bar{x}_{k}\right\|^{2}\right]-\frac{7\mu}{8n}\sum_{i=1}^{n}\mathbb{E}\left[\left\|y_{i,k}-y^{*}\left(\bar{x}_{k}\right)\right\|^{2}\right],
$$

where we have used Young's inequality and strongly-concavity of $f_i$, and

$$
\frac{1}{n}\sum_{i=1}^{n}\mathbb{E}\left[-2\left\langle\left(\frac{u_{i,k+1}^{-\beta}-\bar{u}_{k+1}^{-\beta}}{\bar{u}_{k+1}^{-\beta}}\right)g_{i,k}^{y}, y_{i,k}-y^{*}\left(\bar{x}_{k}\right)\right\rangle\right]
$$

$$
\leqslant\frac{1}{n}\sum_{i=1}^{n}\mathbb{E}\left[\frac{8}{\mu}\left\|\left(\frac{u_{i,k+1}^{-\beta}-\bar{u}_{k+1}^{-\beta}}{\bar{u}_{k+1}^{-\beta}}\right)g_{i,k}^{y}\right\|^{2}+\frac{\mu}{8}\left\|y_{i,k}-y^{*}\left(\bar{x}_{k}\right)\right\|^{2}\right]. \tag{39}
$$

For the consensus error of dual variable on objective function, using strongly-concavity of $f_i$ and Jensen's inequality, we have

$$
\frac{1}{n}\sum_{i=1}^{n}-2\left(f_{i}\left(\bar{x}_{k}, \bar{y}_{k}\right)-f_{i}\left(\bar{x}_{k}, y_{i,k}\right)\right)
$$

$$
\leqslant\frac{1}{n}\sum_{i=1}^{n}2\left\langle\nabla_{y}f_{i}\left(\bar{x}_{k}, \bar{y}_{k}\right), y_{i,k}-\bar{y}_{k}\right\rangle-\frac{\mu}{n}\left\|\mathbf{y}_{k}-\mathbf{1}\bar{y}_{k}\right\|^{2} \tag{40}
$$

$$
\leqslant 2C\frac{1}{n}\sum_{i=1}^{n}\left\|y_{i,k}-\bar{y}_{k}\right\|\leqslant 2C\sqrt{\frac{1}{n}\left\|\mathbf{y}_{k}-\mathbf{1}\bar{y}_{k}\right\|^{2}}.
$$

Letting $\lambda_{k}=\mu\gamma_{y}\bar{u}_{k+1}^{-\beta}/2$, we get

$$
\mathbb{E}\left[f\left(\bar{x}_{k}, \bar{y}^{*}\left(\bar{x}_{k}\right)\right)-f\left(\bar{x}_{k}, \bar{y}_{k}\right)\right]
$$

$$
\leqslant\mathbb{E}\left[\frac{1-3\mu\gamma_{y}\bar{u}_{k+1}^{-\beta}/4}{2\gamma_{y}\bar{u}_{k+1}^{-\beta}n}\left\|\mathbf{y}_{k}-\mathbf{1}y^{*}\left(\bar{x}_{k}\right)\right\|^{2}-\frac{\left\|\mathbf{y}_{k+1}-\mathbf{1}y^{*}\left(\bar{x}_{k+1}\right)\right\|^{2}}{\left(2+\mu\gamma_{y}\bar{u}_{k+1}^{-\beta}\right)\gamma_{y}\bar{u}_{k+1}^{-\beta}n}\right]
$$

$$
+\frac{\gamma_{x}^{2}\kappa^{2}\left(1+\zeta_{v}^{2}\right)G^{2\beta}}{n\mu\gamma_{y}^{2}}\mathbb{E}\left[\bar{v}_{k+1}^{-2\alpha}\left\|\nabla_{x}F\left(\mathbf{x}_{k}, \mathbf{y}_{k}; \xi_{k}^{x}\right)\right\|^{2}\right] \tag{41}
$$

$$
+\frac{\gamma_{y}\left(1+\zeta_{u}^{2}\right)}{n}\sum_{i=1}^{n}\mathbb{E}\left[\bar{u}_{k+1}^{-\beta}\left\|\nabla_{y}F\left(\mathbf{x}_{k}, \mathbf{y}_{k}; \xi_{k}\right)\right\|^{2}\right]+\frac{4\kappa L}{n}\mathbb{E}\left[\left\|\mathbf{x}_{k}-\mathbf{1}\bar{y}_{k}\right\|^{2}\right]
$$

$$
+\frac{4}{\mu}\mathbb{E}\left[\left\|\frac{\tilde{\boldsymbol{u}}_{k+1}^{-\beta}}{n\bar{u}_{k+1}^{-\beta}}\nabla_{y}F\left(\mathbf{x}_{k}, \mathbf{y}_{k}; \xi_{k}^{y}\right)\right\|^{2}\right]+C\mathbb{E}\left[\sqrt{\frac{1}{n}\left\|\mathbf{y}_{k}-\mathbf{1}\bar{y}_{k}\right\|^{2}}\right].
$$

By the $\kappa$-smoothness of $y^*$, we have

$$
\begin{aligned}
&\left\| y^* \left( \bar{x}_{k+1} \right) - y^* \left( \bar{x}_k \right) \right\|^2 \\
&\leqslant \kappa^2 \left\| \bar{x}_{k+1} - \bar{x}_k \right\|^2 \\
&= \kappa^2 \left\| \gamma_x \bar{v}_{k+1}^{-\alpha} \frac{\mathbf{1}^T}{n} \nabla_x F \left( \mathbf{x}_k, \mathbf{y}_k; \xi_k \right) - \gamma_x \frac{\left( \tilde{\boldsymbol{v}}_{k+1}^{-\alpha} \right)^T}{n} \nabla_x F \left( \mathbf{x}_k, \mathbf{y}_k; \xi_k^x \right) \right\|^2 \\
&\leqslant \frac{2\gamma_x^2 \kappa^2 \left( 1 + \zeta_v^2 \right) \bar{v}_{k+1}^{-2\alpha}}{n} \left\| \nabla_x F \left( \mathbf{x}_k, \mathbf{y}_k; \xi_k^x \right) \right\|^2 .
\end{aligned}
\tag{42}
$$

Telescoping the obtained terms from 0 to $k_0 - 1$ and noticing that $\bar{u}_k \leqslant G$ for $k \leqslant k_0 - 1$ we complete the proof. $\qquad\square$

For the second phase, i.e., $k \geqslant k_0$, we have the following lemma.

**Lemma 9** (Second phase). *Suppose Assumption 1-5 hold. If $\bar{u}_k \leqslant G, k = 0, 1, \cdots, k_0 - 1$, we have*

$$
\begin{aligned}
&\sum_{k=k_0}^{K-1} \mathbb{E} \left[ f \left( \bar{x}_k, \bar{y}^* \left( \bar{x}_k \right) \right) - f \left( \bar{x}_k, \bar{y}_k \right) \right] \\
&\leqslant \sum_{k=k_0}^{K-1} \mathbb{E} \left[ S_{1,k} \right] + \frac{8\gamma_x^2 \kappa^2 \left( 1 + \zeta_v^2 \right)}{\mu \gamma_y^2 G^{2\alpha - 2\beta}} \sum_{k=k_0}^{K-1} \left\| \nabla_x f \left( \bar{x}_k, \bar{y}_k \right) \right\|^2 \\
&\quad + \left( \frac{8\gamma_x^2 \kappa^2 L^2 \left( 1 + \zeta_v^2 \right)}{n \mu \gamma_y^2 G^{2\alpha - 2\beta}} + \frac{4\kappa L}{n} \right) \sum_{k=k_0}^{K-1} \mathbb{E} \left[ \Delta_k \right] \\
&\quad + \frac{\gamma_y \left( 1 + \zeta_u^2 \right)}{n} \mathbb{E} \left[ \bar{u}_{k+1}^{-\beta} \left\| \nabla_y F \left( \mathbf{x}_k, \mathbf{y}_k; \xi_k \right) \right\|^2 \right] + C \sum_{k=k_0}^{K-1} \mathbb{E} \left[ \sqrt{\frac{1}{n} \left\| \mathbf{y}_k - \mathbf{1}\bar{y}_k \right\|^2} \right] \\
&\quad + \frac{\gamma_x^2 \left( 1 + \zeta_v^2 \right)}{\gamma_y \bar{v}_1^{\alpha - \beta}} \left( \kappa^2 + \frac{2\gamma_x^2 \left( 1 + \zeta_v^2 \right) C^2 \hat{L}^2}{\mu \gamma_y \bar{v}_1^{2\alpha - \beta}} \right) \sum_{k=k_0}^{K-1} \mathbb{E} \left[ \frac{\bar{v}_{k+1}^{-\alpha}}{n} \left\| \nabla_x F \left( \mathbf{x}_k, \mathbf{y}_k; \xi_k^x \right) \right\|^2 \right] \\
&\quad + \frac{2\gamma_x \kappa \left( 1 + \zeta_v \right) C^2}{\mu \gamma_y \bar{v}_1^{\alpha}} \mathbb{E} \left[ \bar{u}_K^{\beta} \right] + \frac{4}{\mu} \sum_{k=k_0}^{K-1} \mathbb{E} \left[ \left\| \frac{\tilde{\boldsymbol{u}}_{k+1}^{-\beta}}{n \bar{u}_{k+1}^{-\beta}} \nabla_y F \left( \mathbf{x}_k, \mathbf{y}_k; \xi_k^y \right) \right\|^2 \right] .
\end{aligned}
\tag{43}
$$

*Proof.* Firstly, by the non-expansiveness of projection operator, we have

$$
\begin{aligned}
&\left\| y_{i,k+1} - y^* \left( \bar{x}_{k+1} \right) \right\|^2 \\
&\leqslant \left\| \hat{y}_{i,k+1} - y^* \left( \bar{x}_{k+1} \right) \right\|^2 - \left\| y_{i,k+1} - \hat{y}_{i,k+1} \right\|^2 \\
&= \left\| \hat{y}_{i,k+1} - y^* \left( \bar{x}_k \right) \right\|^2 + \left\| y^* \left( \bar{x}_{k+1} \right) - y^* \left( \bar{x}_k \right) \right\|^2 \\
&\quad - 2 \left\langle \hat{y}_{i,k+1} - y^* \left( \bar{x}_k \right), y^* \left( \bar{x}_{k+1} \right) - y^* \left( \bar{x}_k \right) \right\rangle \\
&= \left\| \hat{y}_{i,k+1} - y^* \left( \bar{x}_k \right) \right\|^2 + \left\| y^* \left( \bar{x}_{k+1} \right) - y^* \left( \bar{x}_k \right) \right\|^2 \\
&\quad - 2 \left( \hat{y}_{i,k+1} - y^* \left( \bar{x}_k \right) \right)^T \nabla y^* \left( \bar{x}_k \right) \left( \bar{x}_{k+1} - \bar{x}_k \right)^T \\
&\quad - 2 \left( \hat{y}_{i,k+1} - y^* \left( \bar{x}_k \right) \right)^T \left( y^* \left( \bar{x}_{k+1} \right) - y^* \left( \bar{x}_k \right) - \nabla y^* \left( \bar{x}_k \right) \left( \bar{x}_{k+1} - \bar{x}_k \right)^T \right) .
\end{aligned}
\tag{44}
$$

Then, for the first inner-product term on the RHS, letting $\nabla_x \tilde{F}_k = \nabla_x F\left(\mathbf{x}_k, \mathbf{y}_k; \xi_k\right) - \nabla_x F\left(\mathbf{x}_k, \mathbf{y}_k\right)$, we get

$$
\begin{aligned}
& -2\left(\hat{y}_{i,k+1} - y^*\left(\bar{x}_k\right)\right)^T \nabla y^*\left(\bar{x}_k\right)\left(\bar{x}_{k+1} - \bar{x}_k\right)^T \\
& = 2\gamma_x\left(\hat{y}_{i,k+1} - y^*\left(\bar{x}_k\right)\right)^T \nabla y^*\left(\bar{x}_k\right)\left(\nabla_x F\left(\mathbf{x}_k, \mathbf{y}_k\right)\right)^T \left(\frac{\mathbf{1}\bar{v}_{k+1}^{-\alpha}}{n} + \frac{\tilde{\boldsymbol{v}}_{k+1}^{-\alpha}}{n}\right) \\
& \quad + 2\gamma_x\left(\hat{y}_{i,k+1} - y^*\left(\bar{x}_k\right)\right)^T \nabla y^*\left(\bar{x}_k\right)\left(\nabla_x \tilde{F}_k\right)^T \left(\frac{\mathbf{1}\bar{v}_{k+1}^{-\alpha}}{n} + \frac{\tilde{\boldsymbol{v}}_{k+1}^{-\alpha}}{n}\right) \\
& \leqslant 2\gamma_x \kappa \left\|\hat{y}_{i,k+1} - y^*\left(\bar{x}_k\right)\right\| \left\|\left(\nabla_x F\left(\mathbf{x}_k, \mathbf{y}_k\right)\right)^T \left(\frac{\mathbf{1}\bar{v}_{k+1}^{-\alpha}}{n} + \frac{\tilde{\boldsymbol{v}}_{k+1}^{-\alpha}}{n}\right)\right\| \\
& \quad + 2\gamma_x\left(\hat{y}_{i,k+1} - y^*\left(\bar{x}_k\right)\right)^T \nabla y^*\left(\bar{x}_k\right)\left(\nabla_x \tilde{F}_k\right)^T \left(\frac{\mathbf{1}\bar{v}_{k+1}^{-\alpha}}{n} + \frac{\tilde{\boldsymbol{v}}_{k+1}^{-\alpha}}{n}\right).
\end{aligned}
\tag{45}
$$

wherein the last inequality we have used the fact that $y^*$ is $\kappa$-Lipschitz. Then, using Young's inequality with parameter $\lambda_k$, we get

$$
\begin{aligned}
& -2\left(\hat{y}_{i,k+1} - y^*\left(\bar{x}_k\right)\right)^T \nabla y^*\left(\bar{x}_k\right)\left(\bar{x}_{k+1} - \bar{x}_k\right)^T \\
& \leqslant \lambda_k \left\|\hat{y}_{i,k+1} - y^*\left(\bar{x}_k\right)\right\|^2 \\
& \quad + \frac{2\gamma_x^2 \bar{v}_{k+1}^{-2\alpha} \kappa^2}{\lambda_k}\left(\left\|\frac{\mathbf{1}^T}{n}\nabla_x F\left(\mathbf{x}_k, \mathbf{y}_k\right)\right\|^2 + \left\|\frac{\left(\tilde{\boldsymbol{v}}_{k+1}^{-\alpha}\right)^T}{n\bar{v}_{k+1}^{-\alpha}}\nabla_x F\left(\mathbf{x}_k, \mathbf{y}_k\right)\right\|^2\right) \\
& \quad + 2\gamma_x\left(\hat{y}_{i,k+1} - y^*\left(\bar{x}_k\right)\right)^T \nabla y^*\left(\bar{x}_k\right)\left(\nabla_x \tilde{F}_k\right)^T \left(\frac{\mathbf{1}\bar{v}_{k+1}^{-\alpha}}{n} + \frac{\tilde{\boldsymbol{v}}_{k+1}^{-\alpha}}{n}\right).
\end{aligned}
\tag{46}
$$

For the second inner-product term on the RHS, noticing that $y^*$ is $\hat{L} = \kappa\left(1+\kappa\right)^2$ smooth given in Lemma 3, we have

$$
\begin{aligned}
& 2\left(\hat{y}_{i,k+1} - y^*\left(\bar{x}_k\right)\right)^T \left(y^*\left(\bar{x}_k\right) - y^*\left(\bar{x}_{k+1}\right) + \nabla y^*\left(\bar{x}_k\right)\left(\bar{x}_{k+1} - \bar{x}_k\right)^T\right) \\
& \leqslant 2\left\|\hat{y}_{i,k+1} - y^*\left(\bar{x}_k\right)\right\| \left\|y^*\left(\bar{x}_k\right) - y^*\left(\bar{x}_{k+1}\right) + \nabla y^*\left(\bar{x}_k\right)\left(\bar{x}_{k+1} - \bar{x}_k\right)\right\|^2 \\
& \leqslant 2\left\|\hat{y}_{i,k+1} - y^*\left(\bar{x}_k\right)\right\| \frac{\hat{L}}{2}\left\|\bar{x}_{k+1} - \bar{x}_k\right\|^2 \\
& \leqslant \gamma_x^2 \hat{L}\left\|\hat{y}_{i,k+1} - y^*\left(\bar{x}_k\right)\right\| \left\|\left(\frac{\bar{v}_{k+1}^{-\alpha}\mathbf{1}^T}{n} + \frac{\left(\tilde{\boldsymbol{v}}_{k+1}^{-\alpha}\right)^T}{n}\right)\nabla_x F\left(\mathbf{x}_k, \mathbf{y}_k; \xi_k^x\right)\right\|^2 \\
& \leqslant \gamma_x^2 \hat{L}\left\|\hat{y}_{i,k+1} - y^*\left(\bar{x}_k\right)\right\| \frac{2\bar{v}_{k+1}^{-2\alpha}\left(1+\zeta_v^2\right)C}{n}\left\|\nabla_x F\left(\mathbf{x}_k, \mathbf{y}_k; \xi_k^x\right)\right\| \\
& \leqslant \tau \gamma_x^2 \bar{v}_{k+1}^{-2\alpha}\left(1+\zeta_v^2\right)C^2 \hat{L}\left\|\hat{y}_{i,k+1} - y^*\left(\bar{x}_k\right)\right\|^2 + \frac{\gamma_x^2 \bar{v}_{k+1}^{-2\alpha}\left(1+\zeta_v^2\right)\hat{L}}{\tau n}\left\|\nabla_x F\left(\mathbf{x}_k, \mathbf{y}_k; \xi_k^x\right)\right\|^2,
\end{aligned}
\tag{47}
$$

wherein the last inequality we have used Young's inequality with parameter $\tau$. Plugging the obtained inequalities into (44), we get

$$
\begin{aligned}
&\left\| y_{i,k+1} - y^* \left( \bar{x}_{k+1} \right) \right\|^2 \\
&\leqslant \left( 1 + \lambda_k + \tau \gamma_x^2 \bar{v}_{k+1}^{-2\alpha} \left( 1 + \zeta_v^2 \right) C^2 \hat{L} \right) \left\| \hat{y}_{i,k+1} - y^* \left( \bar{x}_k \right) \right\|^2 \\
&\quad + \frac{\gamma_x^2 \bar{v}_{k+1}^{-2\alpha} \left( 1 + \zeta_v^2 \right)}{n} \left( 2\kappa^2 + \frac{\hat{L}}{\tau} \right) \left\| \nabla_x F \left( \mathbf{x}_k, \mathbf{y}_k; \xi_k \right) \right\|^2 \\
&\quad + \frac{2\gamma_x^2 \bar{v}_{k+1}^{-2\alpha} \kappa^2}{\lambda_k} \left( \left\| \frac{\mathbf{1}^T}{n} \nabla_x F \left( \mathbf{x}_k, \mathbf{y}_k \right) \right\|^2 + \left\| \frac{\left( \tilde{\boldsymbol{v}}_{k+1}^{-\alpha} \right)^T}{n \bar{v}_{k+1}^{-\alpha}} \nabla_x F \left( \mathbf{x}_k, \mathbf{y}_k \right) \right\|^2 \right) \\
&\quad + 2\gamma_x \left( \hat{y}_{i,k+1} - y^* \left( \bar{x}_k \right) \right)^T \nabla y^* \left( \bar{x}_k \right) \left( \nabla_x \tilde{F} \right)^T \left( \frac{\mathbf{1} \bar{v}_{k+1}^{-\alpha}}{n} + \frac{\tilde{\boldsymbol{v}}_{k+1}^{-\alpha}}{n} \right).
\end{aligned}
\tag{48}
$$

Set the parameters for Young's inequalities we used as follows,

$$
\lambda_k = \frac{\mu \gamma_y \bar{u}_{k+1}^{-\beta}}{4}, \quad \tau = \frac{\mu \gamma_y \bar{v}_0^{2\alpha - \beta}}{4 \gamma_x^2 \left( 1 + \zeta_v^2 \right) C^2 \hat{L}},
\tag{49}
$$

we get

$$
\begin{aligned}
&\left\| y_{i,k+1} - y^* \left( \bar{x}_{k+1} \right) \right\|^2 \\
&\leqslant \left( 1 + \frac{\mu \gamma_y \bar{u}_{k+1}^{-\beta}}{2} \right) \left\| \hat{y}_{i,k+1} - y^* \left( \bar{x}_k \right) \right\|^2 \\
&\quad + \frac{\gamma_x^2 \left( 1 + \zeta_v^2 \right)}{n} \left( 2\kappa^2 + \frac{4\gamma_x^2 \left( 1 + \zeta_v^2 \right) C^2 \hat{L}^2}{\mu \gamma_y \bar{v}_0^{2\alpha - \beta}} \right) \bar{v}_{k+1}^{-2\alpha} \left\| \nabla_x F \left( \mathbf{x}_k, \mathbf{y}_k; \xi_k \right) \right\|^2 \\
&\quad + \frac{8\gamma_x^2 \bar{v}_{k+1}^{-2\alpha} \kappa^2}{\mu \gamma_y \bar{u}_{k+1}^{-\beta}} \left( \left\| \frac{\mathbf{1}^T}{n} \nabla_x F \left( \mathbf{x}_k, \mathbf{y}_k \right) \right\|^2 + \left\| \frac{\left( \tilde{\boldsymbol{v}}_{k+1}^{-\alpha} \right)^T}{n \bar{v}_{k+1}^{-\alpha}} \nabla_x F \left( \mathbf{x}_k, \mathbf{y}_k \right) \right\|^2 \right) \\
&\quad + 2\gamma_x \left( \hat{y}_{i,k+1} - y^* \left( \bar{x}_k \right) \right)^T \nabla y^* \left( \bar{x}_k \right) \left( \nabla_x \tilde{F}_k \right)^T \left( \frac{\mathbf{1} \bar{v}_{k+1}^{-\alpha}}{n} + \frac{\tilde{\boldsymbol{v}}_{k+1}^{-\alpha}}{n} \right).
\end{aligned}
\tag{50}
$$

Recalling that

$$
\begin{aligned}
&\frac{1}{n} \sum_{i=1}^n \mathbb{E} \left[ \frac{1}{\gamma_y \bar{u}_{k+1}^{-\beta}} \left\| \hat{y}_{i,k+1} - \bar{y}^* \left( \bar{x}_k \right) \right\|^2 \right] \\
&\leqslant \frac{1}{n} \sum_{i=1}^n \mathbb{E} \left[ \frac{1 - 3\mu \gamma_y \bar{u}_{k+1}^{-\beta}/4}{\gamma_y \bar{u}_{k+1}^{-\beta}} \left\| y_{i,k} - \bar{y}^* \left( \bar{x}_k \right) \right\|^2 \right] + \frac{8\kappa L}{n} \mathbb{E} \left[ \left\| \mathbf{x}_k - \mathbf{1} \bar{y}_k \right\|^2 \right] \\
&\quad + \frac{2\gamma_y \left( 1 + \zeta_u^2 \right)}{n} \mathbb{E} \left[ \bar{u}_{k+1}^{-\beta} \left\| \nabla_y F \left( \mathbf{x}_k, \mathbf{y}_k; \xi_k \right) \right\|^2 \right] - \mathbb{E} \left[ 2 \left( f \left( \bar{x}_k, \bar{y}^* \left( \bar{x}_k \right) \right) - f \left( \bar{x}_k, \bar{y}_k \right) \right) \right] \\
&\quad + \frac{8}{\mu} \mathbb{E} \left[ \left\| \frac{\tilde{\boldsymbol{u}}_{k+1}^{-\beta}}{n \bar{u}_{k+1}^{-\beta}} \nabla_y F \left( \mathbf{x}_k, \mathbf{y}_k; \xi_k^y \right) \right\|^2 \right] + 2C \mathbb{E} \left[ \sqrt{\frac{1}{n} \left\| \mathbf{y}_k - \mathbf{1} \bar{y}_k \right\|^2} \right],
\end{aligned}
$$

multiplying by $\frac{2}{\left(2+\mu\gamma_y\bar{u}_{k+1}^{-\beta}\right)\gamma_y\bar{u}_{k+1}^{-\beta}}$ on both sides of (50), we obtain that

$$
\begin{aligned}
&\mathbb{E}\left[f\left(\bar{x}_k,\bar{y}^*\left(\bar{x}_k\right)\right)-f\left(\bar{x}_k,\bar{y}_k\right)\right]\\
&\leqslant \mathbb{E}\left[S_{1,k}\right]+\frac{\gamma_y\left(1+\zeta_u^2\right)}{n}\mathbb{E}\left[\bar{u}_{k+1}^{-\beta}\left\|\nabla_y F\left(\mathbf{x}_k,\mathbf{y}_k;\xi_k\right)\right\|^2\right]+\frac{4\kappa L}{n}\mathbb{E}\left[\left\|\mathbf{x}_k-\mathbf{1}\bar{y}_k\right\|^2\right]\\
&+\frac{4}{\mu}\mathbb{E}\left[\left\|\frac{\tilde{\boldsymbol{u}}_{k+1}^{-\beta}}{n\bar{u}_{k+1}^{-\beta}}\nabla_y F\left(\mathbf{x}_k,\mathbf{y}_k;\xi_k^y\right)\right\|^2\right]+C\mathbb{E}\left[\sqrt{\frac{1}{n}\left\|\mathbf{y}_k-\mathbf{1}\bar{y}_k\right\|^2}\right]\\
&+\underbrace{\mathbb{E}\left[\frac{4\gamma_x^2\bar{v}_{k+1}^{-2\alpha}\kappa^2}{\mu\gamma_y^2\bar{u}_{k+1}^{-2\beta}}\left(\left\|\frac{\mathbf{1}^T}{n}\nabla_x F\left(\mathbf{x}_k,\mathbf{y}_k\right)\right\|^2+\left\|\frac{\left(\tilde{\boldsymbol{v}}_{k+1}^{-\alpha}\right)^T}{n\bar{v}_{k+1}^{-\alpha}}\nabla_x F\left(\mathbf{x}_k,\mathbf{y}_k\right)\right\|^2\right)\right]}_{\mathbb{E}[S_{2,k}]}\\
&+\underbrace{\frac{\gamma_x^2\left(1+\zeta_v^2\right)}{n}\left(\kappa^2+\frac{2\gamma_x^2\left(1+\zeta_v^2\right)C^2\hat{L}^2}{\mu\gamma_y\bar{v}_0^{2\alpha-\beta}}\right)\mathbb{E}\left[\frac{\bar{v}_{k+1}^{-2\alpha}}{\gamma_y\bar{u}_{k+1}^{-\beta}}\left\|\nabla_x F\left(\mathbf{x}_k,\mathbf{y}_k;\xi_k\right)\right\|^2\right]}_{\mathbb{E}[S_{3,k}]}\\
&+\underbrace{\frac{1}{n}\sum_{i=1}^{n}\mathbb{E}\left[\frac{\gamma_x}{\gamma_y\bar{u}_{k+1}^{-\beta}}\left(\hat{y}_{i,k+1}-y^*\left(\bar{x}_k\right)\right)^T\nabla y^*\left(\bar{x}_k\right)\left(\nabla_x\tilde{F}_k\right)^T\left(\frac{\mathbf{1}\bar{v}_{k+1}^{-\alpha}}{n}+\frac{\tilde{\boldsymbol{v}}_{k+1}^{-\alpha}}{n}\right)\right]}_{\mathbb{E}[S_{4,k}]}.
\end{aligned}
\tag{51}
$$

Telescoping the terms, we get

$$
\begin{aligned}
&\sum_{k=k_0}^{K-1}\mathbb{E}\left[f\left(\bar{x}_k,\bar{y}^*\left(\bar{x}_k\right)\right)-f\left(\bar{x}_k,\bar{y}_k\right)\right]\\
&\leqslant \sum_{k=k_0}^{K-1}\mathbb{E}\left[S_{1,k}\right]+\sum_{k=k_0}^{K-1}\mathbb{E}\left[S_{2,k}\right]+\sum_{k=k_0}^{K-1}\mathbb{E}\left[S_{3,k}\right]+\sum_{k=k_0}^{K-1}\mathbb{E}\left[S_{4,k}\right]\\
&+\frac{\gamma_y\left(1+\zeta_u^2\right)}{n}\mathbb{E}\left[\bar{u}_{k+1}^{-\beta}\left\|\nabla_y F\left(\mathbf{x}_k,\mathbf{y}_k;\xi_k\right)\right\|^2\right]+\frac{4\kappa L}{n}\sum_{k=k_0}^{K-1}\mathbb{E}\left[\left\|\mathbf{x}_k-\mathbf{1}\bar{y}_k\right\|^2\right]\\
&+\frac{4}{\mu}\mathbb{E}\left[\left\|\frac{\tilde{\boldsymbol{u}}_{k+1}^{-\beta}}{n\bar{u}_{k+1}^{-\beta}}\nabla_y F\left(\mathbf{x}_k,\mathbf{y}_k;\xi_k^y\right)\right\|^2\right]+C\sum_{k=k_0}^{K-1}\mathbb{E}\left[\sqrt{\frac{1}{n}\left\|\mathbf{y}_k-\mathbf{1}\bar{y}_k\right\|^2}\right].
\end{aligned}
\tag{52}
$$

Next we need to further bound the sums of term $\mathbb{E}\left[S_{2,k}\right]$, $\mathbb{E}\left[S_{3,k}\right]$ and $\mathbb{E}\left[S_{4,k}\right]$ respectively.

$$
\begin{aligned}
&\sum_{k=k_0}^{K-1}\mathbb{E}\left[S_{2,k}\right]\\
&\leqslant \sum_{k=k_0}^{K-1}\mathbb{E}\left[\frac{4\gamma_x^2\bar{v}_{k+1}^{-2\alpha}\kappa^2}{\mu\gamma_y^2\bar{u}_{k+1}^{-2\beta}}\left(\left\|\frac{\mathbf{1}^T}{n}\nabla_x F\left(\mathbf{x}_k,\mathbf{y}_k\right)\right\|^2+\left\|\frac{\left(\tilde{\boldsymbol{v}}_{k+1}^{-\alpha}\right)^T}{n\bar{v}_{k+1}^{-\alpha}}\nabla_x F\left(\mathbf{x}_k,\mathbf{y}_k\right)\right\|^2\right)\right]\\
&\leqslant \frac{8\gamma_x^2\kappa^2\left(1+\zeta_v^2\right)}{\mu\gamma_y^2 G^{2\alpha-2\beta}}\sum_{k=k_0}^{K-1}\mathbb{E}\left[\left\|\nabla_x f\left(\bar{x}_k,\bar{y}_k\right)\right\|^2+\frac{L^2}{n}\Delta_k\right];
\end{aligned}
\tag{53}
$$

then

$$
\sum_{k=k_0}^{K-1} \mathbb{E}\left[S_{3,k}\right]
$$

$$
\leqslant \sum_{k=k_0}^{K-1} \mathbb{E}\left[\frac{\gamma_x^2\left(1+\zeta_v^2\right)}{n\gamma_y}\left(\kappa^2+\frac{2\gamma_x^2\left(1+\zeta_v^2\right)C^2\hat{L}^2}{\mu\gamma_y\bar{v}_1^{2\alpha-\beta}}\right)\frac{\bar{v}_{k+1}^{-2\alpha}}{\bar{u}_{k+1}^{-\beta}}\left\|\nabla_x F\left(\mathbf{x}_k,\mathbf{y}_k;\xi_k^x\right)\right\|^2\right] \tag{54}
$$

$$
\leqslant \frac{\gamma_x^2\left(1+\zeta_v^2\right)}{\gamma_y\bar{v}_1^{\alpha-\beta}}\left(\kappa^2+\frac{2\gamma_x^2\left(1+\zeta_v^2\right)C^2\hat{L}^2}{\mu\gamma_y\bar{v}_1^{2\alpha-\beta}}\right)\sum_{k=k_0}^{K-1}\mathbb{E}\left[\frac{\bar{v}_{k+1}^{-\alpha}}{n}\left\|\nabla_x F\left(\mathbf{x}_k,\mathbf{y}_k;\xi_k^x\right)\right\|^2\right];
$$

for the term $S_{4,k}$, we denote

$$
e_k := \frac{\gamma_x}{\gamma_y\bar{u}_{k+1}^{-\beta}}\left(\frac{1}{n}\sum_{i=1}^n\left(\hat{y}_{i,k+1}-y^*\left(\bar{x}_k\right)\right)^T\right)\nabla y^*\left(\bar{x}_k\right)\left(\nabla_x\tilde{F}_k\right)^T\left(\frac{\mathbf{1}}{n}+\frac{\tilde{\boldsymbol{v}}_{k+1}^{-\alpha}}{n\bar{v}_{k+1}^{-\alpha}}\right),
$$

then we have

$$
|e_k| \leqslant \frac{\gamma_x\kappa}{\gamma_y\bar{u}_{k+1}^{-\beta}}\frac{1}{n}\sum_{i=1}^n\left\|\hat{y}_{i,k+1}-y^*\left(\bar{x}_k\right)\right\|\left\|\left(\nabla_x\tilde{F}_k\right)^T\left(\frac{\mathbf{1}}{n}+\frac{\tilde{\boldsymbol{v}}_{k+1}^{-\alpha}}{n\bar{v}_{k+1}^{-\alpha}}\right)\right\|
$$

$$
\leqslant \frac{\gamma_x\kappa\left(1+\zeta_v\right)}{\gamma_y\sqrt{n}\bar{u}_{k+1}^{-\beta}}\left(\frac{1}{n}\sum_{i=1}^n\frac{1}{\mu}\left\|\nabla_y f\left(\bar{x}_k,\hat{y}_{i,k+1}\right)\right\|\right)\left\|\nabla_x\tilde{F}\right\| \leqslant \underbrace{\frac{\gamma_x\kappa\left(1+\zeta_v\right)C^2\bar{u}_K^\beta}{\mu\gamma_y}}_{M}, \tag{55}
$$

where we have used the Lipschitz continuity of $y^*$ given in Lemma 3 and Assumption 4. Then, noticing that $\mathbb{E}\left[\nabla_x\tilde{F}_k\right]=0$, we obtain

$$
\sum_{k=k_0}^{K-1}\mathbb{E}\left[S_{4,k}\right]=\sum_{k=k_0}^{K-1}\mathbb{E}\left[e_k\bar{v}_{k+1}^{-\alpha}\right]
$$

$$
=\mathbb{E}\left[e_{k_0}\bar{v}_{k_0+1}^{-\alpha}\right]+\underbrace{\sum_{k=k_0+1}^{K-1}\mathbb{E}\left[e_k\bar{v}_k^{-\alpha}\right]}_{0}+\sum_{k=k_0+1}^{K-1}\mathbb{E}\left[-e_k\underbrace{\left(\bar{v}_k^{-\alpha}-\bar{v}_{k+1}^{-\alpha}\right)}_{>0}\right] \tag{56}
$$

$$
\leqslant \mathbb{E}\left[M\bar{v}_{k_0+1}^{-\alpha}\right]+\sum_{k=k_0+1}^{K-1}\mathbb{E}\left[M\left(\bar{v}_k^{-\alpha}-\bar{v}_{k+1}^{-\alpha}\right)\right]
$$

$$
\leqslant 2\mathbb{E}\left[M\bar{v}_{k_0+1}^{-\alpha}\right]\leqslant\frac{2\gamma_x\kappa\left(1+\zeta_v\right)C^2}{\mu\gamma_y\bar{v}_1^\alpha}\mathbb{E}\left[\bar{u}_K^\beta\right].
$$

Therefore combining the obtained inequalities, we complete the proof. $\qquad\square$

Now, it remains to bound term $S_{1,k}$.

**Lemma 10.** *Suppose Assumption 1-5 hold. We have*

$$
\sum_{k=0}^{K-1}\mathbb{E}\left[S_{1,k}\right]\leqslant\frac{1}{2\gamma_y\bar{u}_1^{-\beta}n}\left\|\mathbf{y}_0-\mathbf{1}y^*\left(\bar{x}_0\right)\right\|^2+\frac{\left(4\beta C^2\right)^{2+\frac{1}{1-\beta}}}{2\mu^{3+\frac{1}{1-\beta}}\gamma_y^{2+\frac{1}{1-\beta}}\bar{u}_1^{2-2\beta}}. \tag{57}
$$

*Proof.* Firstly, we have

$$\sum_{k=0}^{K-1} \mathbb{E} \left[ \frac{1 - 3\mu\gamma_y \bar{u}_{k+1}^{-\beta}/4}{2\gamma_y \bar{u}_{k+1}^{-\beta} n} \|\mathbf{y}_k - \mathbf{1}y^* (\bar{x}_k)\|^2 - \frac{\|\mathbf{y}_{k+1} - \mathbf{1}y^* (\bar{x}_{k+1})\|^2}{\left(2 + \mu\gamma_y \bar{u}_{k+1}^{-\beta}\right) \gamma_y \bar{u}_{k+1}^{-\beta} n} \right]$$

$$\leqslant \frac{1 - 3\mu\gamma_y \bar{u}_1^{-\beta}/4}{2\gamma_y \bar{u}_1^{-\beta} n} \|\mathbf{y}_0 - \mathbf{1}y^* (\bar{x}_0)\|^2$$

$$+ \sum_{k=1}^{K-1} \mathbb{E} \left[ \left( \frac{1 - 3\mu\gamma_y \bar{u}_{k+1}^{-\beta}/4}{2\gamma_y \bar{u}_{k+1}^{-\beta} n} - \frac{1}{2n\gamma_y \bar{u}_k^{-\beta} \left(2 + \mu\gamma_y \bar{u}_k^{-\beta}\right)} \right) \|\mathbf{y}_k - \mathbf{1}y^* (\bar{x}_k)\|^2 \right] \tag{58}$$

$$\leqslant \frac{1 - 3\mu\gamma_y \bar{u}_1^{-\beta}/4}{2\gamma_y \bar{u}_1^{-\beta} n} \|\mathbf{y}_0 - \mathbf{1}y^* (\bar{x}_0)\|^2$$

$$+ \sum_{k=1}^{K-1} \mathbb{E} \left[ \left( \frac{1}{2\gamma_y \bar{u}_{k+1}^{-\beta}} - \frac{1}{4\gamma_y \bar{u}_k^{-\beta}} - \frac{\mu}{8} + \underbrace{\frac{\mu}{2 \left(2 + \mu\gamma_y \bar{u}_k^{-\beta}\right)} - \frac{\mu}{2}}_{<0} \right) \frac{1}{n} \|\mathbf{y}_k - \mathbf{1}y^* (\bar{x}_k)\|^2 \right].$$

Next, we show that the term $\frac{1}{2\gamma_y \bar{u}_{k+1}^{-\beta}} - \frac{1}{2\gamma_y \bar{u}_k^{-\beta}} - \frac{\mu}{8}$ is positive for only a constant number of times. If the term is positive, noticing that

$$\frac{\bar{u}_{k+1}^{\beta}}{2\gamma_y} - \frac{\bar{u}_k^{\beta}}{2\gamma_y} - \frac{\mu}{8}$$

$$\leqslant \bar{u}_k^{\beta} \frac{\left(1 + \|\nabla_y F (\mathbf{x}_k, \mathbf{y}_k; \xi_k^y)\|^2 / n\bar{u}_k^{\beta}\right)^{\beta}}{2\gamma_y} - \frac{\bar{u}_k^{\beta}}{2\gamma_y} - \frac{\mu}{8}$$

$$\leqslant \bar{u}_k^{\beta} \frac{\left(1 + \beta \|\nabla_y F (\mathbf{x}_k, \mathbf{y}_k; \xi_k^y)\|^2 / n\bar{u}_k\right)}{2\gamma_y} - \frac{\bar{u}_k^{\beta}}{2\gamma_y} - \frac{\mu}{8} \tag{59}$$

$$= \frac{\beta \|\nabla_y F (\mathbf{x}_k, \mathbf{y}_k; \xi_k^y)\|^2}{2\gamma_y n\bar{u}_k^{1-\beta}} - \frac{\mu}{8},$$

wherein the last inequality we used Bernoulli's inequality. Then we have the following two conditions,

$$\begin{cases} \frac{1}{n} \|\nabla_y F (\mathbf{x}_k, \mathbf{y}_k; \xi_k)\|^2 \geqslant \frac{\gamma_y \bar{u}_{k+1}^{1-\beta}}{4\beta} \geqslant \frac{\gamma_y \bar{u}_1^{1-\beta}}{4\beta}, \\ \frac{4\beta G^2}{\mu\gamma_y} \geqslant \frac{4\beta \|\nabla_y F (\mathbf{x}_k, \mathbf{y}_k; \xi_k^y)\|^2}{\mu\gamma_y n} \geqslant \bar{u}_{k+1}^{1-\beta}, \end{cases} \tag{60}$$

which implies that we have at most the following iteration

$$\left( \frac{4\beta C^2}{\mu\gamma_y} \right)^{\frac{1}{1-\beta}} \frac{4\beta}{\mu\gamma_y \bar{u}_1^{1-\beta}} \tag{61}$$

when the term is positive. Furthermore, when the term is positive, it is also upper bounded,

$$\left( \frac{1}{2\gamma_y \bar{u}_{k+1}^{-\beta}} - \frac{1}{2\gamma_y \bar{u}_k^{-\beta}} - \frac{\mu}{8} \right) \frac{1}{n} \|\mathbf{y}_k - \mathbf{1}y^* (\bar{x}_k)\|^2$$

$$\leqslant \frac{\beta \|\nabla_y F (\mathbf{x}_k, \mathbf{y}_k; \xi_k^y)\|^2}{2\gamma_y n\bar{u}_1^{1-\beta}} \frac{1}{n} \|\mathbf{y}_k - \mathbf{1}y^* (\bar{x}_k)\|^2$$

$$\leqslant \frac{\beta C^2}{2\mu^2 \gamma_y \bar{u}_1^{1-\beta}} \frac{1}{n} \sum_{i=1}^{n} \|\nabla f (\bar{x}_k, y_{i,k})\|^2 \tag{62}$$

$$\leqslant \frac{\beta C^4}{2\mu^2 \gamma_y \bar{u}_1^{1-\beta}},$$

and we have

$$\sum_{k=1}^{K-1} \mathbb{E}\left[\left(\frac{1}{2\gamma_y \bar{u}_{k+1}^{-\beta}} - \frac{1}{2\gamma_y \bar{u}_k^{-\beta}} - \frac{\mu}{8}\right) \frac{1}{n} \|\mathbf{y}_k - \mathbf{1}y^*\left(\bar{x}_k\right)\|^2\right]$$

$$\leqslant \frac{\beta C^4}{2\mu^2 \gamma_y \bar{u}_1^{1-\beta}} \left(\frac{4\beta C^2}{\mu \gamma_y}\right)^{\frac{1}{1-\beta}} \frac{4\beta}{\mu \gamma_y \bar{u}_1^{1-\beta}} \tag{63}$$

$$\leqslant \frac{\left(4\beta C^2\right)^{2+\frac{1}{1-\beta}}}{2\mu^{3+\frac{1}{1-\beta}} \gamma_y^{2+\frac{1}{1-\beta}} \bar{u}_1^{2-2\beta}},$$

which completes the proof. $\qquad\square$

In this part, we bound the term of inconsistency of stepsizes $S_3$, which is crucial for the convergence of the proposed algorithm with stepsize control. The formal lemma is given in Lemma 1 in the main text, we give the proof as below.

### B.3 PROOF OF LEMMA 1

*Proof of Lemma 1.* By the definition of $v_{i,k}$ in (3), we have

$$\mathbb{E}\left[\left\|\frac{\left(\tilde{\boldsymbol{v}}_{k+1}^{-\alpha}\right)^T}{n\bar{v}_{k+1}^{-\alpha}} \nabla_x F\left(\mathbf{x}_k, \mathbf{y}_k; \xi_k^x\right)\right\|^2\right]$$

$$\leqslant \mathbb{E}\left[\frac{1}{n^2} \sum_{i=1}^n \left(\bar{v}_{k+1}^\alpha - v_{i,k+1}^\alpha\right)^2 \frac{\left\|g_{i,k}^x\right\|^2}{v_{i,k+1}^{2\alpha}}\right] \tag{64}$$

$$\leqslant \mathbb{E}\left[\frac{1}{n^2} \sum_{i=1}^n \left(\bar{v}_{k+1}^\alpha - v_{i,k+1}^\alpha\right)^2 \frac{\bar{v}_{k+1}^\alpha}{v_{i,k+1}^{2\alpha}} \frac{\left\|g_{i,k}^x\right\|^2}{\bar{v}_{k+1}^\alpha}\right].$$

Noticing that $\frac{\left|\bar{v}_{k+1}^\alpha - v_{i,k+1}^\alpha\right|}{v_{i,k+1}^\alpha} \leqslant \zeta_v$, we have

$$\mathbb{E}\left[\left\|\frac{\left(\tilde{\boldsymbol{v}}_{k+1}^{-\alpha}\right)^T}{n\bar{v}_{k+1}^{-\alpha}} \nabla_x F\left(\mathbf{x}_k, \mathbf{y}_k; \xi_k^x\right)\right\|^2\right]$$

$$\leqslant \mathbb{E}\left[\frac{1}{n^2} \sum_{i=1}^n \left(\bar{v}_{k+1}^\alpha - v_{i,k+1}^\alpha\right)^2 \left(\frac{\bar{v}_{k+1}^\alpha - v_{i,k+1}^\alpha}{v_{i,k+1}^{2\alpha}} + \frac{1}{v_{i,k+1}^\alpha}\right) \frac{\left\|g_{i,k}^x\right\|^2}{\bar{v}_{k+1}^\alpha}\right]$$

$$\leqslant \mathbb{E}\left[\frac{1}{n^2} \sum_{i=1}^n \frac{\left(\bar{v}_{k+1}^\alpha - v_{i,k+1}^\alpha\right)^2}{v_{i,k+1}^{2\alpha}} \left|\bar{v}_{k+1}^\alpha - v_{i,k+1}^\alpha\right| \frac{\left\|g_{i,k}^x\right\|^2}{\bar{v}_{k+1}^\alpha}\right] \tag{65}$$

$$+ \mathbb{E}\left[\frac{1}{n^2} \sum_{i=1}^n \frac{\left|\bar{v}_{k+1}^\alpha - v_{i,k+1}^\alpha\right|}{v_{i,k+1}^\alpha} \left|\bar{v}_{k+1}^\alpha - v_{i,k+1}^\alpha\right| \frac{\left\|g_{i,k}^x\right\|^2}{\bar{v}_{k+1}^\alpha}\right]$$

$$\leqslant \left(1 + \zeta_v\right) \zeta_v \mathbb{E}\left[\frac{1}{n} \sum_{i=1}^n \left|\bar{v}_{k+1}^\alpha - v_{i,k+1}^\alpha\right| \frac{1}{n} \sum_{i=1}^n \frac{\left\|g_{i,k}^x\right\|^2}{\bar{v}_{k+1}^\alpha}\right].$$

By Lemma 6, we get

$$\frac{1}{K} \sum_{k=0}^{K-1} \mathbb{E}\left[\left\|\frac{\left(\tilde{\boldsymbol{v}}_{k+1}^{-\alpha}\right)^T}{n\bar{v}_{k+1}^{-\alpha}} \nabla_x F\left(\mathbf{x}_k, \mathbf{y}_k; \xi_k^x\right)\right\|^2\right]$$

$$\leqslant \left(1+\zeta_v\right) \zeta_v \mathbb{E}\left[\frac{1}{n} \sum_{i=1}^n \left|\bar{v}_{k+1}^\alpha - v_{i,k+1}^\alpha\right| \frac{1}{K} \sum_{k=0}^{K-1} \frac{\frac{1}{n} \sum_{i=1}^n \left\|g_{i,k}^x\right\|^2}{\bar{v}_{k+1}^\alpha}\right]$$ (66)

$$\leqslant \left(1+\zeta_v\right) \zeta_v \mathbb{E}\left[\frac{1}{n} \sum_{i=1}^n \left|\bar{v}_{k+1}^\alpha - v_{i,k+1}^\alpha\right|\right] \frac{C^{2-2\alpha}}{(1-\alpha) K^\alpha}$$

$$\leqslant \left(1+\zeta_v\right) \zeta_v \sqrt{\frac{1}{n} \mathbb{E}\left[\left\|\boldsymbol{v}_{k+1} - \mathbf{1}\bar{v}_{k+1}\right\|^{2\alpha}\right]} \frac{C^{2-2\alpha}}{(1-\alpha) K^\alpha}.$$

Next, for the term of inconsistency of the stepsize $\left\|\boldsymbol{v}_k - \mathbf{1}\bar{v}_k\right\|^2$, we consider two cases since the max operator we used. At iteration $k$, for the case $\mathbf{m}_k^x \geqslant \mathbf{m}_k^y$, we have

$$\mathbb{E}\left[\left\|\boldsymbol{v}_{k+1} - \mathbf{1}\bar{v}_{k+1}\right\|^2\right] = \mathbb{E}\left[\left\|\mathbf{m}_{k+1}^x - \mathbf{1}\bar{m}_{k+1}^x\right\|^2\right]$$

$$= \mathbb{E}\left[\left\|(W-\mathbf{J})\left(\mathbf{m}_k^x - \mathbf{1}\bar{m}_k^x\right) + \eta_k (W-\mathbf{J}) \boldsymbol{h}_k^x\right\|^2\right]$$

$$\leqslant \frac{1+\rho_W}{2} \mathbb{E}\left[\left\|\mathbf{m}_k^x - \mathbf{1}\bar{m}_k^x\right\|^2\right] + \frac{(1+\rho_W)\rho_W}{1-\rho_W} \mathbb{E}\left[\left\|\boldsymbol{h}_k^x\right\|^2\right]$$ (67)

$$\leqslant \left(\frac{1+\rho_W}{2}\right)^k \mathbb{E}\left[\left\|\mathbf{m}_0^x - \mathbf{1}\bar{m}_0^x\right\|^2\right] + \frac{nC^2(1+\rho_W)\rho_W}{1-\rho_W} \sum_{t=0}^k \left(\frac{1+\rho_W}{2}\right)^{k-t}$$

$$\leqslant \frac{2nC^2(1+\rho_W)\rho_W}{(1-\rho_W)^2},$$

where we set $\left\|\mathbf{m}_0^x - \mathbf{1}\bar{m}_0^x\right\|^2 = 0$; for the case $\mathbf{m}_k^x < \mathbf{m}_k^y$, with $\left\|\mathbf{m}_0^y - \mathbf{1}\bar{m}_0^y\right\|^2 = 0$,

$$\mathbb{E}\left[\left\|\boldsymbol{v}_{k+1} - \mathbf{1}\bar{v}_{k+1}\right\|^2\right] = \mathbb{E}\left[\left\|\mathbf{m}_{k+1}^y - \mathbf{1}\bar{m}_{k+1}^y\right\|^2\right] \leqslant \frac{2nC^2(1+\rho_W)\rho_W}{(1-\rho_W)^2},$$ (68)

Combining these two cases, and using Lemma 6 and the fact $\left\|\boldsymbol{v}_k^\alpha - \mathbf{1}\bar{v}_k^\alpha\right\|^2 \leqslant \left\|\boldsymbol{v}_k - \mathbf{1}\bar{v}_k\right\|^{2\alpha}$ for $\alpha \in (0,1)$, we complete the proof. $\qquad\square$

We further give the following lemma to show that the inconsistency of stepsize remains uniformly bounded for the vanilla D-TiAda algorithm (2).

**Lemma 11** (Inconsistency for D-TiAda). *Suppose Assumption 1-5 hold. For D-TiAda, we have*

$$\frac{1}{K} \sum_{k=0}^{K-1} \mathbb{E}\left[\left\|\frac{\left(\tilde{\boldsymbol{v}}_{k+1}^{-\beta}\right)^T}{n\bar{v}_{k+1}^{-\beta}} \nabla_x F\left(\mathbf{x}_k, \mathbf{y}_k; \xi_k^x\right)\right\|^2\right] \leqslant \zeta_v^2 C^2,$$

$$\frac{1}{K} \sum_{k=0}^{K-1} \mathbb{E}\left[\left\|\frac{\left(\tilde{\boldsymbol{u}}_{k+1}^{-\beta}\right)^T}{n\bar{u}_{k+1}^{-\beta}} \nabla_y F\left(\mathbf{x}_k, \mathbf{y}_k; \xi_k^y\right)\right\|^2\right] \leqslant \zeta_u^2 C^2.$$ (69)

*Proof.* By the definition of inconsistency of stepsizes in (7) and Assumption 4 on bounded gradient, we immediately get the result. $\qquad\square$

### B.4 Proof of Theorem 1

*Proof of Theorem 1.* Consider a complete graph with 3 nodes where the functions corresponding to the nodes are:

$$f_1(x, y) = -\frac{1}{2}y^2 + xy - \frac{1}{2}x^2,$$

$$f_2(x, y) = f_3(x, y) = -\frac{1}{2}y^2 - (1 + \frac{1}{a} + \frac{1}{b})xy - \frac{1}{2}x^2,$$

where

$$a = 2^{\frac{-1}{2\alpha-1}} \quad \text{and} \quad b = 2^{\frac{-1}{2\beta-1}}.$$

Notice that the only stationary point of $f(x, y) = (f_1(x, y) + f_2(x, y) + f_3(x, y))/3$ is $(0, 0)$. We denote $g_{i,k}^x = \nabla_x f_i(x_k, y_k)$ and $g_{i,k}^y = \nabla_y f_i(x_k, y_k)$.

Now we consider points initialized in line

$$y = -\frac{1+a}{a+\frac{a}{b}}x, \tag{70}$$

where we have

$$g_{1,0}^x = y_0 - x_0 = -\frac{2ab+a+b}{ab+a}x_0$$

$$g_{2,0}^x = g_{3,0}^x = -\left(1 + \frac{1}{b} + \frac{1}{a}\right)y_0 - x_0 = \frac{2ab+a+b}{a^2(b+1)}x_0$$

$$g_{1,0}^y = x_0 - y_0 = \frac{2ab+a+b}{ab+a}x_0$$

$$g_{2,0}^y = g_{2,0}^y = -\frac{2ab+a+b}{ab(b+1)}x_0.$$

Note that by our assumptions of the range of $\alpha$ and $\beta$, we have $a < b$, so we have

$$|g_{1,0}^x| = |g_{1,0}^y| \quad \text{and} \quad |g_{2,0}^x| > |g_{2,0}^y|,$$

which means gradient of $x$ would be chosen in the maximum operator in the denominator of TiAda stepsize for $x$. Therefore, after one step, we have

$$x_1 = x_0 - \eta^x \underbrace{\left(\frac{g_{1,0}^x}{\left(|g_{1,0}^x|^2\right)^\alpha} + \frac{g_{2,0}^x}{\left(|g_{2,0}^x|^2\right)^\alpha} + \frac{g_{3,0}^x}{\left(|g_{3,0}^x|^2\right)^\alpha}\right)}_{=0}$$

$$y_1 = y_0 - \eta^y \underbrace{\left(\frac{g_{1,0}^y}{\left(|g_{1,0}^y|^2\right)^\beta} + \frac{g_{2,0}^y}{\left(|g_{2,0}^y|^2\right)^\beta} + \frac{g_{3,0}^y}{\left(|g_{3,0}^y|^2\right)^\beta}\right)}_{=0}.$$

Next, we will use induction to show that $x$ and $y$ will stay in $x_0$ and $y_0$ for any iteration. Assume for all iterations $k$ in $1, \dots, t$, $x_k = x_0$ and $y_k = y_0$, then we have in next step

$$x_{t+1} = x_t - \eta^x \left(\frac{g_{1,0}^x}{\left(t \cdot |g_{1,0}^x|^2\right)^\alpha} + \frac{g_{2,0}^x}{\left(t \cdot |g_{2,0}^x|^2\right)^\alpha} + \frac{g_{3,0}^x}{\left(t \cdot |g_{3,0}^x|^2\right)^\alpha}\right).$$

Note that $g_{1,0}^x = -a \cdot g_{2,0}^x$, then we get

$$x_{t+1} = x_t - \eta^x \left(\frac{-p \cdot g_{2,0}^x}{t^\alpha \cdot a^{2\alpha} \cdot |g_{2,0}^x|^{2\alpha}} + \frac{2g_{2,0}^x}{t^\alpha \cdot |g_{2,0}^x|^{2\alpha}}\right)$$

$$= x_t - \frac{g_{2,0}^x}{t^\alpha \cdot |g_{2,0}^x|^{2\alpha}} \underbrace{\left(2 - a^{1-2\alpha}\right)}_{=0 \text{ (by definition of } a)}$$

$$= x_t.$$

Similarly, we can show that $y_{t+1} = y_t$. Therefore all iterates will stay at $(x_0, y_0)$ if initialized at line $y = -\frac{ab+b}{ab+a}x$. The initial gradient norm can be arbitrarily large by picking $x_0$ to be large. $\square$

## B.5 PROOF OF THEOREM 2

*Proof of Theorem 2.* Combining the results in Lemma 8, 9 and 10, we get

$$
\sum_{k=0}^{K-1} \mathbb{E}\left[f\left(\bar{x}_k, y^*\left(\bar{x}_k\right)\right) - f\left(\bar{x}_k, \bar{y}_k\right)\right]
$$

$$
= \sum_{k=0}^{k_0-1} \mathbb{E}\left[f\left(\bar{x}_k, y^*\left(\bar{x}_k\right)\right) - f\left(\bar{x}_k, \bar{y}_k\right)\right] + \sum_{k=k_0}^{K-1} \mathbb{E}\left[f\left(\bar{x}_k, y^*\left(\bar{x}_k\right)\right) - f\left(\bar{x}_k, \bar{y}_k\right)\right]
$$

$$
\leqslant \frac{1}{2\gamma_y \bar{u}_1^{-\beta} n} \mathbb{E}\left[\left\|\mathbf{y}_0 - \mathbf{1}y^*\left(\bar{x}_0\right)\right\|^2\right] + \frac{\left(4\beta C^2\right)^{2+\frac{1}{1-\beta}}}{2\mu^{3+\frac{1}{1-\beta}}\gamma_y^{2+\frac{1}{1-\beta}}\bar{u}_1^{2-2\beta}}
$$

$$
+ \frac{2\gamma_x^2\kappa^2\left(1+\zeta_v^2\right)G^{2\beta}}{n\mu\gamma_y^2}\sum_{k=0}^{k_0-1}\mathbb{E}\left[\bar{v}_{k+1}^{-2\alpha}\left\|\nabla_x F\left(\mathbf{x}_k, \mathbf{y}_k; \xi_k^x\right)\right\|^2\right]
$$

$$
+ \frac{\gamma_y\left(1+\zeta_u^2\right)}{n}\sum_{k=0}^{K-1}\mathbb{E}\left[\bar{u}_{k+1}^{-\beta}\left\|\nabla_y F\left(\mathbf{x}_k, \mathbf{y}_k; \xi_k\right)\right\|^2\right] + C\sum_{k=0}^{K-1}\mathbb{E}\left[\sqrt{\frac{1}{n}\left\|\mathbf{y}_k - \mathbf{1}\bar{y}_k\right\|^2}\right]
$$

$$
+ \frac{4}{\mu}\sum_{k=0}^{K-1}\mathbb{E}\left[\left\|\frac{\tilde{\boldsymbol{u}}_{k+1}^{-\beta}}{n\bar{u}_{k+1}^{-\beta}}\nabla_y F\left(\mathbf{x}_k, \mathbf{y}_k; \xi_k^y\right)\right\|^2\right] + \frac{8\gamma_x^2\kappa^2\left(1+\zeta_v^2\right)}{\mu\gamma_y^2 G^{2\alpha-2\beta}}\sum_{k=k_0}^{K-1}\left\|\nabla_x f\left(\bar{x}_k, \bar{y}_k\right)\right\|^2
$$

$$
+ \left(\frac{8\gamma_x^2\kappa^2 L^2\left(1+\zeta_v^2\right)}{n\mu\gamma_y^2 G^{2\alpha-2\beta}} + \frac{4\kappa L}{n}\right)\sum_{k=0}^{K-1}\mathbb{E}\left[\Delta_k\right] + \frac{2\gamma_x\kappa\left(1+\zeta_v\right)C^2}{\mu\gamma_y\bar{v}_1^{\alpha}}\mathbb{E}\left[\bar{u}_K^{\beta}\right]
$$

$$
+ \frac{\gamma_x^2\left(1+\zeta_v^2\right)}{\gamma_y\bar{v}_1^{\alpha-\beta}}\left(\kappa^2 + \frac{2\gamma_x^2\left(1+\zeta_v^2\right)C^2\hat{L}^2}{\mu\gamma_y\bar{v}_1^{2\alpha-\beta}}\right)\sum_{k=k_0}^{K-1}\mathbb{E}\left[\frac{\bar{v}_{k+1}^{-\alpha}}{n}\left\|\nabla_x F\left(\mathbf{x}_k, \mathbf{y}_k; \xi_k^x\right)\right\|^2\right]. \tag{71}
$$

Letting the dividing point between the two phase in Lemma 8 and 9 satisfy

$$
G = \left(\frac{16\left(1+\zeta_v^2\right)\gamma_x^2\kappa^4}{\gamma_y^2}\right)^{\frac{1}{2\alpha-2\beta}}, \tag{72}
$$

then, plugging above inequality into (18) in Lemma 5, with the help of Lemma 6-10 and Lemma 1, we can get the following result,

$$
\frac{1}{K}\sum_{k=0}^{K-1}\mathbb{E}\left[\left\|\nabla\Phi\left(\bar{x}_k\right)\right\|^2\right]
$$

$$
\leqslant E_0 + E_G + E_W + \frac{8C^{2\alpha}\left(\Phi^{\max} - \Phi^*\right)}{\gamma_x K^{1-\alpha}}
$$

$$
+ \frac{16\gamma_x\kappa^3\left(1+\zeta_v\right)C^{2+2\beta}}{\gamma_y\bar{v}_1^{\alpha}K^{1-\beta}} + \frac{8\kappa L\gamma_y\left(1+\zeta_u^2\right)C^{2-2\beta}}{\left(1-\beta\right)K^{\beta}}
$$

$$
+ \left(\gamma_x L_\Phi + \frac{\kappa^3 L\gamma_x^2}{\gamma_y\bar{v}_1^{\alpha-\beta}} + \frac{2\gamma_x^4\kappa^2\left(1+\zeta_v^2\right)C^2\hat{L}^2}{\gamma_y^2\bar{v}_1^{3\alpha-2\beta}}\right)\frac{8\left(1+\zeta_v^2\right)C^{2-2\alpha}}{\left(1-\alpha\right)K^{\alpha}} \tag{73}
$$

$$
+ \sqrt{\frac{1}{n^{1-\alpha}}\left(\frac{4\rho_W}{\left(1-\rho_W\right)^2}\right)^{\alpha}}\frac{16\left(1+\zeta_v\right)\zeta_v C^{2-\alpha}}{\left(1-\alpha\right)K^{\alpha}}
$$

$$
+ \sqrt{\frac{1}{n^{1-\beta}}\left(\frac{4\rho_W}{\left(1-\rho_W\right)^2}\right)^{\beta}}\frac{32\kappa^2\left(1+\zeta_u\right)\zeta_u C^{2-\beta}}{\left(1-\beta\right)K^{\beta}}
$$

$$
+ 8\kappa LC\sqrt{\frac{8\rho_W\gamma_y^2\left(1+\zeta_u^2\right)}{\left(1-\rho_W\right)^2}\left(\frac{C^{2-4\beta}}{\left(1-2\beta\right)K^{2\beta}}\mathbb{I}_{\beta<1/2} + \frac{1+\log u_K - \log v_1}{K\bar{u}_1^{2\beta-1}}\mathbb{I}_{\beta\geqslant 1/2}\right)},
$$

---

**Algorithm 2 DAS$^2$C with coordinate-wise adaptive stepsize**

---

**Initialization:** $x_{i,0}, y_{i,0} \in \mathbb{R}^p$, buffers $m_{i,0}^x, m_{i,0}^y > 0$, stepsizes $\gamma_x, \gamma_y > 0$ and $0 < \beta < \alpha < 1$.

1: **for** iteration $k = 0, 1, \cdots$, each node $i \in [n]$, **do**
2:     Sample i.i.d $\xi_{i,k}^x$ and $\xi_{i,k}^y$, compute:

$$g_{i,k}^x = \nabla_x f_i \left( x_{i,k}, y_{i,k}; \xi_{i,k}^x \right), g_{i,k}^y = \nabla_y f_i \left( x_{i,k}, y_{i,k}; \xi_{i,k}^y \right).$$

3:     Update local gradient sums with Schur product:

$$m_{i,k+1}^x = m_{i,k}^x + g_{i,k}^x \odot g_{i,k}^x, \; m_{i,k+1}^y = m_{i,k}^y + g_{i,k}^y \odot g_{i,k}^y$$

4:     Compute the ratio between two gradient sums:

$$\psi_{i,k+1} = \left\| m_{i,k+1}^x \right\|^{2\alpha} / \max \left\{ \left\| m_{i,k+1}^x \right\|^{2\alpha}, \left\| m_{i,k+1}^y \right\|^{2\alpha} \right\} \leqslant 1.$$

5:     Update primal and dual variables locally:

$$x_{i,k+1} = x_{i,k} - \gamma_x \psi_{i,k+1} \left( m_{i,k+1}^x \right)^{-\alpha} \odot g_{i,k}^x,$$
$$y_{i,k+1} = y_{i,k} - \gamma_y \left( m_{i,k+1}^y \right)^{-\beta} \odot g_{i,k}^y.$$

6:     Communicate parameters over network:

$$\left\{ m_{i,k+1}^x, m_{i,k+1}^y, x_{i,k+1}, y_{i,k+1}, \right\} \leftarrow \sum_{j \in \mathcal{N}_i} W_{i,j} \left\{ m_{j,k+1}^x, m_{j,k+1}^y, x_{j,k+1}, y_{j,k+1}, \right\}.$$

7:     Projection of dual variable on to set $\mathcal{Y}$: $y_{i,k+1} \leftarrow \mathcal{P}_{\mathcal{Y}} \left( y_{i,k+1} \right)$.
8: **end for**

---

where $\hat{L} = \kappa (1 + \kappa)^2$, $L_\Phi = L (1 + \kappa)$, and

$$E_0 := \frac{4\kappa L}{\gamma_y \bar{u}_1^{-\beta} nK} \mathbb{E} \left[ \|\mathbf{y}_0 - \mathbf{1} y^* (\bar{x}_0)\|^2 \right] + \frac{4\kappa^2 \left( 4\beta C^2 \right)^{2 + \frac{1}{1-\beta}}}{\mu^{2 + \frac{1}{1-\beta}} \gamma_y^{2 + \frac{1}{1-\beta}} \bar{u}_1^{2 - 2\beta} K},$$

$$E_G := \frac{16 \gamma_x^2 \kappa^4 \left( 1 + \zeta_v^2 \right) G^{2\beta}}{\gamma_y^2} \left( \frac{C^{2-4\alpha}}{(1 - 2\alpha) K^{2\alpha}} \mathbb{I}_{\alpha < 1/2} + \frac{1 + \log v_K - \log v_1}{K \bar{v}_1^{2\alpha - 1}} \mathbb{I}_{\alpha \geqslant 1/2} \right),$$

$$E_W := \frac{32 \left( 8\kappa L + 3L^2 \right) \rho_W \gamma_x^2 \left( 1 + \zeta_v^2 \right)}{(1 - \rho_W)^2} \left( \frac{C^{2-4\alpha}}{(1 - 2\alpha) K^{2\alpha}} \mathbb{I}_{\alpha < 1/2} + \frac{1 + \log v_K - \log v_1}{K \bar{v}_1^{2\alpha - 1}} \mathbb{I}_{\alpha \geqslant 1/2} \right)$$

$$+ \frac{32 \left( 8\kappa L + 3L^2 \right) \rho_W \gamma_y^2 \left( 1 + \zeta_u^2 \right)}{(1 - \rho_W)^2} \left( \frac{C^{2-4\beta}}{(1 - 2\beta) K^{2\beta}} \mathbb{I}_{\beta < 1/2} + \frac{1 + \log u_K - \log v_1}{K \bar{u}_1^{2\beta - 1}} \mathbb{I}_{\beta \geqslant 1/2} \right).$$

Let the total iteration $K$ satisfy the conditions in (12) such that the terms $E_G$ and $E_W$ are dominated, we thus complete the proof. $\square$

### B.5.1 PROOF OF COROLLARY 1

*Proof of Corollary 1.* With the help of Lemma 11, we can directly adapt the proof of Theorem 2 to get the result in (14). $\square$

### B.6 EXTEND THE PROOF TO COORDINATE-WISE STEPSIZE

In this subsection, we show how to extend our convergence analysis of DAS$^2$C to the coordinate-wise adaptive stepsize (Zhou et al., 2018) variant. We first present this variant in the Algorithm 2, which can be rewritten in a compact form with the Schur product denoted by $\odot$.

$$\mathbf{m}_{k+1}^x = W\left(\mathbf{m}_k^x + \mathbf{h}_k^x\right), \tag{74a}$$

$$\mathbf{m}_{k+1}^y = W\left(\mathbf{m}_k^y + \mathbf{h}_k^y\right), \tag{74b}$$

$$\mathbf{x}_{k+1} = W\left(\mathbf{x}_k - \gamma_x V_{k+1}^{-\alpha} \odot \nabla_x F\left(\mathbf{x}_k, \mathbf{y}_k; \xi_k\right)\right), \tag{74c}$$

$$\mathbf{y}_{k+1} = \mathcal{P}_{\mathcal{Y}}\left(W\left(\mathbf{y}_k + \gamma_y U_{k+1}^{-\beta} \odot \nabla_y F\left(\mathbf{x}_k, \mathbf{y}_k; \xi_k\right)\right)\right), \tag{74d}$$

where

$$\boldsymbol{h}_k^x = \left[\cdots, g_{i,k}^x \odot g_{i,k}^x, \cdots\right]^T \in \mathbb{R}^{n \times p}, \ \boldsymbol{h}_k^y = \left[\cdots, g_{i,k}^y \odot g_{i,k}^y, \cdots\right]^T \in \mathbb{R}^{n \times d},$$

and the matrices $U_k^{\alpha}$ and $V_k^{\beta}$ are redefined as follows:

$$V_k^{-\alpha} = \left[\cdots, v_{i,k}^{-\alpha}, \cdots\right]^T, \ [v_{i,k}]_j = \max\left\{\left[m_{i,k}^x\right]_j, \left[m_{i,k}^y\right]_j\right\}, \ j \in [p],$$
$$U_k^{-\beta} = \left[\cdots, u_{i,k}^{-\beta}, \cdots\right]^T, \ [u_{i,k}]_j = \left[m_{i,k}^x\right]_j, \ j \in [d] \tag{75}$$

where $[\cdot]_j$ denotes the $j$-th element of a vector.

Recalling the definitions of inconsistency of stepsize in (7), we give the following notations:

$$\tilde{V}_k = V_k - \bar{v}_k \mathbf{1}\mathbf{1}_p^T, \ \bar{v}_k = \frac{1}{np}\sum_{i=1}^n \sum_j^p V_{ij}, \ \bar{v}_{i,k} = \frac{1}{p}\sum_j^p V_{ij}, \ \bar{v}_{j,k} = \frac{1}{n}\sum_{i=1}^n V_{ij},$$
$$\tilde{U}_k = U_k - \bar{u}_k \mathbf{1}\mathbf{1}_p^T, \ \bar{u}_k = \frac{1}{nd}\sum_{i=1}^n \sum_j^d U_{ij}, \ \bar{u}_{i,k} = \frac{1}{d}\sum_j^d U_{ij}, \ \bar{u}_{j,k} = \frac{1}{n}\sum_{i=1}^n U_{ij}, \tag{76}$$

and

$$\zeta_V^2 = \sup_{k \geqslant 0}\left\{\frac{\left\|V_k^{-\alpha} - \bar{v}_k^{-\alpha}\mathbf{1}\mathbf{1}_p^T\right\|^2}{np\left(\bar{v}_k^{-\alpha}\right)^2}\right\}, \ \hat{\zeta}_v^2 = \sup_{k \geqslant 0}\left\{\frac{\left\|V_k^{-\alpha} - (V_k \mathbf{J}_p)^{-\alpha}\right\|^2}{np\left(\bar{v}_k^{-\alpha}\right)^2}\right\},$$

$$\zeta_U^2 = \sup_{k \geqslant 0}\left\{\frac{\left\|U_k^{-\beta} - \bar{u}_k^{-\beta}\mathbf{1}\mathbf{1}_d^T\right\|^2}{nd\left(\bar{u}_k^{-\beta}\right)^2}\right\}, \ \hat{\zeta}_u^2 = \sup_{k \geqslant 0}\left\{\frac{\left\|U_k^{-\beta} - (U_k \mathbf{J}_d)^{-\beta}\right\|^2}{nd\left(\bar{u}_k^{-\beta}\right)^2}\right\}.$$

According to the two definitions of inconsistency of stepsize for Option I and II, we can give the following lemma to show their difference.

**Lemma 12** (Inconsistency, coordinate-wise). *Suppose Assumption 1-5 hold. For the proposed DAS$^2$C algorithm, we have*

$$\frac{1}{K}\sum_{k=0}^{K-1}\mathbb{E}\left[\left\|\frac{\mathbf{1}^T}{n\bar{v}_{k+1}^{-\alpha}}\tilde{V}_{k+1}^{-\alpha} \odot \nabla_x F\left(\mathbf{x}_k, \mathbf{y}_k; \xi_k^x\right)\right\|^2\right]$$
$$\leqslant 2\left(1 + \zeta_v\right)\zeta_v\sqrt{\frac{1}{n^{1-\alpha}}\left(\frac{4C^2\rho_W}{\left(1 - \rho_W\right)^2}\right)^\alpha \frac{C^{2-2\alpha}}{\left(1 - \alpha\right)K^\alpha}} + 2np\hat{\zeta}_v^2 C^2 \tag{77}$$

*and*

$$\frac{1}{K}\sum_{k=0}^{K-1}\mathbb{E}\left[\left\|\frac{\mathbf{1}^T}{n\bar{u}_{k+1}^{-\alpha}}\tilde{U}_{k+1}^{-\alpha} \odot \nabla_y F\left(\mathbf{x}_k, \mathbf{y}_k; \xi_k^y\right)\right\|^2\right]$$
$$\leqslant 2\left(1 + \zeta_u\right)\zeta_u\sqrt{\frac{1}{n^{1-\beta}}\left(\frac{4C^2\rho_W}{\left(1 - \rho_W\right)^2}\right)^\beta \frac{C^{2-2\beta}}{\left(1 - \beta\right)K^\beta}} + 2nd\hat{\zeta}_u^2 C^2. \tag{78}$$

*In contrast, for D-TiAda, we have*

$$\frac{1}{K} \sum_{k=0}^{K-1} \mathbb{E}\left[\left\|\frac{\mathbf{1}^T}{n\bar{v}_{k+1}^{-\alpha}} \tilde{V}_{k+1}^{-\alpha} \odot \nabla_x F\left(\mathbf{x}_k, \mathbf{y}_k; \xi_k^x\right)\right\|^2\right] \leqslant p\zeta_V^2 C^2,$$

$$\frac{1}{K} \sum_{k=0}^{K-1} \mathbb{E}\left[\left\|\frac{\mathbf{1}^T}{n\bar{u}_{k+1}^{-\alpha}} \tilde{U}_{k+1}^{-\beta} \odot \nabla_y F\left(\mathbf{x}_k, \mathbf{y}_k; \xi_k^y\right)\right\|^2\right] \leqslant d\zeta_U^2 C^2. \tag{79}$$

*Proof.* For the coordinate-wise stepsize, with the help of Lemma 4, we have

$$\mathbb{E}\left[\left\|\frac{\mathbf{1}^T}{n\bar{v}_{k+1}^{-\alpha}} \tilde{V}_{k+1}^{-\alpha} \odot \nabla_x F\left(\mathbf{x}_k, \mathbf{y}_k; \xi_k^x\right)\right\|^2\right]$$

$$= \mathbb{E}\left[\left\|\frac{\mathbf{1}^T}{n\bar{v}_{k+1}^{-\alpha}} \left(V_{k+1}^{-\alpha} - (V_{k+1}\mathbf{J})^{-\alpha} + (V_{k+1}\mathbf{J})^{-\alpha} - \bar{v}_{k+1}^{-\alpha}\mathbf{1}\mathbf{1}_p^T\right) \odot \nabla_x F\left(\mathbf{x}_k, \mathbf{y}_k; \xi_k^x\right)\right\|^2\right]$$

$$\leqslant 2\mathbb{E}\left[\left\|\frac{\mathbf{1}^T}{n\bar{v}_{k+1}^{-\alpha}} \left((V_{k+1}\mathbf{J})^{-\alpha} - \bar{v}_{k+1}^{-\alpha}\mathbf{1}\mathbf{1}_p^T\right) \odot \nabla_x F\left(\mathbf{x}_k, \mathbf{y}_k; \xi_k^x\right)\right\|^2\right] \tag{80}$$

$$+ 2\mathbb{E}\left[\left\|\frac{\mathbf{1}^T}{n\bar{v}_{k+1}^{-\alpha}} \left(V_{k+1}^{-\alpha} - (V_{k+1}\mathbf{J})^{-\alpha}\right) \odot \nabla_x F\left(\mathbf{x}_k, \mathbf{y}_k; \xi_k^x\right)\right\|^2\right].$$

For the first term in the last line, by Lemma 4, we have

$$\mathbb{E}\left[\left\|\frac{\mathbf{1}^T}{n\bar{v}_{k+1}^{-\alpha}} \left((V_{k+1}\mathbf{J})^{-\alpha} - \bar{v}_{k+1}^{-\alpha}\mathbf{1}\mathbf{1}_p^T\right) \odot \nabla_x F\left(\mathbf{x}_k, \mathbf{y}_k; \xi_k^x\right)\right\|^2\right]$$

$$\leqslant \mathbb{E}\left[\frac{1}{n^2\bar{v}_{k+1}^{-2\alpha}} \sum_{i=1}^n \left(\bar{v}_{i,k+1}^\alpha - \bar{v}_{k+1}^\alpha\right)^2 \left\|\nabla_x f_i\left(x_{i,k}, y_{i,k}; \xi_{i,k}^x\right)\right\|^2\right]. \tag{81}$$

which is a similar term with Option I and is convergent. Then, for the second part,

$$\mathbb{E}\left[\left\|\frac{\mathbf{1}^T}{n\bar{v}_{k+1}^{-\alpha}} \left(V_{k+1}^{-\alpha} - (V_{k+1}\mathbf{J})^{-\alpha}\right) \odot \nabla_x F\left(\mathbf{x}_k, \mathbf{y}_k; \xi_k^x\right)\right\|^2\right]$$

$$\leqslant \frac{1}{n}\mathbb{E}\left[\left\|\frac{V_{k+1}^{-\alpha} - (V_{k+1}\mathbf{J})^{-\alpha}}{\bar{v}_{k+1}^{-\alpha}}\right\|^2 \|\nabla_x F\left(\mathbf{x}_k, \mathbf{y}_k; \xi_k^x\right)\|^2\right] \tag{82}$$

$$\leqslant p\hat{\zeta}_v^2 \mathbb{E}\left[\|\nabla_x F\left(\mathbf{x}_k, \mathbf{y}_k; \xi_k^x\right)\|^2\right].$$

where the term $\hat{\zeta}_v^2$ cannot be guaranteed to be convergent because the step size between the different dimensions of each node is inconsistent and uncontrolled. Noticing that for D-TiAda,

$$\mathbb{E}\left[\left\|\frac{\mathbf{1}^T}{n\bar{v}_{k+1}^{-\alpha}} \tilde{V}_{k+1}^{-\alpha} \odot \nabla_x F\left(\mathbf{x}_k, \mathbf{y}_k; \xi_k^x\right)\right\|^2\right] \leqslant \frac{1}{n}\mathbb{E}\left[\left\|\frac{\tilde{V}_{k+1}^{-\alpha}}{\bar{v}_{k+1}^{-\alpha}}\right\|^2 \|\nabla_x F\left(\mathbf{x}_k, \mathbf{y}_k; \xi_k^x\right)\|^2\right] \leqslant p\zeta_V^2 C^2, \tag{83}$$

Using Lemma 1, we complete the proof. □

**Theorem 3.** *Suppose Assumption 1-5 hold. Let $0 < \alpha < \beta < 1$ and the total iteration satisfy*

$$K = \Omega\left(\max\left\{1, \ \left(\frac{\gamma_x^2\kappa^4}{\gamma_y^2}\right)^{\frac{1}{\alpha-\beta}}, \ \left(\frac{1}{(1-\rho_W)^2}\right)^{\max\left\{\frac{1}{\alpha}, \frac{1}{\beta}\right\}}\right\}\right).$$

*to ensure time-scale separation and quasi-independence of network. For DAS$^2$C with coordinate-wise adaptive stepsize, we have*

$$\frac{1}{K} \sum_{k=0}^{K-1} \mathbb{E}\left[ \left\| \nabla \Phi\left(\bar{x}_k\right) \right\|^2 \right]$$

$$= \tilde{\mathcal{O}}\left( \frac{1}{K^{1-\alpha}} + \frac{1}{\left(1-\rho_W\right)^\alpha K^\alpha} + \frac{1}{K^{1-\beta}} + \frac{1}{\left(1-\rho_W\right) K^\beta} \right) + \mathcal{O}\left( n \left( p\hat{\zeta}_v^2 + \kappa^2 d\hat{\zeta}_u^2 \right) C^2 \right).$$

(84)

*Proof.* With the help of Lemma 12 and the obtained result (73) in the proof of Theorem 2, we can derive the convergence results for DAS$^2$C with coordinate-wise adaptive stepsize. □

**Remark 5.** *In Theorem 3, we show that there is a steady-state error in the upper bound of the coordinate-wise variant of DAS$^2$C depending on the number of nodes and the problem's dimension. However, it's worth noting that the coordinate-wise scheme exhibits strong performance in numerous real-world experiments, particularly for high-dimensional problems (Li et al., 2023) at the cost of increased communication overhead. The observed gap between the theoretical analysis and experimental results can be attributed to our assumption of bounded gradients (c.f., Assumption 4, i.e., $\left\| \nabla_z f_i\left(x, y; \xi_i\right) \right\|^2 \leqslant C$), which hides the information about the dimension of the problem. We believe an interesting direction for future work is to find effective ways to close the gap.*

