# OpenReview forum: "DAS$^2$C: A Distributed Adaptive Minimax Method with Near-Optimal Convergence"
_ICLR.cc/2024/Conference — ICLR 2024 Conference Withdrawn Submission_

### Official Review · Reviewer_Wx7b · 2023-10-30

**Soundness:** 3 good
**Presentation:** 1 poor
**Contribution:** 3 good
**Rating:** 6
**Confidence:** 3

**Summary:**

The paper delves into distributed minimax optimization problems, specifically addressing a min-max problem with costs allocated across network-connected nodes. The authors introduce an adaptive stepsize distributed method, notably agnostic to the problem's inherent parameters.

**Strengths:**

The paper introduces a novel adaptive step size approach tailored for distributed optimization, aiming to ensure consistency among all nodes. Leveraging this strategy, the authors put forth a distributed method, demonstrating its convergence at near-optimal rates.

**Weaknesses:**

The paper's assumption that every stochastic gradient remains bounded presents a restrictive condition.
The overall presentation and structure of the paper need refinement. The paper initiates with equations without offering adequate motivation or a streamlined introduction. This lack of organization is particularly concerning given the dense notation utilized in the work. Additionally, there's a notable absence of discussions surrounding the principal results and the definitions of each parameter. Delving into these results, understanding their implications, and drawing clear comparisons with other works are essential steps that should not be overlooked.

**Questions:**

Regarding the primary result, does it imply that the method converges for all step sizes $\gamma_{x,y}$, provided the iteration count is sufficiently large?

What drove the development of the proposed method? How did you derive those specific updates for the step size?

Is it possible to do away with the bounded gradient assumption? Generally, to demonstrate the convergence of analogous decentralized methods, only the conditions of bounded variance and bounded gradient disagreement are required.

---

> ### Author Response · Authors · 2023-11-22
> **To reviewer Wx7b**
>
> Thanks for the valuable comments and suggestions.
>
>
>
> > __W1:__ The paper's assumption that every stochastic gradient remains bounded presents a restrictive condition.
>
> __Response:__ As discussed in Remark 2, Assumption 4 on bounded stochastic gradient is widely used to establish convergence rates of adaptive methods. Furthermore, to the best of our knowledge, in the field of stochastic nonconvex-strongly-concave minimax optimization, no existing parameter-agnostic method achieves a near-optimal convergence rate while also eliminating the bounded gradient assumption. This remains an open question for future research. See also the Response 1 to all reviewers for a more detailed discussion.
>
>
> > __W2:__ The overall presentation and structure of the paper need refinement. The paper initiates with equations without offering adequate motivation or a streamlined introduction. This lack of organization is particularly concerning given the dense notation utilized in the work. Additionally, there's a notable absence of discussions surrounding the principal results and the definitions of each parameter. Delving into these results, understanding their implications, and drawing clear comparisons with other works are essential steps that should not be overlooked.
>
> __Response:__ Thanks for the useful suggestions. We have now added proper comparison with more related works in the revised version. We will also carefully refine the presentation and structure of the paper for better readability, and add more discussions on the notations and theoretical results in the final version.
>
>
>  > __Q1:__ Regarding the primary result, does it imply that the method converges for all step sizes $\gamma_{x,y}$, provided the iteration count is sufficiently large?
>
> __Response:__ Yes, as shown in Theorem 2, the proposed DAS$^2$C algorithm always converges without the need to design the stepsizes $\gamma_x$ and $\gamma_y$ as in most existing works, which makes it robust to stepsize tuning as evidenced in our experiments. Nevertheless, it is worth to note that the transient time required for time-scale separation depends on the ratio between $\gamma_x$ and $\gamma_y$ as discussed in Remark 4.
>
>
> > __Q2:__ What drove the development of the proposed method? How did you derive those specific updates for the step size?
>
> __Response:__ As discussed in Section 2, we first adapt the centralized TiAda to distributed scenario, which adopts the similar stepsizes with exponential factors $\alpha$ and $\beta$ to ensure adaptive time-scale separation for centralized nonconvex minimax problem. We thus construct counterexamples with lower bound showing that directly applying adaptive methods might lead to non-convergence in distributed settings. By carefully analyzing the average system as shown in Eq. (4), we propose the first distributed adaptive minimax method, named DAS$^2$C, that incorporates an efficient stepsize control mechanism to maintain consistency across local stepsizes, which involves transmission of merely two extra (scalar) variables. The effectiveness of DAS$^2$C have been illustrated in both theoretical and experimental results.
>
> > __Q3:__ Is it possible to do away with the bounded gradient assumption? Generally, to demonstrate the convergence of analogous decentralized methods, only the conditions of bounded variance and bounded gradient disagreement are required.
>
> __Response:__ To the best of our knowledge, in the field of stochastic nonconvex-strongly-concave minimax optimization, no existing parameter-agnostic method achieves a near-optimal convergence rate while also eliminating the bounded gradient assumption. This remains an open question for future research. In fact, this challenge -- being near-rate-optimal and agnostic to parameters without the need for bounded gradients -- persists even in stochastic minimization with strongly-convex objectives. See also the Response 1 to all reviewers for a more detailed discussion.

---

### Official Review · Reviewer_wTqn · 2023-10-30

**Soundness:** 3 good
**Presentation:** 3 good
**Contribution:** 2 fair
**Rating:** 6
**Confidence:** 4

**Summary:**

This paper studied the adaptive minimax method in distributed problems. They proposed a distributed adaptive method DAS2C with time-scale separated stepsize control for minimax optimization. By leveraging the transmission of two extra (scalar) variables, non-convergence issue is solved. For nonconvex-strongly-concave distributed minimax problems, it gets a near-optimal convergence rate of $\tilde{O}  (\epsilon^{4+ \delta})$.

**Strengths:**

1. Minimax is an important optimization problem in machine learning and the study of minimax in a distributed setting is necessary.

2. The paper is organized well. it presents a counterexample to show how to design the algorithm.

3. The strategy is simple to use and extra transmission variables are scalar.

**Weaknesses:**

1. The motivation behind this paper is not clear. What is the advantage of the adaptive method and why do we need these types of methods in the distributed setting?   Although few papers study the application of adaptive methods in distributed minimax problems, are adaptive methods necessary in distributed or federated learning compared with non-adaptive algorithms?

2. Some recent related distributed minimax works are missing.

[1] A faster decentralized algorithm for nonconvex minimax problems, NeurIPS 2021

[2] Taming Communication and Sample Complexities in Decentralized Policy Evaluation for Cooperative Multi-Agent Reinforcement Learning, NeurIPS 2021

[3] FedNest: Federated bilevel, minimax, and compositional optimization. ICML 2022

[4] Decentralized Riemannian Algorithm for Nonconvex Minimax Problems. AAAI 2023

[5] Solving a Class of Non-Convex Minimax Optimization in Federated Learning. NeurIPS 2023.


3. For the convergence results, if this paper focuses on the nonconvex-strongly-concave, results should include the depends on $\kappa$.

4. The baselines in experiments seem to only solve the issues about the design of the adaptive method in distributed learning. It does not present why we need adaptive  algorithms in (minimax) distributed problems.

**Questions:**

1. "$DAS^2C$ is the first distributed adaptive method guaranteeing exact convergence without requiring to know any problem-dependent parameters for nonconvex minimax problems". Could you explain what are the  "problem-dependent parameters "?

2. The convergence is $\tilde{O}  (\epsilon^{4+ \delta})$. What is the result of its centralized counterpart? It seems that related works does not include their term and this result is not as tight as others.

3. In the eq. (4), first equality should be correct. But for the second equality, if you add a projection operator, is the equality still valid?

4. Why $\min _x \max _y 1 / n \sum_{j=1}^n f_i\left(x ; \xi_i+y\right)-\eta\|y\|^2,$ in Robust training of neural network tasks and Generative Adversarial Networks is NC-SC question?

---

> ### Author Response · Authors · 2023-11-22
> **To reviewer wTqn (part 1/2)**
>
> Thanks for the valuable comments and suggestions.
>
> > __W1:__ The motivation behind this paper is not clear. What is the advantage of the adaptive method and why do we need these types of methods in the distributed setting? Although few papers study the application of adaptive methods in distributed minimax problems, are adaptive methods necessary in distributed or federated learning compared with non-adaptive algorithms?
>
> __Response:__ We believe that the application of adaptive methods to distributed minimax problems is an important and underexplored area. We would like to emphasize that: i) distributed optimization has seen significant research progress over the last decade due to its wide applications in various fields; ii) adaptive methods such as AdaGrad and Adam have become the default choice of optimization algorithms in many machine learning applications owing to their robustness to hyper-parameter selection and fast empirical convergence. Therefore, applying adaptive methods as per-node optimizers to distributed optimization algorithms makes sense to improve performance in many applications. More importantly, the design and analysis of distributed adaptive algorithms are challenging due the existence of inconsistency in stepsizes as illustrated by the counterexamples (c.f., Figure 1) and lower bound analysis (c.f., Theorem 1) in Section 2.
>
>
> > __W2:__ Some recent related distributed minimax works are missing.
>
> __Response:__  Thank you for bring these papers to our attention. As listed by the reviewer, the works in [1-3, 5] use variance reduction methods to improve convergence at the cost of additional computation or memory. The work in [4] considers
> Riemannian optimization, which beyond the scope of this paper. We have added all the related references in the introduction and made proper comparison among them. Please refer to the general Response 3 to all reviewers for more details.
>
>
> > __W3:__ For the convergence results, if this paper focuses on the nonconvex-strongly-concave, results should include the depends on $\kappa$.
>
> __Response:__ Thanks for the valuable suggestion. The dependence on $\kappa$ of DAS$^2$C can be found in Eq. (75), which is comparable with the centralized TiAda algorithm. It is worth to note that the proposed DAS$^2$C algorithm exhibits parameter-agnostic capability without requiring to know the smoothness or strongly-convexity of the objective functions, which makes it robust to hyper-parameter tuning.
>
> > __W4:__ The baselines in experiments seem to only solve the issues about the design of the adaptive method in distributed learning. It does not present why we need adaptive algorithms in (minimax) distributed problems.
>
> __Response:__ In general, distributed algorithms can accelerate optimization/learning tasks in a data-parallelized manner comparing to centralized method, thus have wide applications in various fields.  Given the robustness to hyperparameter selection and fast empirical convergence of adaptive methods, applying adaptive methods as per-node optimizers to distributed optimization algorithms makes sense to improve performance in many applications. We will consider to add more comprehensive comparisons with the non-adaptive methods in the final version.
>
> (continued below)

---

> > ### Author Response · Authors · 2023-11-22
> > **To reviewer wTqn (part 2/2)**
> >
> > > __Q1:__ Could you explain what are the "problem-dependent parameters "?
> >
> > __Response:__ By problem-dependent parameters, we mean the properties of the minimax problem such as the smoothness and strong-concavity parameters, which are often unknown or difficult to estimate in real-world tasks like training deep neural networks. As illustrated in Theorem 2, DAS$^2$C always converge to a stationary point without requiring to know any problem-dependent parameters for setting the algorithmic hyper-parameters. In contrast, most of the existing distributed adaptive methods need to artificially utilize these parameters to design the stepsizes $\gamma_x$ and $\gamma_y$.
> >
> > > __Q2:__ The convergence is $\tilde{ \mathcal{O}} \left( \epsilon^{ -\left( 4 + \delta \right) } \right) $. What is the result of its centralized counterpart? It seems that related works does not include their term and this result is not as tight as others.
> >
> > __Response:__ As shown in Theorem 2, the obtained convergence rate can be reduced to the result of centralized method TiAda by letting $\rho_w = 0$, which indicates a fully connected graph. It is worth to note that, for the methods based on stochastic gradient descent ascent without adopting variance reduction methods, the lower bound of sampling complexity for nonconvex-strongly-concave problems is $\tilde{\mathcal{O}} \left( \epsilon ^{-4} \right) $ (Li et al., 2021; Zhang et al., 2021). The obtained complexity $\tilde{\mathcal{O}} \left( \epsilon ^{-\left( 4+\delta \right)} \right)$ for any $\delta>0$ is near-optimal comparing to the best-known result, and recovers the centralized TiAda algorithm as special case.
> >
> > * H. Li, Y. Tian, J. Zhang and A. Jadbabaie. Complexity lower bounds for nonconvex-strongly-concave min-max optimization. NeurIPS 2021.
> > * S. Zhang, J. Yang, C. Guzman, N. Kiyavash and N. He. The complexity of nonconvex-strongly-concave minimax optimization. In Uncertainty in Artificial Intelligence 2021.
> >
> > > __Q3:__ In the eq. (4), first equality should be correct. But for the second equality, if you add a projection operator, is the equality still valid?
> >
> > __Response:__ Thank you for pointing this out. In Eq.(4), the average system is only used to illustrate the existence of inconsistency in stepsizes. To improve readability, we have removed the second equality in the revised version.
> >
> > > __Q4:__ Why $\min_x \max_y \,1/n\sum_{i=1}^n{f_i\left( x;\xi _i+y \right) -\eta |\left\| y |\right\| ^2}$, in Robust training of neural network tasks and Generative Adversarial Networks is NC-SC question?
> >
> > __Response:__ For robust training of neural network, if $\gamma$ is larger enough, it can be verified that the objective function $1/n\sum_{i=1}^n{f_i\left( x;\xi _i+y \right)} -\eta |\left\| y |\right\| ^2$ is nonconvex-strongly-concave in $x$ and $y$ respectively. For training of GANs, we used WGAN-GP loss with penalty on gradient (Gulrajani et al., 2017), which is known as a nonconvex-nonconcave objective function. The purpose of including the GANs example is to show the effectiveness of our proposed distributed adaptive method even in real world scenarios.

---

### Official Review · Reviewer_GJ7M · 2023-11-01

**Soundness:** 1 poor
**Presentation:** 1 poor
**Contribution:** 1 poor
**Rating:** 3
**Confidence:** 5

**Summary:**

This paper investigates the decentralized minimax optimization problem. It developed an adaptive algorithm. However, it missed some important references and introduced strong assumptions and have errors in convergence analysis.

**Strengths:**

1. The problem studied in interesting.

2. The counterexample is good.

**Weaknesses:**

1. This paper missed some state-of-the-art literature.


[1] A Faster Decentralized Algorithm for Nonconvex Minimax Problems
[2]  Taming Communication and Sample Complexities in Decentralized Policy Evaluation for Cooperative Multi-Agent Reinforcement Learning
[3] Decentralized stochastic gradient descent ascent for finite-sum minimax problems
[4] Jointly Improving the Sample and Communication Complexities in Decentralized Stochastic Minimax Optimization

2. This paper introduces strong assumptions so that the proof is simplified too much. In particular, it assumes the gradient is upper-bounded in Assumption 4, which is not used in the original TiAda paper. In addition, assuming that the function is strongly concave and has a bounded gradient is not common because the simple quadratic function does not satisfy this assumption.

3. There are some errors in the proof. $\mathcal{P}$ is not a linear operator so eq (36) is not correct.

4. This paper didn't compare with the aforementioned SOTA algorithms.

**Questions:**

1. This paper missed some state-of-the-art literature.


[1] A Faster Decentralized Algorithm for Nonconvex Minimax Problems
[2]  Taming Communication and Sample Complexities in Decentralized Policy Evaluation for Cooperative Multi-Agent Reinforcement Learning
[3] Decentralized stochastic gradient descent ascent for finite-sum minimax problems
[4] Jointly Improving the Sample and Communication Complexities in Decentralized Stochastic Minimax Optimization

2. This paper introduces strong assumptions so that the proof is simplified too much. In particular, it assumes the gradient is upper-bounded in Assumption 4, which is not used in the original TiAda paper. In addition, assuming that the function is strongly concave and has a bounded gradient is not common because the simple quadratic function does not satisfy this assumption.

3. There are some errors in the proof. $\mathcal{P}$ is not a linear operator so eq (36) is not correct.

4. This paper didn't compare with the aforementioned SOTA algorithms.

---

> ### Author Response · Authors · 2023-11-22
> **To reviewer GJ7M**
>
> Thanks for the valuable comments and suggestions.
>
> > __W1:__ This paper missed some state-of-the-art literature.
>
> __Response:__ Thank you for bring these papers to our attention. We have carefully read through them and found that all of them use variance reduction techniques to improve convergence performance at the cost of additional computation or memory. It should be noted that our method can be also easily integrated with VR technique, leading to a similar improvement. We have now added all the related references in the introduction and made proper comparison among them. Please refer to the general Response 3 to all reviewers for more details.
>
> > __W2:__ This paper introduces strong assumptions so that the proof is simplified too much. In particular, it assumes the gradient is upper-bounded in Assumption 4, which is not used in the original TiAda paper. In addition, assuming that the function is strongly concave and has a bounded gradient is not common because the simple quadratic function does not satisfy this assumption.
>
> __Response:__ We respectfully disagree with the reviewer. As discussed in Remark 2, Assumption 4 on bounded stochastic gradient is widely used for establishing convergence rates of adaptive methods. In fact, this assumption is also imposed in TiAda (c.f., Assumption 3.4). Furthermore, it should be noted that, with compact and convex constraint set $\mathcal{Y}$, the assumptions on bounded gradient and strong concavity can be satisfied. Please also refer to the Response 1 to all reviewers for a more detailed discussion.
>
>
> > __W3:__ There are some errors in the proof. $\mathcal{P} $ is not a linear operator so eq (36) is not correct.
>
> __Response:__ Thanks for pointing this out. Indeed, in Eq.~(36), we used the standard non-expansiveness of projection operator, i.e., $|\left\| \mathcal{P} _{\mathcal{Y}}\left( y \right) -y^* |\right\| ^2\leqslant |\left\| y-y^* |\right\| ^2- |\left\| \mathcal{P} _{\mathcal{Y}}\left( y \right) -y |\right\| ^2$ (c.f., Lemma 1 in Nedich et al., 2010), in the proof of Lemma 7. We have modified the statement about projection in the revised version.
>
> * A. Nedic, A. Ozdaglar, P.A. Parrilo. Constrained consensus and optimization in multi-agent networks. IEEE TAC 2010.
>
> > __W4:__ This paper didn't compare with the aforementioned SOTA algorithms.
>
> __Response:__ All of the above references use variance reduction methods to improve convergence at the cost of additional computation or memory. Instead, for fair comparison, we mainly focus on __distributed adaptive methods__ that exhibit parameter-agnostic capability for minimax problems, such as distributed version of TiAda and NeAda, in this work. These methods use mini-batch stochastic gradient without requiring to tune the batch-size or additional memory.  We have added all the related references pointed out by the reviewer in the introduction and made proper comparison among them. See also the Response 2 and 3 to all reviewers for more details.

---

> ### Author Response · Authors · 2023-11-23
> **Sincerely hoping for feedback from Reviewer GJ7M. Has our response addressed your concerns?**
>
> Dear Reviewer GJ7M,
>
> The discussion period is ending soon within one day, and we appreciate it if we can have an opportunity to engage with the reviewer. Please let us know whether our response has addressed your concerns.
>
> Best regards,
>
> The Authors

---

### Official Review · Reviewer_x9Sp · 2023-11-02

**Soundness:** 2 fair
**Presentation:** 2 fair
**Contribution:** 2 fair
**Rating:** 6
**Confidence:** 5

**Summary:**

This paper studied the decentralized distributed nonconvex-strongly-concave minimax problems, and proposed an efficient adaptive decentralized algorithm to solve these problems. Theoretically, it proved that the proposed algorithm obtain a near-optimal convergence rate. Experimentally, it provided some experimental results to demonstrate the efficiency of the proposed algorithms.

**Strengths:**

This paper studied the decentralized distributed nonconvex-strongly-concave minimax problems, and proposed an efficient adaptive decentralized algorithm to solve these problems. Theoretically, it proved that the proposed algorithm obtain a near-optimal convergence rate. Experimentally, it provided some experimental results to demonstrate the efficiency of the proposed algorithms.

**Weaknesses:**

Although the proposed DAS2C algorithm is the first decentralized distributed adaptive method for nonconvex minimax problem, it basically extends the existing adaptive method [1] to decentralized distributed settings. Meanwhile, the DAS2C algorithm can address the issue of inconsistent stepsizes across different nodes by communicating the adaptive step-sizes, which basically follows the same trick in the adaptive federated learning.


[1] Li, X., YANG, J., and He, N. (2023). Tiada: A time-scale adaptive algorithm for nonconvex minimax optimization. In The Eleventh International Conference on Learning Representations.

**Questions:**

1)	In DAS2C  algorithm, why use the exponential factors satisfying $\beta < \alpha$ ?

2)	The DAS2C algorithm needs some stricter assumptions (e.g., $f_i$ is second-order Lipschitz continuous for $y$) than the existing decentralized minimax optimization methods.

3)	In the experiments, the authors should add some existing  decentralized  minimax optimization algorithms such as the DPOSG of [2] as the comparison methods.

[2] Liu, M., Zhang, W., Mroueh, Y., Cui, X., Ross, J., Yang, T., and Das, P. (2020). A decentralized parallel algorithm for training generative adversarial nets. Advances in Neural Information Processing Systems, 33:11056–11070.

4) Some related references are missing. E.g.,

[a] A Simple and Efficient Stochastic Algorithm for Decentralized Nonconvex-Strongly-Concave Minimax Optimization

[b] Jointly Improving the Sample and Communication Complexities in Decentralized Stochastic Minimax Optimization

[c] A faster decentralized algorithm for nonconvex minimax problems

---

> ### Author Response · Authors · 2023-11-22
> **To reviewer x9Sp (part 1/2)**
>
> Thanks for the valuable comments and suggestions.
>
> >__W1:__ Although the proposed DAS$^2$C algorithm is the first decentralized distributed adaptive method for nonconvex minimax problem, it basically extends the existing adaptive method [1] to decentralized distributed settings. Meanwhile, the DAS$^2$C algorithm can address the issue of inconsistent stepsizes across different nodes by communicating the adaptive step-sizes, which basically follows the same trick in the adaptive federated learning.
>
> __Response:__ We respectfully disagree with the reviewer on the algorithm design. In this work, we propose the *first*  distributed adaptive minimax method, named DAS$^2$C, that incorporates an efficient stepsize control mechanism to maintain consistency across local stepsizes, which involves transmission of merely two extra (scalar) variables. This design is non-trivial in the sense that we construct counterexamples showing that directly applying centralized adaptive methods in distributed settings might lead to non-convergence (c.f., Figure 1 and Theorem 1). We emphasize that the proposed stepsize control mechanism is significantly different from adaptive federated learning algorithm, such as AdaFGDA (Huang et al., 2022) where a centralized server node averages all the variables and there is thus no inconsistency in stepsizes (c.f., Line 5 and 6 of Algorithm 1 in Huang et al., 2022).  In this regard, the existing federated adaptive method is not readily applicable to decentralized scenarios. More importantly, our proposed algorithm achieves near-optimal convergence rate matching that of centralized TiAda algorithm, and exhibits parameter-agnostic capability, i.e., without requiring knowledge of problem-dependent parameters, which is also not available in AdaFGDA.
>
> * F. Huang. Adaptive federated minimax optimization with lower complexities. arXiv 2022.
>
>
> >__Q1:__ In DAS$^2$C algorithm, why use the exponential factors satisfying $\beta < \alpha$?
>
> __Response:__ The exponential factors $\beta <\alpha$ are designed to satisfy adaptive time-scale separation in stepsizes between the minimization and maximization processes, which is necessary for the convergence of adaptive minimax methods with nonconvex-strongly-concave objective (NC-SC) functions, as illustrated in Lin et al. (2020); Yang et al. (2022). Intuitively, for NC-SC problem, the maximization process needs to be faster than minimization to ensure that the strongly-concave function $f(\cdot, y)$ can be solved with sufficient precision to ensure convergence of minimax problem. This is also the reason why some adaptive minimax algorithms, such as NeAda (Yang et al., 2022), require an extra inner loop for dual variable. Instead, through the design of $\beta <\alpha$, we propose the single-loop algorithm DAS$^2$C exhibiting timescale separation in stepsizes.
>
> * T. Lin, C. Jin and M. Jordan. On gradient descent ascent for nonconvex-concave minimax problems. ICML 2020.
> * J. Yang, X. Li and N. He. Nest your adaptive algorithm for parameter-agnostic nonconvex minimax optimization. NuerIPS 2022.
>
>
>
> >__Q2:__ The DAS$^2$C algorithm needs some stricter assumptions (e.g., $f_i$ is second-order Lipschitz continuous for $y$) than the existing decentralized minimax optimization methods.
>
> __Response:__ Assumption 3 on second-order Lipschitz continuity for $y$ is used in existing work to achieve improved convergence rate for minimax problem (Chen et al., 2021; Li et al., 2023). In specific, as discussed in Remark 2, Assumption 3 ensures that $y^{*} (\cdot)$ is smooth (c.f., Eq.~(46)), which is essential for achieving (near) optimal convergence rate. However, without this assumption, Lin et al. (2020) only show a worse complexity of $\tilde{\mathcal{O}}\left( \epsilon ^{-5} \right)$ without a large batchsize. For further discussion of assumptions, please refer to Response 1 to all reviewers.
>
> * T. Chen, Y. Sun and W. Yin. Closing the gap: Tighter analysis of alternating stochastic gradient methods for bilevel problems. NeurIPS 2021.
> * X. Li, J. Yang and N. He. Tiada: A time-scale adaptive algorithm for nonconvex minimax optimization. ICLR 2023.
> * T. Lin, C. Jin and M. Jordan. On gradient descent ascent for nonconvex-concave minimax problems. ICML 2020.
>
> (continued below)

---

> > ### Author Response · Authors · 2023-11-22
> > **To reviewer x9Sp (part 2/2)**
> >
> > >__Q3:__ In the experiments, the authors should add some existing decentralized minimax optimization algorithms such as the DPOSG of [2] as the comparison methods.
> >
> > __Response:__ We found that DPOSG is a decentralized optimistic gradient descent ascent algorithm with constant stepsizes that utilizes the gradient information of two consecutive steps for updating the variables. Instead, in the experiments of this work, we mainly focused on the comparison of distributed adaptive methods that exhibit __parameter-agnostic capability__ for minimax problems, such as the proposed DAS$^2$C, distributed TiAda and NeAda. We have a brief comparison with DPOSG in the introduction and will also add experimental results for comparison with DPOSG  in the final version.
> >
> > >__Q4:__ Some related references are missing.
> >
> > __Response:__ Thank you for bring these papers to our attention. We have added all the above references in the introduction and made proper comparison among them. Please refer to the general Response 3 to all reviewers for more details.

---

> > > ### Comment · Reviewer_x9Sp · 2023-11-23
> > > **Reply to Authors**
> > >
> > > Thanks for your responses. My concerns have basically been addressed, so I will increase my score.

---

### Author Response · Authors · 2023-11-22
**General response to all reviewers (part 1/2)**

We thank the reviewers for their valuable comments and suggestions. Here, we would like to further clarify the contributions of this work in the following three aspects: i) we are the first to propose the distributed adaptive method for nonconvex-strongly-concave minimax problems, named DAS$^2$C, guaranteeing near-optimal convergence __without requiring to know any problem-dependent parameters__, which calls for non-trivial convergence analysis comparing to centralized counterparts; ii) by carefully constructing counterexamples, we prove the existence of a constant steady-state error in both the lower and upper bounds when directly applying a centralized adaptive method, necessitating the design of a novel stepsize control strategy; iii) we conduct extensive experiments on real-world datasets to verify our theoretical findings and the effectiveness of DAS$^2$C on a variety of tasks.
In what follows, we provide the general replies to the common comments raised by most of the reviewers, which is followed by the specific one-to-one reply to all the comments of the reviewers.

>__Q1:__  On the assumption of bounded gradient.
>
**Response 1:** As discussed in Remark 2, Assumption 4 on bounded stochastic gradient is widely used to establish convergence rates of __adaptive methods__ (Zou et al., 2019; Kavis et al., 2022), which can be easily satisfied by imposing constraints of compact domain of $y$ in many real-world tasks. For example, in neural networks with rectified activation, imposing bound on $y$ does not affect expressiveness because of its scale-invariance property (Dinh et al., 2017). Wasserstein GANs (Arjovsky et al., 2017) also use projections on the critic to constrain the weights to a small cube around the origin.
To the best of our knowledge, in the field of stochastic nonconvex-strongly-concave minimax optimization, no existing parameter-agnostic method achieves a near-optimal convergence rate while also eliminating the bounded gradient assumption. In fact, this challenge -- being near-rate-optimal and agnostic to parameters without the need for bounded gradients -- persists even in stochastic minimization with strongly-convex objectives (Orvieto et al., 2022), a sub-problem of our minimax problem. This remains an open question for future research.

* F. Zou, L. Shen, Z. Jie, W. Zhang and W. Liu. A sufficient condition for convergences of adam and rmsprop. CVPR 2019.
* A. Kavis, K.Y. Levy and V. Cevher. High probability bounds for a class of nonconvex algorithms with adagrad stepsize. ICLR 2021.
* L. Dinh, R. Pascanu, S. Bengio and Y.Bengio. Sharp minima can generalize for deep nets. ICML 2017.
* M. Arjovsky, S. Chintala and L. Bottou. Wasserstein generative adversarial networks. ICML 2017.
* A. Orvieto, S lacoste-Julien and N. Loizou. Dynamics of SGD with Stochastic Polyak Stepsizes: Truly Adaptive Variants and Convergence to Exact Solution. NeurIPS 2022.


>__Q2:__ On the experimental comparison with existing works
>
__Response 2:__ In our experiments, for fair performance comparison, we are mainly focused on __distributed adaptive methods__ that exhibit parameter-agnostic capability for minimax problems, such as distributed version of TiAda and NeAda. These methods use mini-batch stochastic gradient without requiring to tune the batch-size or additional memory. We believe that it is also interesting and important to make comparison with other existing methods, such as non-adaptive methods and VR-based methods, which we will consider to include in the final version.

(continued below)

---

> ### Author Response · Authors · 2023-11-22
> **General response to all reviewers (part 2/2)**
>
> >__Q3:__ On some missing references.
> >>1. L. Chen, H. Ye and L. Luo. Simple and Efficient Stochastic Algorithm for Decentralized Nonconvex-Strongly-Concave Minimax Optimization. arXiv 2023
> >>2. X. Zhang, Z. Liu, J. Liu, Z. Zhu and S. Lu. Taming Communication and Sample Complexities in Decentralized Policy Evaluation for Cooperative Multi-Agent Reinforcement Learning. NeurIPS 2021.
> >>3. H. Gao. Decentralized stochastic gradient descent ascent for finite-sum minimax problems. arXiv 2022.
> >>4. X. Zhang, G. Mancino-Ball, N.S. Aybat and Y. Xu. Jointly Improving the Sample and Communication Complexities in Decentralized Stochastic Minimax Optimization. arXiv 2023.
> >>5. W. Xian, F. Huang, Y. Zhang and H. Huang. A faster decentralized algorithm for nonconvex minimax problems. NeurIPS 2021.
> >>6. D.A. Tarzanagh, M.n Li, C. Thrampoulidis and S. Oymak. FedNest: Federated bilevel, minimax, and compositional optimization. ICML 2022.
> >>7. X. Wu, J. Sun, Z. Hu, A. Zhang and H. Huang. Solving a Class of Non-Convex Minimax Optimization in Federated Learning. NeurIPS 2023.
> >>8. X. Wu, Z. Hu and H. Huang. Decentralized Riemannian Algorithm for Nonconvex Minimax Problems. AAAI 2023.
>
> __Response 3:__ We carefully checked and compared all the references pointed out by the reviewers. The works in [1-7] use variance reduction (VR) methods to estimate local gradients, which are generally recognized to improve the convergence rate at the cost of additional computation or memory. For instance, the proposed algorithm DREAM in Chen et al. (2023) adopts recursive-gradient requiring a large batch-size of $\varOmega \left( \epsilon ^{-2} \right) $ or full gradient, which may exceed the memory limit of the node in practice. Instead, in this work, we focus on the methods using mini-batch stochastic gradient without requiring to tune the batch-size and additional memory. We expect that our approach can be also integrated with VR techniques, yielding an improvement of sampling complexity as the above work. Additionally, the work in [8] considers Riemannian optimization, which beyond the scope of this paper. we have added all the related references pointed out by the reviewers in the introduction and made proper comparison among them in the revised version.

---

> ### Author Response · Authors · 2023-11-23
> **The author-reviewer discussion period is ending. Sincerely hoping for feedback from the reviewers**
>
> Dear Reviewers,
>
> As the author/reviewer discussion period for ICLR submissions is drawing to a close soon, we would like to take this opportunity to encourage your engagement with our submission. We greatly value your insights and feedback. If you have already reviewed our response, we kindly ask if you could let us know whether it has sufficiently addressed your concerns. Your input is crucial for us to enhance and refine our work.
>
> Thank you for your time.
>
> Best regards,
>
> The Authors